# Spatially heterogeneous acetylcholine dynamics in the striatum promote behavioral flexibility

Gideon A. Sarpong [1], Rachel Pass [1], Kavinda Liyanagama [1], Kang-Yu Chu [1], Kiyoto Kurima[1], Yumiko Akamine [1], Julie A. Chouinard [1], Loren L. Looger [2] & Jeffery R. Wickens [1] ✉

Being able to switch from established choices to new alternatives when conditions change – behavioral flexibility – is essential for survival. Cholinergic signaling in the striatum contributes to such flexible behavior, yet the timing and spatial organization of acetylcholine release during contingency changes remain unclear, limiting conceptual understanding of its role in behavioral flexibility. Using a genetically encoded acetylcholine sensor and 2-photon imaging in the dorsal striatum of behaving mice, we visualized acetylcholine dynamics during acquisition and reversal learning in a virtual reality Y-maze. Rewarded outcomes evoked phasic decreases in acetylcholine, whereas unexpected non-reward following reversal triggered widespread increases that predicted lose-shift behavior. Targeted inhibition of cholinergic interneurons reduced this adaptive response. Spatial analysis revealed heterogeneous, temporally distinct signals forming functionally diverse microdomains. These findings suggest that widespread and focal acetylcholine release during unexpected outcomes promotes adaptive response shifts, offering a mechanistic framework for understanding disorders such as addiction and obsessive-compulsive rituals.

Flexible behavior is essential for survival in changing environments. It requires ceasing previously reinforced responses to explore alternatives[1,2] and involves multiple brain regions, such as the prefrontal cortex[3–7] and the striatum[8–12]. Here, we define behavioral flexibility as the ability to adjust behavior in response to changes in environmental contingencies. Thus, behavioral flexibility requires adopting a new strategy, which is a step beyond learning under constant rules. Cholinergic interneurons (CINs), the prime source of acetylcholine (ACh) in the striatum[13–15], are important for behavioral flexibility. Although striatal cholinergic signaling has been implicated in flexible behaviors[16–23], the precise timing and spatial patterns of striatal ACh release during reversal learning, a paradigm of behavioral flexibility, remain uncharacter-

ized. In particular, high-resolution measurements of ACh spatiotemporal dynamics during behavioral tasks requiring flexible responses have been lacking.

To address this gap, we monitored striatal ACh dynamics in awake, head-fixed mice performing a virtual-reality response learning (VR-RL) task during initial acquisition and reversal learning, where reversals introduced unexpected non-reward. Using two-photon microscopy through a GRIN lens implanted over the dorsal striatum (DS) and a genetically encoded fluorescent ACh sensor, we visualized ACh release with high temporal and spatial precision. Following reversal, unexpected non-reward evoked widespread increases in ACh release, characterized by distinct spatiotemporal dynamics that predicted lose-shift behavior. These increases were preferentially more

[1]Okinawa Institute of Science and Technology Graduate University, Okinawa, Japan. [2]Howard Hughes Medical Institute, University of California, San Diego, San Diego, California, USA. ✉e-mail: wickens@oist.jp

medially distributed and showed a trend toward the anterior portions of the striatum. Spatial tiling analysis of 2-photon images revealed heterogeneous and temporally distinct ACh signals across the dorsal striatum, identifying functionally diverse microdomains with outcome-specific response profiles. We further investigated the role of striatal CINs and found that chemogenetic inhibition of these neurons disrupted the adaptive shift in behavior following reversal. Together, these findings elucidate the spatiotemporal organization of striatal ACh signaling and demonstrate its essential role in adaptive decision making and flexible responses.

## Results

### Head-fixed mice develop a preference for the rewarded arm in a virtual Y-maze task

We first confirmed that awake, head-fixed mice could learn to choose the rewarded arm in a virtual reality Y-maze (Fig. 1a). The rewarded arm of the maze was not marked by cues, and mice had to explore both arms of the maze and learn by trial-and-error which arm was rewarded. A correct choice triggered reward delivery immediately after entry into the outcome zone (Fig. 1b). An incorrect choice triggered a black screen, signifying no reward. Mice were trained in daily sessions of ~40

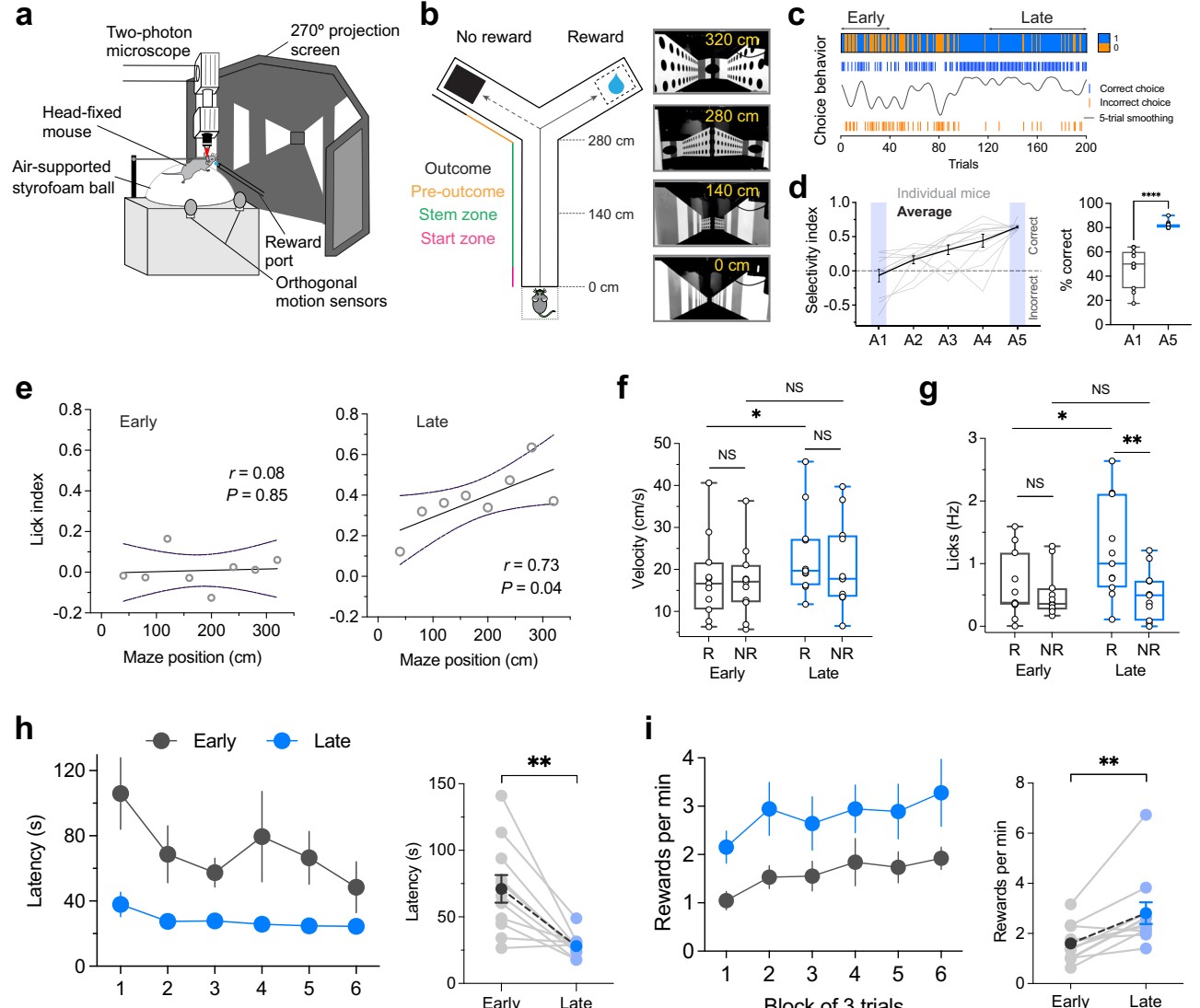

**Fig. 1 | Mice choice performance and adaptive motivation in the VR-RL task.**
**a** Schematic of the experimental setup. **b** Left, design of the VR-based Y-maze (2D linear track; 350 × 30 cm). Right, Representative screenshots of the virtual environment at different maze positions. **c** Example choice behavior from a representative mouse. Blue and orange ticks indicate correct and incorrect trials, respectively; the black trace shows a five-trial moving average of choice performance. The upper heatmap depicts trial-by-trial performance (1 = correct; 0 = incorrect). **d** Left, Maze-arm selectivity indices across training sessions (A1–A5) for all mice (n = 11). Gray lines, individual means; black, group mean ± S.E.M. Note the improved ratio of correct choices with learning (two-tailed, paired t test, P = 3.81 × 10⁻⁵). **e** Mean lick frequency index vs maze position (0–320 cm, 40 cm bins) across sessions. Anticipatory licking correlated positively with maze position in late (simple linear regression, Pearson's correlation coefficient (r) = 0.73, P = 0.04, n = 11) but not early learning (r = 0.08, P = 0.85, n = 11). Solid lines, linear

fits; dotted lines, 95% confidence intervals. **f** Box plot showing the distribution of approach velocity to reward and no-reward arms (n = 11). Repeated measures ANOVA: main effect of session (F(1,20) = 11.69, P = 0.0027), outcome (F(1,20) = 0.17, P = 0.68); interaction (F(1,20) = 0.28, P = 0.60). **g** Anticipatory licking in the pre-outcome zone (280–320 cm) for reward and no-reward arms (n = 11 mice): main effect of outcome (F(1,20) = 5.14, P = 0.03); session (F(1,20) = 3.51, P = 0.076); interaction (F(1,20) = 3.78, P = 0.066) in repeated measures ANOVA. **h** Trial latency broken down by trial block (first 6 block of trials, 3 trials/block). Trial latency decreased with learning (two-tailed, paired t test, P = 0.001). **i**, Reward rate increased across sessions (two-tailed, paired t test, P = 0.003). Transparent dots, individual mouse; solid dots, group mean ± S.E.M. (n = 11). In box plots, center lines depict the median, box limits represent the 25th and 75th percentiles, and whiskers, data range. *P < 0.05, **P < 0.01, and ****P < 0.0001, NS, not significant. Source data are provided as a Source Data file.

trials until they reached the acquisition criterion of 80% correct choices in at least one session, after which the rewarded side was reversed.

Over successive sessions preceding the reversal, mice progressively increased the proportion of correct choices (Fig. 1c, d), indicating successful acquisition of the rewarded arm contingency. In the late learning phase, defined as the final session before reversal, mice achieved an average correct choice rate of 82.1% ± 0.9% (mean ± S.E.M., $P = 3.81 \times 10^{-5}$, $n = 11$ mice, Fig. 1d and Supplementary Fig. 1a). Across sessions, mice displayed licking movements while approaching the reward location, suggesting anticipation of the reward. Increases in directional preference in anticipatory licking were positively correlated with advancement in the maze (Fig. 1e; simple linear regression, Pearson's $r = 0.73$, $P = 0.04$, $n = 11$), signifying that mice had learned the maze structure and accurately predicted the reward location[24,25].

Across the population, trained mice showed reduced trial latencies and increased velocity when approaching the maze arms, indicating heightened motivation and reward-driven decision-making (Fig. 1f–i and Supplementary Fig. 1b–d). Average running velocity increased from early to late sessions, with a trend toward higher velocities in rewarded trials, suggesting that motivational drive strengthens with task experience (Fig. 1f; main effect of session (F(1,20) = 11.69, $P = 0.0027$); main effect of outcome (F(1,20) = 0.17, $P = 0.68$); session×outcome interaction (F(1,20) = 0.28, $P = 0.60$), repeated measures ANOVA). Similarly, anticipatory licking was robustly driven by reward outcome across learning, with a trend toward higher overall levels in late sessions (Fig. 1g; main effect of outcome (F(1,20) = 5.14, $P = 0.03$); main effect of session (F(1,20) = 3.51, $P = 0.076$); session×outcome interaction (F(1,20) = 3.78, $P = 0.066$). The overall increase in locomotor velocity and reduced trial latencies (Fig. 1h; two-tailed, paired $t$ test, $P = 0.001$) resulted in a higher reward rate across sessions (Fig. 1i; two-tailed, paired $t$ test, $P = 0.003$).

### Dorsal striatal ACh transients contain reward-related signals

We observed the time course of ACh transients in response to trial outcomes during maze learning using in vivo two-photon imaging. Adult mice ($n = 11$) received stereotactic injections of adeno-associated virus (AAV) expressing the ACh sensor iAChSnFR[26,27] under control of the human *synapsin-1* promoter with Tet *trans*-activator amplification[28] into the DS and had a 1 mm diameter gradient index (GRIN) lens implanted above the injected region (Fig. 2a) (see Methods). iAChSnFR is derived from a microbial periplasmic binding protein (PBP), and like other sensors derived from PBPs[29] has rise and decay kinetics on a millisecond time scale, thus permitting useful imaging at >1 kHz. Previous studies using purified iAChSnFR and stopped-flow experiments confirmed that activation has a time constant of ~140 ms at 10 μM ACh and just a few ms at 1 mM ACh, with $k_{on}$ of 0.62 μM$^{-1}$s$^{-1}$ and $k_{off}$ of 0.73 s$^{-1}$ [26,27].

Two-photon images revealed expression of iAChSnFR in the striatum, on the surface of somata as well as neuropil (Fig. 2b and Supplementary Fig. 10). Sensitivity of iAChSnFR to ACh was separately confirmed by fluorescence responses to exogenously added ACh in striatal slices from injected mice. To validate sensor specificity, we first applied exogenous ACh to brain slices in ACSF, followed by application of neostigmine (50 μM), an acetylcholinesterase inhibitor, 15–20 min later, which blocked the response to ACh (Fig. 2c, $n = 4$ mice). We utilized in vivo imaging at 30 Hz frame rate to measure the time course of spatially averaged ACh levels during the outcome period (time window, 0-1 s) after rewarded correct or unrewarded incorrect outcomes (Fig. 2d, e). Correct choices followed by reward outcomes caused transient decreases in the fluorescence of iAChSnFR, indicating dips in ACh in response to reward (Fig. 2e). The features of these dips in ACh levels (Fig. 2f, g; dip amplitude, − 1.64 ± 0.07 z-scored Δ*F/F*; onset, 0.23 ± 0.01 s; dip latency *i.e.*, time to dip, 0.48 ± 0.03 s, mean ± S.E.M., $n = 11$) are consistent with pauses in tonic activity of striatal tonically active neurons (TANs, putative CINs)[30–33] and similar to dip latencies observed in ACh release[34]. This timing is also consistent with the

timecourse of pauses in firing activity of CINs (onset around 0.16 – 0.20 s and duration 0.20 – 1.0 s)[32,35–37] when combined with the kinetics of ACh diffusion and hydrolysis[38].

In contrast to rewarded outcomes, unrewarded incorrect outcomes evoked dips significantly smaller than those evoked by reward[31,39] (Fig. 2f, $P = 1.65 \times 10^{-5}$; dip amplitude, − 0.89 ± 0.10 z-scored Δ*F/F*; onset, 0.25 ± 0.03 s; and latency, 0.65 ± 0.06 s, mean ± S.E.M.; Fig. 2g, two-tailed, paired $t$ test, $P = 0.03$, $n = 11$). We also noted some trial-to-trial variability in reward responses (example mouse in Fig. 2e, left; Supplementary Fig. 2a) as previously reported[22,31], which was greater than measurement noise (confirmed by comparison with ACh-insensitive tdTomato signals; Supplementary Fig. 2). Notably, tdTomato was imaged in a subset of trials and did not display the large positive-going transients observed with iAChSnFR, supporting the conclusion that our reported ACh signals reflect genuine cholinergic activity rather than measurement noise[40,41]. These fluctuations in ACh may encode within-session variations in behavioral contexts as perceived by the mouse, potentially reflecting dynamic recruitment of different cholinergic response modes depending on contextual or internal state factors. As mice approached the outcome point of the maze, spatially averaged ACh signals increased progressively until the mouse reached the outcome zone (Supplementary Fig. 3a). The increase was similar in rewarded and non-rewarded outcomes. Across trials, ACh signals were classified according to whether linear regression analysis showed a significant positive or negative correlation during the last 5 s preceding the trial outcome. The greatest proportion of rewarded and unrewarded trials showed positive ramping (52% and 48%, respectively), and less commonly negative (32% and 31%, respectively) or non-significant changes (Supplementary Fig. 3b, c). These distributions were not statistically different across conditions ($X^2$ test: $X^2(2) = 1.52$, $P = 0.46$).

### Spatiotemporal patterns of functionally heterogeneous ACh transients in DS

Spatiotemporal analysis of ACh dynamics from two-photon imaging revealed heterogeneous release patterns linked to distinct behavioral outcomes. Although traditionally considered a volume conducted signal, recent findings indicate more spatially localized release of ACh[27,38,42,43]. By tiling the field of view into a matrix of quadrats (~ 50 × 50 μm, Fig. 3a), we uncovered spatially heterogeneous patterns of task-related ACh transients in the DS, previously unexplored in this context. To characterize the release patterns, we performed hierarchical clustering of feedback responses to reward and no-reward outcomes (Fig. 3b). Hierarchical clustering provides an unsupervised and unbiased, heuristic framework to capture local similarities and to visualize graded relationships, even in the absence of sharply discrete groups. Four clusters were identified (see Methods for details), of which the response profiles are shown in Fig. 3c, d and Supplementary Fig. 4a–c. The clustering approach revealed gradations in signal morphology rather than sharply discrete groups (silhouette coefficient mean 0.21, range, −0.55 – +0.64 for reward clusters, and mean 0.15, range, −0.44 – +1.0 for no-reward clusters) indicating weak separation between clusters lying along a continuum (Supplementary Fig. 4d). This is consistent with the fact that the concentration measures in the analysis are from physically adjacent quadrats along continuous concentration gradients. These clusters provide a heuristic grouping that captures local similarity in response patterns.

The activity patterns of quadrats recorded from the DS were grouped based on their responses and temporal release profiles (Fig. 3e and Supplementary Fig. 4f). Reward delivery commonly caused brief decreases in ACh activity (Cluster 1; 60.1%, 1488/2475). Increases were uncommon: a subset (Cluster 2; 8.2%, 202/2475) showed an increase followed by a decrease, and another subset (Cluster 3; 8.2%, 204/2475) showed a decrease followed by a rebound burst in ACh levels. Activity patterns other than the above were few in proportion

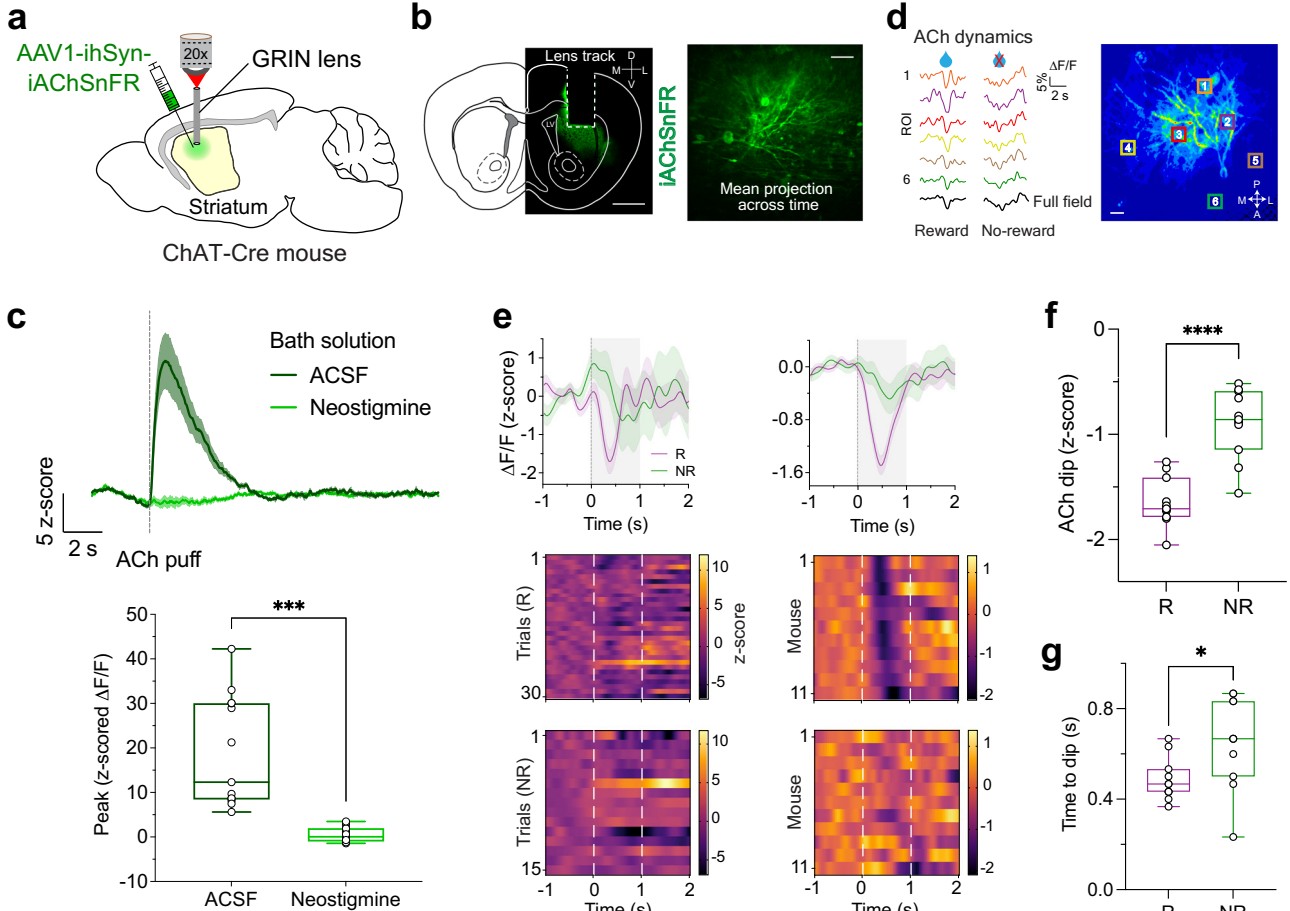

**Fig. 2 | Two-photon imaging of DS ACh transients show reward-related signals.**
**a** Schematic of experimental procedure. Sagittal section adapted from atlas image[87].
**b** Left, example histology (coronal section) showing immuno-enhanced iAChSnFR
sensor expression and the track from GRIN lens. Similar results were obtained in
$n = 10$ other brains. Slice is projected onto a corresponding atlas image[87]. D, dorsal; V,
ventral; M, medial; L, lateral; LV, lateral ventricle. Scale bar, 1 mm. Right, mean
fluorescence projection image from a representative field-of-view. Note diffuse
expression of iAChSnFR on somata and neuropil. Differences in apparent feature size
across panels reflect varying zoom levels. Scale bar, 50 μm. **c** Top, mean fluorescence
traces ($n = 13$ slices from 4 mice) showing changes in iAChSnFR responses aligned to
ACh puff in the presence of ACSF (dark green) and the acetylcholinesterase (AChE)
inhibitor neostigmine (light green). Shaded regions, S.E.M. Bath application of
neostigmine (50 μM) blunts the ACh signal, confirming that the sensor is reporting
changes in ACh fluorescence. Bottom, quantification of peak ACh signals (two-tailed,

paired $t$ test, $P = 0.0001$). **d** Mean ΔF/F traces (green) for each of 6 equal-sized
quadrats (~50 × 50 μm) on a representative pseudocolor image (right). Similar
results were obtained in 10 other mice. Note the heterogeneous ACh signals in
response to example trials (R/NR). Mean (full field) is shown in black (bottom). Water
drop denotes reward delivery. Scale bar, 50 μm. **e** Example iAChSnFR signals (z-
scored ΔF/F) aligned to trial outcomes (reward, purple; no reward, green) for a
representative mouse (left) and averaged across all 11 mice (right; reward trials = 291,
no-reward, 134). Shaded regions, S.E.M. (Bottom) Heatmaps of mean normalized
responses to R and NR outcomes. **f, g** Box plot showing the distribution of ACh
responses (z-score) during outcome period (**f**) (significant difference between con-
ditions, two-tailed, paired $t$ test, $1.65 \times 10^{-5}$), and latency of outcome-evoked ACh dips
(**g**) (two-tailed, paired $t$ test, $P = 0.03$) across all mice ($n = 11$). In box plots, center lines
depict the median, box limits represent the 25th and 75th percentiles, and whiskers,
data range. Source data are provided as a Source Data file.

(Cluster 4; 2.3%, 56/2475) (Fig. 3c). In addition, quadrats exhibiting
decreased (pause) and burst-pause response patterns were more
medially distributed ($P = 0.05$ and $P = 0.04$, respectively) and showed a
trend toward anterior localization (median position, ~0.9 mm from
bregma). In contrast, quadrats exhibiting pause-burst response pat-
terns were preferentially distributed in relatively more posterior
regions ($P = 0.005$) of the DS (Fig. 3f). Notably, while four distinct ACh
response patterns were identified during late learning, only two pri-
mary response clusters emerged during early learning (Supplementary
Fig. 2b, c, and 4g, h).

In contrast to reward responses, no-reward outcomes following
incorrect choices elicited brief decreases in activity in a moderate
subset of quadrats (Cluster 1; 32.9%, 815/2475). Overall, the propor-
tions of the observed response types indicated differential ACh
dynamics depending on trial outcome. We further analyzed the reward
and no-reward feedback responses of identified quadrats. As shown in
Fig. 3g, we found significant differences in response amplitude (z-

scored ACh dip; two-tailed, unpaired $t$ test, $P = 1.40 \times 10^{-43}$) and tem-
poral characteristics (time to dip; $P = 6.44 \times 10^{-69}$). As previously noted,
iAChSnFR is very fast, so the timing of these dips is reflective of ACh
dynamics, not a sensor artifact.

## ACh encodes bidirectional signals in response to contingency reversal

Striatal CINs have been implicated in behavioral flexibility[16–18]. To
investigate how ACh activity relates to adaptive switching behavior,
the rewarded arm of the maze was reversed after mice achieved at least
80% correct performance in a session (Fig. 4a). This reversal of con-
tingencies occurred without external cues. Following reversal, mice
initially persisted in selecting the previously rewarded arm, resulting in
a pronounced decline in correct choices and variability in their reversal
learning rates (Fig. 4b, c; average percent correct was 46.7% ± 4.6%,
mean ± S.E.M.). Across animals, the proportion of perseverative errors
was 17.1% ± 2.7% (reversal session, R1; range of 10.0%–39.4%, mean ±

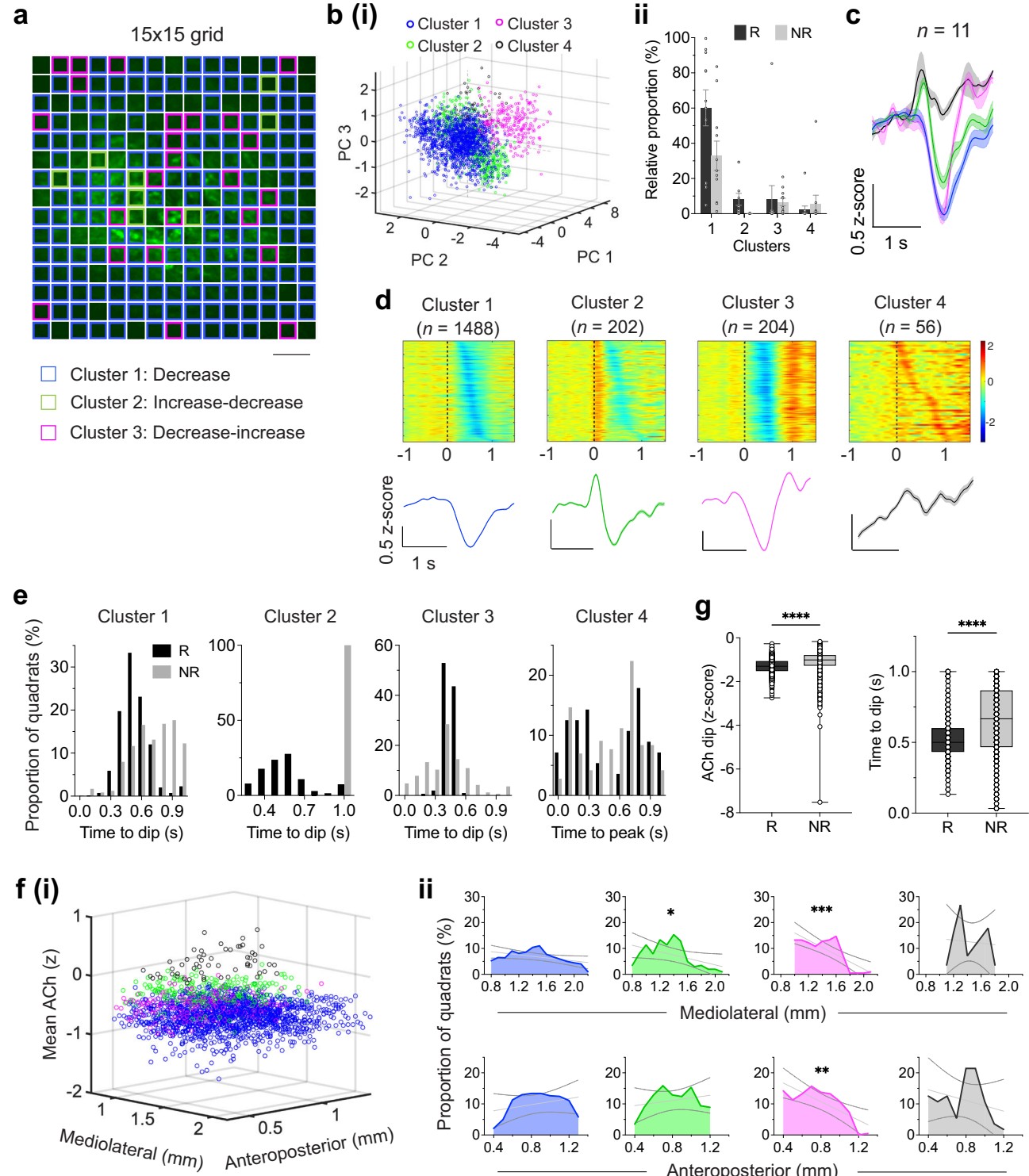

S.E.M., $n$ = 409 trials from 11 mice; Supplementary Fig. 4a). Across learning ("early" vs "late" stages), mice exhibited marked differences in measures of reward anticipation (licking rate) and goal approach behaviors (*i.e.*, run velocity) (Fig. 4d, e and Supplementary Fig. 4b–h), which ultimately resulted in gradual increases in selection of the new rewarded arm. These observations indicated that mice successfully adapted to the change in contingencies.

Simultaneous recordings of ACh activity and behavioral responses revealed distinct signaling patterns for rewarded and unrewarded outcomes after reversal. Correct responses – reflecting successful adaptation to the new contingency – were associated with a rapid,

phasic decrease in ACh activity following reward delivery. In contrast, incorrect responses were associated with a gradual increase in spatially averaged ACh levels (Fig. 4f, g and Supplementary Fig. 6a). Across animals, these responses differed significantly (Fig. 4h; two-tailed, paired $t$ test, $P$ = 1.40 × 10$^{-3}$, $n$ = 11). We propose that when a previously rewarded action unexpectedly fails to yield a reward, ACh release signals the condition of unexpected non-reward – reflecting a state change in outcome expectation. Thus, the ACh signal in response to non-reward is proposed to be specific to when a previously correct response, without warning, ceases to be rewarded.

**Fig. 3 | Spatially resolved dynamics and functional organization of DS ACh release. a** Mean fluorescence projection image for a representative mouse. Scale bar, 100 μm. Colored grids represent the spatial distribution of each response type. **b (i)**, All imaged quadrats ($n = 2475$) were sorted via a hierarchical clustering algorithm based on their trial-averaged activities, revealing four clusters in the DS with distinct response dynamics (reward clusters are shown in figure). Also see Supplementary Fig. 3a, b for NR clusters. **(ii)** Relative proportion (mean ± S.E.M.) of all R and NR clusters with significant responses (see methods). **c** Mean fluorescence traces showing changes in iAChSnFR signals to reward outcomes ($n = 11$ mice; Cluster 1: decrease, blue; Cluster 2: increase-decrease, green; Cluster 3: decrease-increase, magenta; Cluster 4: increase). Shaded regions, S.E.M. **d** Heatmaps showing mean normalized responses of all clusters with significant responses to reward. Bottom traces represent the population average within each cluster. Shaded regions, S.E.M. Individual quadrats are shown on the y-axis and sorted by the time of their minimum ACh dip (clusters 1-3), or maximum ACh activity (cluster 4).

**e** Distribution of the mean latency of ACh decreases (*i.e.*, time-to-dip; clusters 1-3) and increases (*i.e.*, time-to-peak; cluster 4) during the outcome period of R (black) and NR (gray). **f(i)**, Three-dimensional map showing the spatial localization of all clusters in (**d**) above on a standardized imaging field. Anatomical coordinates were centered using the recorded mediolateral (ML) and anteroposterior (AP) values for each GRIN lens placement (see Methods). **(ii)**, Relative proportion of all response types across the mediolateral (top) and anteroposterior (bottom) axes, simple linear regression, *$P < 0.05$, **$P < 0.01$, ***$P < 0.001$, solid line represents linear fit, dotted lines represent 95% confidence intervals. **g** Left, box plot showing the distribution of ACh dips; significant difference between conditions (two-tailed, unpaired *t* test, $P = 1.40 \times 10^{-43}$). Right, latency of outcome-evoked dips (reward, $n = 1894$ quadrats, no-reward, 981); significant difference between conditions (two-tailed, unpaired *t* test, $P = 6.44 \times 10^{-69}$). In box plots, center lines depict the median, box limits represent the 25th and 75th percentiles, and whiskers, data range. Source data are provided as a Source Data file.

## ACh transients in DS predict adaptive reversal after unexpected non-reward

We hypothesized that ACh elevations following unexpected non-reward promote adaptive shifts in behavior. To test this, we examined trial-by-trial relationships between ACh activity and subsequent choice patterns. The cholinergic signal evoked by unexpected no-reward was more pronounced on trials that preceded a behavioral switch than on those followed by a stay response (Fig. 4i). Mice with stronger ACh responses exhibited a greater tendency to switch behavior following unexpected non-reward (lose-shift), whereas those with comparably weaker ACh responses were more likely to repeat the same choice (lose-stay) (Fig. 4j, Pearson's $r = 0.67$, $P = 0.03$, $n = 11$; Supplementary Fig. 6b). To rule out potential confounds related to trial timing after reversal, we compared lose-shift and lose-stay trials within matched post-reversal trial windows (first and last five trials for each condition). ACh responses remained heightened for lose-shift trials, indicating that this effect was not solely driven by when the trials occurred (Fig. 4i, right). We further observed that certain response profiles—particularly those with delayed peaks—occurred more frequently during early reversal trials. Together, these findings indicate that both the direction and magnitude of ACh activity during non-reward predict future behavioral shifts.

Comparison of ACh responses to non-reward before and after reversal within the same animals revealed a significant increase in the spatially averaged ACh signal during reversal relative to non-reward in task acquisition phases (early learning and pre-reversal periods), (Fig. 4g, h and Supplementary Fig. 6d). This enhancement suggests that ACh signaling becomes more spatially widespread and pronounced when behavioral switching is required. Notably, modest ACh increases observed during non-reward in the acquisition phase, localized to subsets of quadrats (see below; Fig. 5b), may represent a preexisting substrate for outcome-related cholinergic modulation that is amplified and more broadly recruited under heightened demands for behavioral flexibility.

The increase in ACh activity following unexpected non-reward could, in principle, reflect motor-related orienting responses. In our setup, however, mice were head-fixed and therefore unable to perform the postural or head movements typical of orienting behaviors. Although limited orienting could be expressed through locomotor changes, analysis of locomotor velocity on a trial-by-trial basis revealed no significant correlation between mean ACh signals and velocity during the outcome period (Fig. 4k). Mice typically decelerated or stopped during this period, with near-zero velocity in the same time window when ACh levels decreased during task acquisition (Fig. 2e) but increased after reversal (Fig. 4g and Supplementary Fig. 6c). We also observed a gradual decline in anticipatory licking following unexpected non-reward (Supplementary Fig. 5f), suggesting reduced reward expectation (on entries to the previously rewarded arm) as mice adapted to the new contingency. These behavioral adjustments followed the elevated ACh activity

preceding lose-shift decisions, arguing against movement as a primary driver and supporting a role for ACh in feedback-driven behavioral updating, rather than a reactive motor signal.

It is also possible that the long-latency ACh increases reflect attentional shifts rather than learning-related feedback. However, if primarily driven by attention, such signals would be expected to occur more uniformly across task epochs, particularly during initial learning when attentional load is high (Supplementary Fig. 6d). Instead, the magnitude and timing of these responses correlated specifically with behavioral switching after unexpected non-reward, particularly during lose-shift trials (Fig. 4i). This relationship between ACh activity and adaptive switching supports a role in learning from outcome violations, as indicated by the pre-outcome anticipatory responses to the now-unrewarded arm (Supplementary Fig. 6e). Moreover, the ACh increases emerged selectively in the context of contingency reversals and were absent during earlier learning phases. Together, these findings indicate that the enhanced ACh responses to unexpected non-reward convey a teaching signal that promotes adaptive behavioral updating, rather than merely reflecting motor or attentional engagement.

## Spatially resolved representations of future switch/stay responses

The full-field ACh responses described above (Fig. 4f-i) represent spatially localized variations in ACh, *i.e.*, combining all quadrats within the imaging field into an average measure. To resolve and characterize local spatiotemporal dynamics, we next examined ACh activity at the level of individual quadrats and how these were modulated by trial outcomes after reversal. Local ACh responses during the outcome period were heterogeneous, with most quadrats exhibiting bidirectionally distinct activities for reward and non-reward outcomes (Fig. 5a and Supplementary Fig. 7a, b).

Following unexpected non-reward after reversal, ACh activity increased in the majority of quadrats (NR+; 85.3%, 2112/2475). A smaller subset showed decreases (NR-; 4.5%, 111/2475) or no significant change (non-selective, 10.2%, 252/2475). In contrast, rewarded outcomes primarily evoked decreases in ACh (R-; 80.6%, 1996/2475), whereas only a fraction showed increases (R+; 2.7%, 66/2475) or no response (16.7%, 413/2475) (Fig. 5a). Overall, ACh increases were predominantly associated with post-reversal unrewarded outcomes (Fig. 5b, c), reinforcing the conclusion striatal ACh release encodes negative outcomes during contingency changes.

Spatial mapping of quadrat responses along the mediolateral and anteroposterior axes revealed distinct topographical organization during unexpected non-reward. Quadrats with ACh increases were significantly more medially distributed (simple linear regression, $r = -0.55$, $P = 0.03$; median position, 1.4 mm from midline) and showed a trend toward anterior portions (median position, 0.9 mm from bregma). Conversely, quadrats exhibiting decreases were relatively

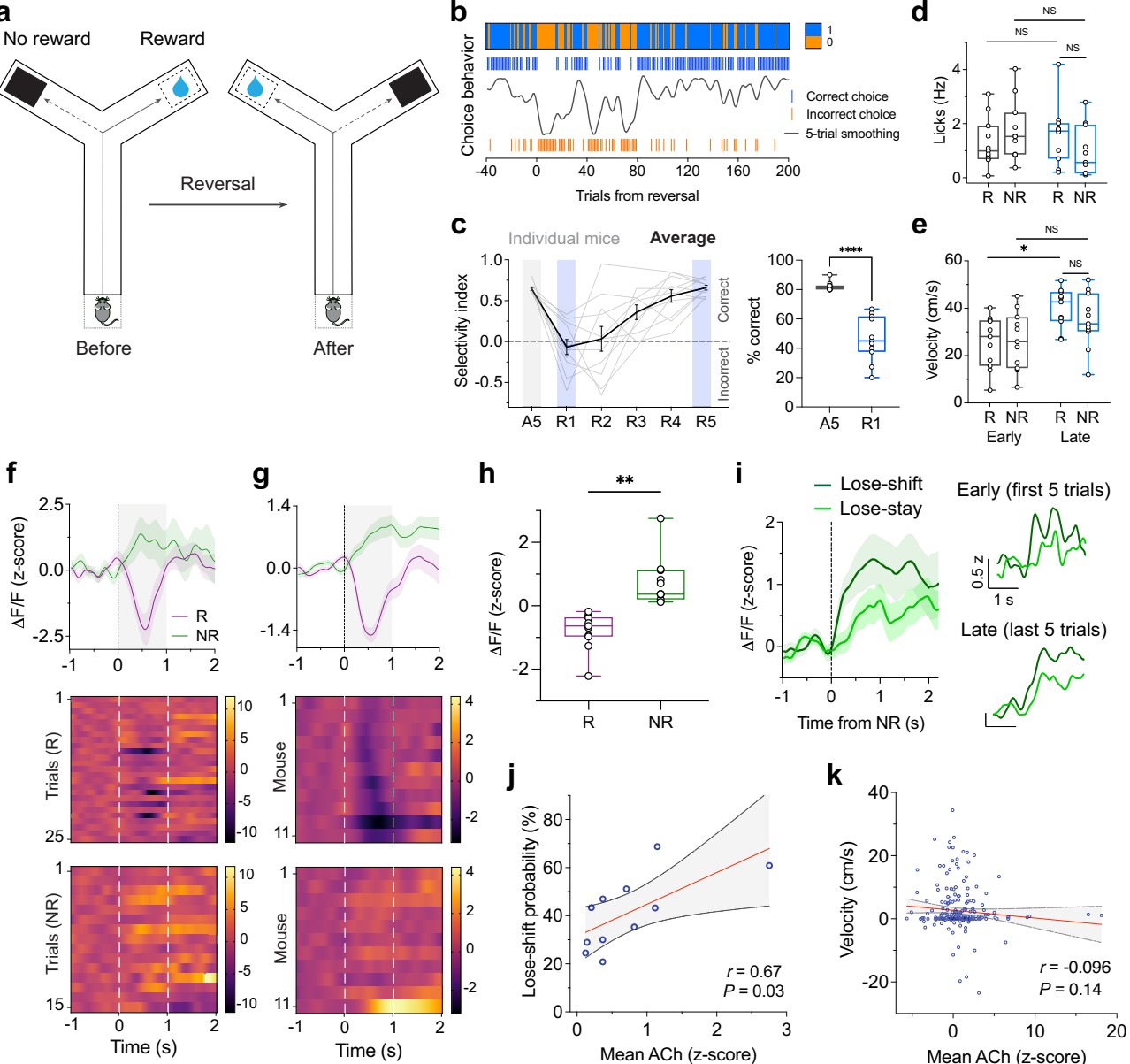

**Fig. 4 | ACh increases following reversal predict lose-shift behavior. a** Schematic of the VR-based reversal learning paradigm. **b** Example choice behavior from a representative mouse during the last 40 pre-reversal trials and across reversal. Blue and orange ticks indicate correct and incorrect choices, respectively; black trace, five-trial moving average of choice performance. **c** Left, Maze arm selectivity indices across reversal sessions (R1-R5) for all animals ($n = 11$). Gray lines, individual means; black, group mean ± S.E.M. Right, choice performance before and after reversal (two-tailed, paired $t$ test, $P = 3.93 \times 10^{-5}$). **d** Anticipatory licking in the pre-outcome zone (280–320 cm) during early and late reversal phases: Repeated measures ANOVA: main effect of session ($F_{(1,20)} = 0.57$, $P = 0.46$); outcome ($F_{(1,20)} = 0.14$, $P = 0.71$); interaction ($F_{(1,20)} = 2.96$, $P = 0.10$). Mice showed a trend toward higher anticipatory licking when approaching the new rewarded arm. **e** Same as (**d**) but for approach velocity: main effect of session ($F_{(1,20)} = 9.75$, $P = 0.0054$); outcome ($F_{(1,20)} = 0.60$, $P = 0.45$); interaction ($F_{(1,20)} = 0.58$, $P = 0.45$). **f, g** Example iAChSnFR signals aligned to trial outcomes during reversal

for a representative mouse (**f**), and across mice ($n = 11$) (**g**). Bottom, heatmaps show mean normalized responses (reward, $n = 212$ trials, no-reward, 235). **h** Mean ACh activity (z-scored ΔF/F) during the outcome period (reward vs no reward). Significant difference between conditions, two-tailed, paired $t$ test, $P = 1.40 \times 10^{-3}$. **i** Left, mean iAChSnFR transients preceding lose-shift (dark green) and lose-stay (light green) responses. Right, trials matched across early (top) and late (bottom) post-reversal windows. **j** Mean ACh signals were positively correlated with lose-shift probability following unrewarded outcomes (simple linear regression, Pearson's $r = 0.67$, $P = 0.03$, $n = 11$). **k** ACh signals did not correlate with velocity during unrewarded outcomes (0-2 s window), (Pearson's $r = -0.096$, $P = 0.14$, $n = 235$ trials). In (**j**, **k**), solid lines denote linear fits, dotted lines, 95% CIs. Shaded areas, S.E.M. In box plots, center lines depict the median, box limits represent the 25th and 75th percentiles, and whiskers, data range. Source data are provided as a Source Data file.

more laterally distributed ($r = 0.58$, $P = 0.046$) within the dorsal striatum (Fig. 5d).

To assess how individual quadrats encoded both outcomes, we further classified them by their responsiveness to reward and non-reward events. This analysis revealed marked spatial heterogeneity (Supplementary Fig. 7c). A large proportion of quadrats displayed

opposing responses to the two outcomes (R⁻ x NR⁺; 69.7%), while smaller fractions were selectively responsive to reward (R⁻ only; 7.5%) or to no-reward (NR⁺ only; 13.0%). These findings demonstrate that striatal ACh activity encodes multiple reinforcement contingencies within spatially distinct microdomains, consistent with prior observations that cholinergic signaling is strongly context-dependent[44–46].

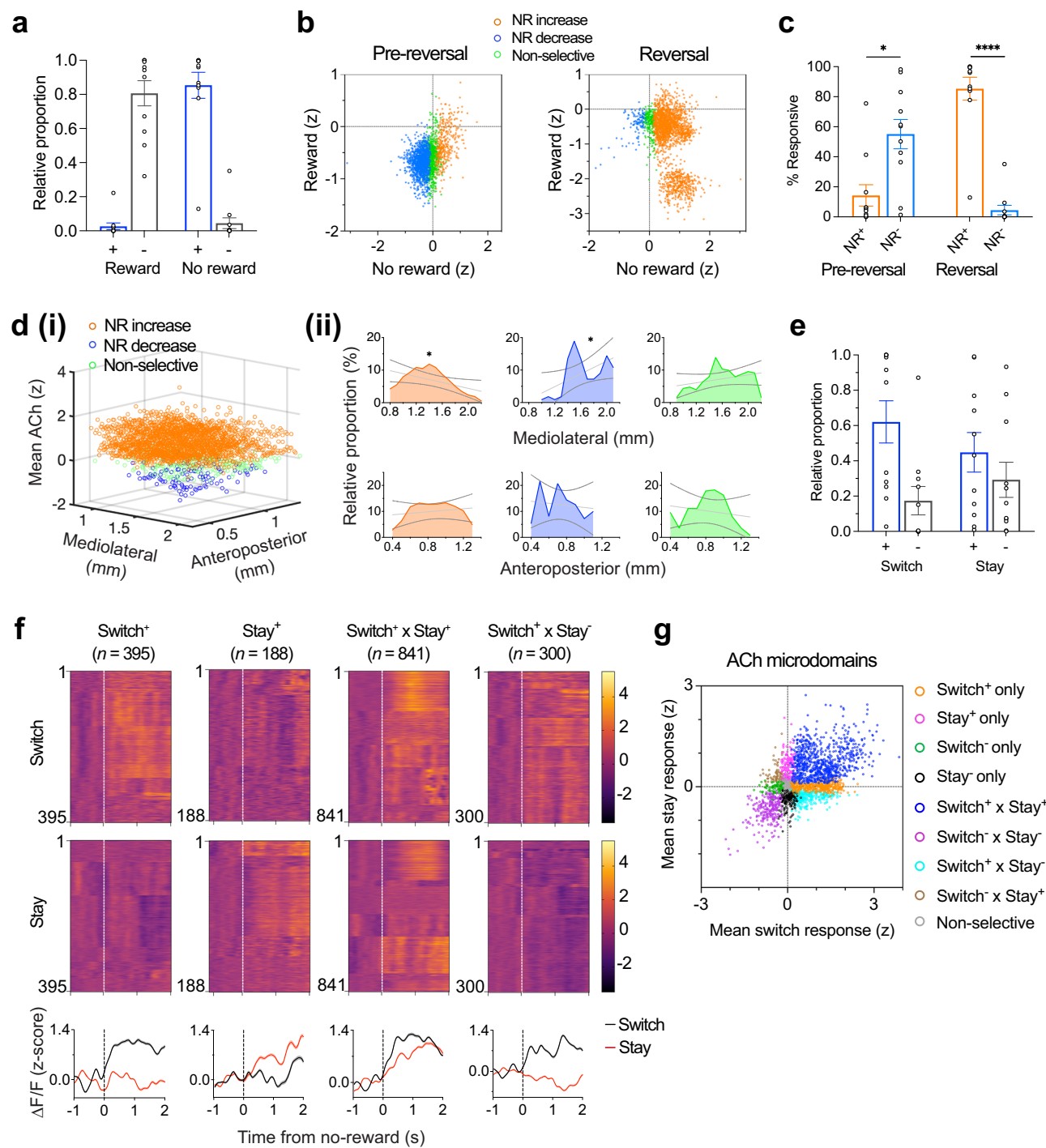

Given our finding that ACh fluctuations predicted behavioral switching after reversal (Fig. 4i, j), we next examined whether similar patterns were evident at the quadrat level. Indeed, microscale analysis revealed comparable effects: a substantial fraction of quadrats significantly encoded future switch responses following non-reward (Fig. 5e), in line with our observation of future switch encoding after contingency change (for definition of switch and stay encoding quadrats, see Methods). Most quadrats exhibited greater increases in activity preceding future switching (switch+, 62.1%) than staying (stay+, 44.8%). A notable subset was active preceding both switch and stay trials (34.0%; switch+ x stay+), while smaller populations were selectively active preceding switching (16.0%; switch+ only) or staying (7.6%; stay+ only) (Supplementary Fig. 5d). The responses of these functional subtypes are shown in Fig. 5f, where certain quadrats differentially

encoded future choice direction, whereas others showed comparable activity regardless of the anticipated choice (Fig. 5f and Supplementary Fig. 7e).

To further characterize the organization of these functionally distinct populations, we computed the mean activity of individual quadrats and projected them onto a two-dimensional space according to the magnitude and sign of their outcome responses. This analysis revealed distinct populations of quadrats selectively tuned to future behavioral responses following non-rewarded outcomes (Fig. 5g and Supplementary Fig. 8a–c), underscoring the multifaceted and spatially heterogeneous nature of striatal cholinergic signaling[47]. Collectively, these results indicate that, at the microscale level, ACh activity encodes information predictive of both future switching and staying behaviors. Such heterogeneous response patterns may

**Fig. 5 | Spatiotemporal organization of DS ACh release following reversal.**
**a** Proportion of quadrats per mouse showing outcome-selective ACh responses during reversal. Many quadrats exhibited decreased(-) ACh responses to reward and increased(+) responses to no-reward, respectively. Quadrats were classified as responsive (positive, negative, or non-selective) if the outcome-evoked responses significantly differed from baseline activity (two-tailed, paired *t* test, $P < 0.05$; See "Methods"). The proportions of quadrats showing increases vs decreases differed significantly between R and NR outcomes ($X^2$ test: $X^2(1) = 436$, $P = 8.05 \times 10^{-97}$). **b** Spatiotemporal representation of mean ACh activity before (pre-reversal) and during reversal. Each circle denotes a quadrat's z-scored ΔF/F projected onto a two-dimensional response space. Color code: orange, increased activity to NR; blue, decreased activity to NR; green, non-selective. Dotted lines mark the boundaries between positive and negative response axes. In the pre-reversal phase, most quadrats decreased ACh activity to NR (decrease, $n = 1364$ quadrats, increase, 354). Following reversal, most quadrats increased activity to NR (decrease, $n = 111$ quadrats, increase, 2112). **c** Distribution of responses in (**b**). Significant difference

between conditions (two-tailed, paired *t* test: pre-reversal, $P = 2.98 \times 10^{-2}$; reversal, $P = 1.96 \times 10^{-5}$). **d (i)**, Three-dimensional spatial map illustrating the distribution of functionally responsive quadrats along the mediolateral and anteroposterior axes in response to unexpected NR. **(ii)**, Relative proportions of response types along the mediolateral (top) and anteroposterior (bottom) axes. Simple linear regression, *$P < 0.05$, solid lines denote linear fits, dotted lines, 95% confidence intervals. **e** Proportions of quadrats encoding future behavioral responses following NR (switch⁺, increased activity preceding future switching; switch⁻, decreased activity). Quadrats showing increases and decreases in ACh differed significantly in proportion between switch and stay responding: ($X^2$ test: $X^2(1) = 75.8$, $P = 3.1 \times 10^{-18}$). **f** Heatmaps showing normalized activity of task-responsive quadrats aligned to future switching (top) or staying (bottom). Each row denotes a single quadrat; traces below show population-averaged activity (mean ± S.E.M). Also see Supplementary Fig. 7e. **g** Scatter plot showing all response types (clusters); each circle represents a quadrat's response, color-coded by cluster identity. In (**a**, **c**, **e**), data are mean ± S.E.M. ($n = 11$). Source data are provided as a Source Data file.

represent a substrate through which ACh modulates adaptive decision-making.

To correct for spherical aberrations inherent in GRIN lens imaging, which results in a lower baseline fluorescence signal (Fo) at the edges of the field of view compared to the center, we used ΔF/Fo normalization, which scales the signal relative to Fo. To ensure that this normalization effectively accounts for baseline intensity differences and that our main findings are not simply driven by expression gradients or brightness alone, we assessed whether ΔF/F values were dependent on Fo. We found no significant correlation between the mean of the behavior-linked signals and baseline fluorescence across quadrats (Supplementary Fig. 8d, e). Thus, regions with lower overall brightness still exhibited robust ΔF/F responses. We also verified that quadrats in peripheral and central areas show similar patterns of behavioral modulation (Supplementary Fig. 8f, g). These analyses indicate that our results are not biased by location within the field of view.

### Chemogenetic inhibition of ACh-releasing neurons in DS disrupts lose-shift behavior

To test whether the increased acetylcholine we observed after unexpected non-reward caused the associated lose-shift behavior, we inhibited CIN activity using the designer receptor exclusively activated by designer drug (DREADD) approach[48]. Injection of AAV expressing Cre-dependent hM4D(Gi)-mCherry bilaterally into the DS of *ChAT*-Cre mice (Fig. 6a) produced hM4D(Gi)-mCherry expression in cell bodies of CINs (Fig. 6b). After mice reached an acquisition criterion of 80% correct performance, each mouse received an injection of either JHU37160 dihydrochloride[49] (J60; a high-affinity DREADD agonist, $n = 5$ mice) or vehicle (buffered saline, $n = 5$ mice), and then commenced reversal learning (Fig. 6c).

CIN inhibition caused a significant decrease in the percentage of correct choices during early reversal sessions, with no effect observed in late sessions (Fig. 6d; $P = 0.028$, Cohen's $d = (-)1.54$, 95% CI [−2.98, −0.09], early and $P = 0.11$, Cohen's $d = (-)1.01$, 95% CI [−2.35, 0.32], late, two-tailed, unpaired *t* test). Following the reversal, saline-treated mice displayed an accelerated increase in correct choice performance (Fig. 6e; $F_{(1,8)} = 7.225$, $P = 0.028$). Cumulative errors over the first 80 trials post-reversal were higher in CIN-inhibited mice, reflecting impaired overall task adaptation beyond specific error types (Fig. 6f).

Lose-shift behavior was significantly reduced by CIN inhibition (Fig. 6g, left; two-tailed, unpaired *t* test, $P = 0.015$, Cohen's $d = (-)1.77$, 95% CI [−3.27, −0.26]). However, CIN inhibition effects on win-stay probability following the reversal were not significant (Fig. 6g, right; $P = 0.14$, Cohen's $d = (-)0.94$, 95% CI [−2.27, 0.38]). The impairment in lose-shift behavior led to an increased number of trials required to reach the performance criterion (63 ± 14 and 115 ± 11 trials, mean ± S.E.M.; saline and J60-treated mice, respectively, $P = 0.019$, Cohen's $d = 1.67$, 95% CI [0.19, 3.15]). While the difference in perseverative

errors was not significant ($P = 0.47$, Cohen's $d = 0.43$, 95% CI [−0.82, 1.69]), a significant difference in the proportion of regressive errors was observed between the two groups ($P = 0.007$, Cohen's $d = 2.04$, 95% CI [0.45, 3.62]) (Supplementary Fig. 9a–c).

Previous studies have proposed a role for striatal CINs in locomotion[50–52]. Therefore, we sought to determine whether the observed differences were attributable to locomotor impairments. Our findings revealed that CIN inhibition did not affect running velocity ($P = 0.72$, Cohen's $d = 0.21$, 95% CI [−1.03, 1.45], Supplementary Fig. 9d). Although our inhibition protocol lacked temporal specificity, the behavioral findings are consistent with the observed changes in ACh associated with lose-shift behavior. Combined, these results demonstrate that dorsal striatal ACh is essential for timely behavioral adaptation to changing reinforcement contingencies and suggest a role for ACh in accelerating the rate of learning during new learning processes.

## Discussion

We identified spatiotemporally distinct patterns of ACh release in the dorsal striatum and established its causal contribution to behavioral flexibility. Distinct ACh dynamics emerged before and after an uncued reversal of reward contingencies in a virtual Y-maze. Following reversal, choices of the previously rewarded arm – and thus not receiving the expected reward – elicited a delayed, progressive increase in ACh activity. This newly observed ACh elevation in response to unexpected non-reward was tightly associated with lose-shift behavior, suggesting that striatal ACh facilitates adaptive switching when established contingencies change. Supporting this view, chemogenetic inhibition of CINs in the dorsal striatum impaired lose-shift responses. By segmenting the imaging field into spatial quadrats, we uncovered non-uniform spatiotemporal dynamics of ACh release. Individual quadrats displayed heterogeneous response profiles to rewards, non-rewards after errors, and unexpected non-rewards, suggesting the presence of functionally distinct microdomains. These spatially distinct populations may be differentially driven by context-related thalamostriatal or corticostriatal synaptic inputs. Collectively, our findings identify a causal role for striatal ACh in guiding lose-shift behavior after unexpected non-reward and point to the possibility of regional specificity in cholinergic signaling that supports behavioral flexibility.

Our observation of elevated striatal ACh in association with unexpected non-reward following uncued reversal is significant. Previous studies have shown increased ACh efflux in medial[53] and dorsomedial striatum (DMS)[23] during place reversal learning. Increased ACh efflux has also been reported during switching between place and response strategies[54]. A sustained decrease in the ACh precursor, choline, indicating increased ACh activity, has also been reported in humans during reversal learning[55]. However, the measurements of ACh in these studies did not have sufficient temporal resolution to

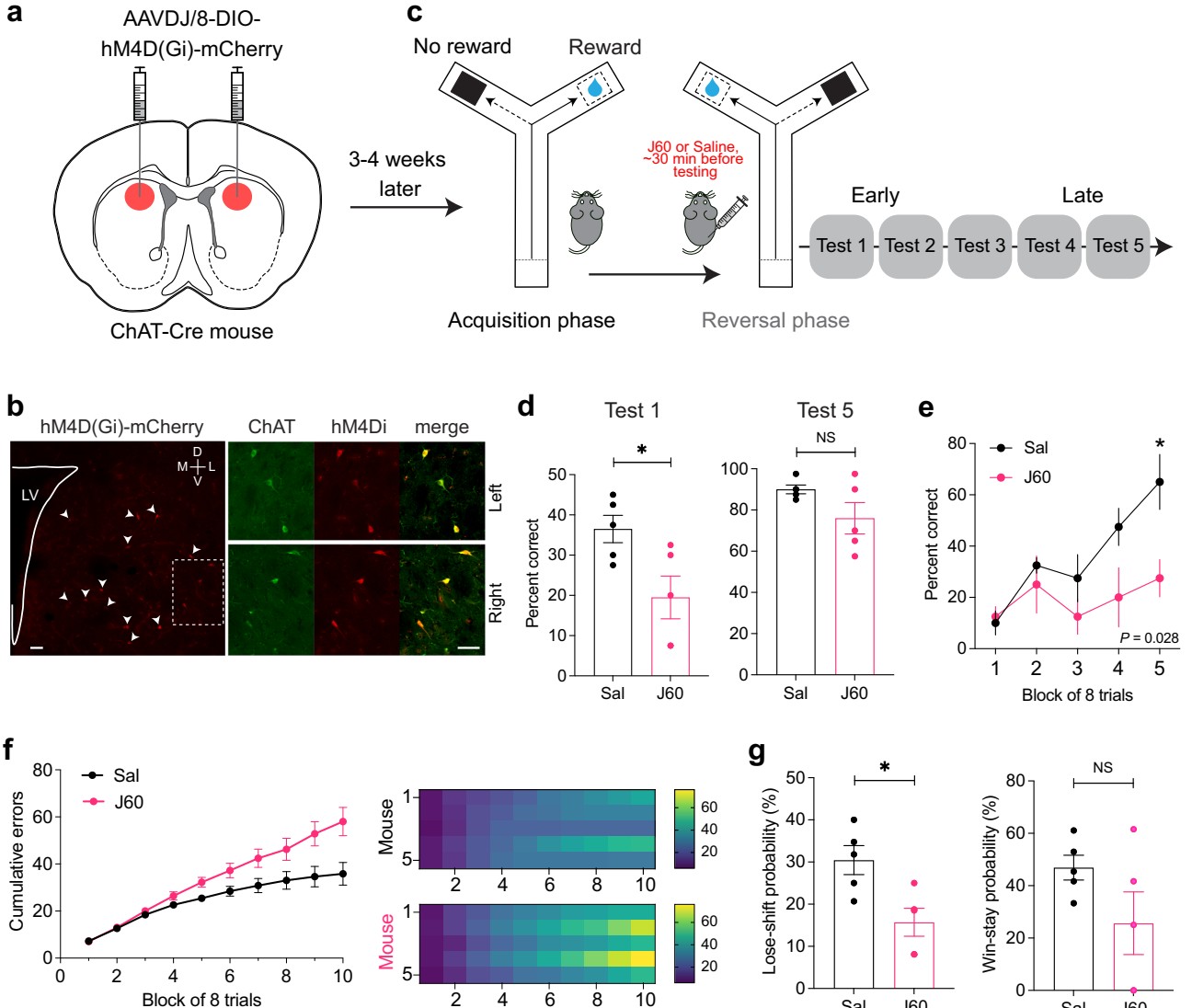

**Fig. 6 | Inhibition of striatal CINs impairs lose-shift behavior. a** Schematic of viral strategy showing stereotaxic injection location on atlas image[87]. **b** Left, Post hoc immunohistochemistry and confocal microscopy confirmed co-localization between Gi-DREADD and *ChAT* immunohistochemistry. Similar results were obtained in 9 other brains. Labeled cells are marked with arrow heads. D, dorsal; V, ventral; M medial; L, lateral; LV, lateral ventricle. Inset is shown on the right (bottom). Scale bars, 50 μm. **c** Timeline of the reversal learning procedure with Gi-DREADD inhibition: after animal recovery and viral expression, mice were initially trained in task acquisition, following which the reversal phase began (Tests 1–5). **d** Performance (percent correct, mean ± S.E.M.) of J60 and saline- treated animals in the reversal learning phase. Mean performance was significantly increased in the saline group compared with the J60 group in Test 1 (two-tailed, unpaired *t* test, *P* = 0.028, Cohen's *d* = (−)1.54, 95% CI [− 2.98, − 0.09], *n* = 5 mice/group) but comparable in Test 5 (*P* = 0.11, Cohen's *d* = (−)1.01, 95% CI [− 2.35, 0.32]). **e** Choice

performance (mean ± S.E.M.) in Test 1 broken down by block of trials (8 trials/ block). Two-way repeated measures ANOVA, main effect of group ($F_{(1,8)}$ = 7.225, *P* = 0.028) and block ($F_{(4,32)}$ = 6.193, *P* = 0.0008), but no interaction ($F_{(4,32)}$ = 2.188, *P* = 0.093), post hoc Tukey test). **f** Left, inhibition of CINs impaired the learning rate of the J60- compared with the saline-treated group. Learning curve showing trial-averaged cumulative errors broken down by block of trials (mean ± S.E.M.; 8 trials/ block, first 80 trials across *n* = 10 mice). Note the rate of increase in the number of errors after the 3^rd block in J60- relative to saline- treated mice. Right, Block-by-block heatmap of the cumulative errors for individual mice in each group. **g** Probability (percent, mean ± S.E.M.) of lose-shift (two-tailed, unpaired *t* test, *P* = 0.015, Cohen's *d* = (−)1.77, 95% CI [− 3.27, − 0.26], *n* = 5 mice/group) and win-stay (two-tailed, unpaired *t* test, *P* = 0.14, Cohen's *d* = (−)0.94, 95% CI [− 2.27, 0.38]) behaviors in the reversal phase. In (**d**, **g**), shaded circles represent individual mice, bars represent averages across mice. Source data are provided as a Source Data file.

distinguish between rewarded and unrewarded trials. Using fiber photometry, extinction has been associated with increased ACh release in the anterior dorsal striatum[24,56] and DMS[57], consistent with the present study. However, unlike Pavlovian conditioning paradigms, our task incorporates two active choice outcomes, allowing direct assessment of behavioral flexibility in action selection. The increase in ACh is consistent with evidence that striatal CINs are sensitive to behavioral context[44] and encode deviations in reinforcement contingencies[39,46,58]. In primate studies, tonically active neurons (TANs), presumed CINs, exhibit enhanced firing during tasks requiring

suppression of a previously learned response[46], and a subset of TANs show increased activity following reward omission in probabilistic learning paradigms[39]. These findings, together with our results, support the view that striatal ACh signals reward omission during states of contingency change and contributes to encoding unexpected outcomes[59,60]. An alternative interpretation is that the ACh signal reflects the sensory salience of the non-reward cue. However, this response was specific to the reversal phase and absent during initial acquisition, despite identical cue presentation. The delayed onset of the ACh increase, occurring after anticipatory licking, further suggests

that it represents a violation of reward expectation rather than a cue-evoked sensory response. Moreover, electrophysiological studies report a transient pause and rebound activation in CINs following sensory stimuli, whose timing does not align with the long-latency ACh increases observed here[30–32,61]. Thus, the signal likely reflects an internal (cognitive) state of expectant waiting, disrupted by its violation, which is more characteristic of a prediction error signal associated with unexpected non-reward.

The observation that ACh activity predicted lose-shift behavior further implies a causal role in behavioral adaptation *i.e.*, the increase not only signals the mismatch implicit in unexpected non-reward but also contributes to the behavioral change. Consistently, chemogenetic inhibition of CINs reduced lose-shift responses, paralleling prior reports that CIN lesions or muscarinic blockade in the DMS impair set shifting[18,62] and reversal learning[21,22,63]. During initial acquisition, ACh responses were primarily reward-related. When no reward was delivered due to an incorrect choice, there is a much diminished response in the same direction. This suggests that when an animal makes an incorrect response in a constant environment (*i.e.*, stable contingency), ACh signaling is not engaged and behavioral alterations are driven by other neuromodulators. Thus, we suggest that CINs are not essential for all learning but play a specific role in behavioral flexibility when environmental contingencies change[16,18]. We speculate that the ACh increases observed following unexpected non-reward reflect an early component of behavioral adaptation, serving as a learning signal in which the CINs are signaling that the expected outcome did not occur. However, distinguishing whether ACh changes reflect learning per se or more immediate affective or motivational reactions (such as frustration) that coincide with or are influenced by changes in reward prediction remains critical. These mechanisms cannot be separated with the current dataset, and further experiments will be required to investigate these aspects.

The distinct response clusters we observed during task acquisition—particularly decrease, increase, and biphasic patterns—were not only defined by statistical thresholds but also displayed non-random spatial organization within the DS. This supports the existence of spatial heterogeneity in cholinergic signaling, consistent with recent anatomical and functional studies demonstrating topographical variation in cholinergic interneuron activity and receptor distribution[47,62,64]. Structurally, the cell bodies of CINs in the dorsal striatum are not clustered but have a random spatial distribution[65–67]. The axon of individual CINs arborises profusely to fill a spheroidal volume with a long axis of 500–700 μm[68–71]. Therefore, the axonal arborization of each CIN would extend over multiple 50×50 μm quadrats. This might explain why the clustering results indicate gradations rather than sharply discrete groups. On the other hand, the clusters are not necessarily due to anatomical structures but may be dynamically generated. Macroscopic features such as traveling and stationary waves can arise from the dynamical interactions of multiple CIN axon arborizations overlapping in the same volumes[72]. Finally, given that interactions among CIN axons can occur over distances greater than 50 μm[33] and corticostriatal afferents typically project to multiple striatal locations[73,74], it is not surprising that quadrats from the same cluster are not always physically adjacent. These findings suggest that different microdomains within the striatum may differentially process action outcomes, potentially contributing to the flexible updating of behavior. The spatially heterogeneous nature of cholinergic signaling was also evident following the reversal of contingencies. While most quadrats exhibited ACh increases in response to unexpected non-reward, a subset displayed distinct response patterns. Our analyses revealed that quadrats showing increased ACh responses to non-reward outcomes were predominantly located in more medial and anterior regions of the imaging field. This spatial pattern aligns with previous reports implicating the medial striatum in feedback processing and behavioral adaptation[23,60,75], and with the established role of the DMS in behavioral flexibility and outcome monitoring. Although regional differences were not the primary focus of our study, these findings suggest that the enhanced ACh responses may arise, at least in part, from functional subpopulations within the DS. Future studies employing more targeted imaging or manipulations could further delineate these regional contributions.

Flexible switching from one strategy to another requires suppression of previously acquired responses and acquisition of the new ones. Mechanistically, this may involve spatially selective actions of ACh on specific striatal circuits – potentiating certain synaptic connections and leaving others unaltered. Such spatially heterogeneous ACh distributions may help protect previously acquired connectivity while facilitating new associations in selected regions. Therefore, the spatial organization of ACh signaling may contribute to behavioral flexibility by combining local fluctuations with a permissive modulatory signal that governs long-term changes in the efficacy of corticostriatal synapses[76].

Recent work shows that long-term potentiation of cortical inputs to striatal projection neurons requires coincidence of pauses in CINs with phasic dopamine activation[77], which occurs in response to unexpected reward. Thus, we propose that neural pathways exposed to decreased ACh on unexpected reward delivery – causing a CIN pause and dopamine activation after successfully shifting – would experience synaptic potentiation. At the end of behavioral acquisition, it is likely that reward will be expected, and dopamine will not be released, so there will be no potentiation. Pathways exposed to increased ACh on unexpected non-reward would not be potentiated, preventing reinforcement of incorrect responses. This would ultimately lead to a reduction in interference from pre-existing memory[16]. Our observations of spatially heterogeneous responses suggest that both processes may occur in parallel in different quadrats. Thus, the spatiotemporal diversity of striatal ACh signaling underlies its role in mediating diverse adaptive responses. Further, the spatially heterogeneous ACh responses we observed—occurring without clear propagation across the imaging field—may resemble a "standing wave" profile[72] in which acetylcholine locally activates dopamine fibers without inducing a traveling wave of activation. This suggests that local ACh dynamics can generate spatially confined patterns of neuromodulator release, potentially contributing to functional compartmentalization within the striatum.

Two-photon imaging imposes practical limitations on session duration due to photobleaching risk, laser heating, and imaging stability – especially in awake, behaving animals[78,79]. Unlike brief Pavlovian trials, our VR task produces longer, continuous ACh recordings during decision-making. We therefore used fewer, longer trials, consistent with maze reversal studies that often use just two sessions: acquisition and reversal[8,19,21]. Our task extended this by using multiple acquisition sessions, with reversal triggered only after mice reached 80% accuracy. This ensured learning was complete before reversal, thereby providing a clear contrast at the moment of contingency change. Although this design led to fewer trials per session, it enhanced the interpretability of ACh responses specifically tied to well-learned behavior. We used a single reversal to elicit responses to an unexpected change in reinforcement contingencies under well-learned conditions. We elected not to use repeated reversals, which diminish error-driven responses due to growing expectation of change[80,81], potentially reducing the salience of non-reward. While this limits trial count, it strengthens the relevance of the observed ACh dynamics to unanticipated outcome shifts requiring behavioral flexibility. In general, a lower number of trials (40 per session across five sessions) does not impair learning if there is a sufficient interval between rewards[82] (average of ~50 s in our task).

Finally, our findings prompt the question of what upstream inputs drive the increased striatal cholinergic activity. CINs receive dense inputs from the parafascicular (Pf) thalamic nucleus[83,84], a major source

of excitatory input to the striatum[85,86]. Lesions of the Pf reduce CIN firing and intrinsic activity, and loss of thalamostriatal afferents impairs learning when behavioral adjustment to reversed contingencies is required, a deficit attributed to reduced cholinergic activity[16]. These observations underscore the functional significance of Pf input as a driver of striatal cholinergic activity. Future studies combining recording and manipulation of Pf activity during tasks that demand flexible responding could directly test this hypothesis. Nonetheless, our account of the distinct spatiotemporal patterns of dorsal striatal ACh release and its association with reversal learning provides a valuable framework for future experimental and theoretical studies.

## Methods

### Animals

All experimental protocols were performed in an AAALAC (Association for Assessment and Accreditation of Laboratory Animal Care) certified facility, reviewed and approved by the Animal Care and Use Committee (ACUC; protocol numbers ACUP-2021-018 and ACUP-2021-011-2) of Okinawa Institute of Science and Technology Graduate University (OIST). Male and female *ChAT*-IRES-Cre::SV40pA::Δneo (*Chat*$^{tm1(cre)Lowl}$; Stock No. 031661; Jackson Laboratories, Bar Harbor, ME, USA) mice (20–35 g, postnatal 3–4 months at the start of surgery) were used in all experiments. Until the start of surgical procedures, all mice were group-housed (3–5 mice/cage) and provided with food and water *ad libitum* under standard laboratory conditions (22–24 °C, 40–60% humidity). To prevent damage to GRIN lens implants, experimental mice were housed individually after surgical implantations. All behavioral experiments were performed during the dark phase of a reverse 12 hr light/dark cycle, with darkness from 7:00 am to 7:00 pm. A week before the start of the virtual-reality experiments, mice were water-restricted and maintained at 85–90% of *ad libitum* weight. All mice were weighed and handled daily to ensure weight maintenance. For all behavioral sessions, the mice were transferred to the behavior testing room at least 30 min before the tests to allow them to acclimatize to the environment. The mice were given free access to water at least 1 h after the end of behavioral training. All experiments in this study were carried out using both male ($n = 17$) and female ($n = 8$) mice, but statistical analyses for sex differences were not performed because of limited sample sizes. Sex was not considered as a biological variable.

### Stereotaxic surgery and viral vector injections

All surgical procedures were conducted under general anesthesia using isoflurane (3–4% for induction, 1–1.5% for maintenance; flow rate, 1.5 L/min). For acetylcholine imaging in the dorsal striatum, surgeries were conducted on adult mice under aseptic conditions. First, the mice were weighed and then placed in a plexiglas chamber filled with gaseous isoflurane to induce anesthesia. Anesthetized mice were immobilized in a stereotaxic frame (Kopf Instruments) with their heads secured by non-rupture ear bars and their noses placed into a specialized nose cone for continuous inhalation of isoflurane. The body temperature was monitored and maintained at 36 °C with a heating pad. The scalp was shaved, and a local anesthetic, lidocaine, was applied, after which a small incision was performed along the midline using a sterile scalpel. Then, the skull was exposed and the periosteum removed using a delicate bone scraper to assure clear viewing of the skull sutures. Afterwards, the bregma and lambda were then leveled to within 0.1 mm. A 0.8–1.2 mm diameter craniotomy was then drilled (centered around target coordinates for virus injection and GRIN lens implantation) on the skull above the dorsal striatum using a motorized drill. To achieve non-specific viral expression, an adeno-associated virus (AAV) encoding iAChSnFR under control of the human *synapsin-1* promoter with Tet-Off amplification[28], AAV1-*ihSyn*-tTa-sv40/TRE-iAChSnFR-minWPRE (titer 3 ×10$^{12}$ genome copies/mL, 200 nL volume) was injected unilaterally into the dorsal striatum, in combination with

the static red fluorescent label AAV1-*CAG*-DIO-tdTomato (Addgene #104112; titer 4 ×10$^{13}$ gc/mL, 100 nL volume) as control. Stereotactic coordinates for the DS were determined using the mouse brain atlas[87]. Two injections were made at stereotactic coordinates (relative to the skull surface at bregma): +1.0 anteroposterior (AP), ±1.6 and 1.9 mediolateral (ML), and –2.9 dorsoventral (DV). To minimize backflow and to allow spread of the virus, the solution was slowly injected, and the pipette was left in place for 10–15 min after viral infusion before being slowly withdrawn. Viral infusions were carried out at a rate of 1 nL/sec using a nanoliter injector (Nanoject III, Drummond) through a pulled glass micropipette filled with mineral oil (MP Biomedicals). Following viral injection, a metal head-plate (CF-10, Narishige) was secured to the skull (using a 3D-printed custom tool) with adhesive cement (Super-Bond C&B, Sun Medical) and a self-curing dental acrylic resin (Unifast II, GC Corporation). Mice were given a carprofen injection (for post-operative treatment, 5 mg/kg, s.c.) following surgery, and daily carprofen tablets for three subsequent days. Animals were returned to their home cages and allowed to recover for a minimum of 7 days before lens implantation.

### GRIN lens implantation

All surgical procedures were performed under aseptic conditions as above. About 1 week after viral injection, mice underwent a second stereotaxic survival surgery to implant a 1 mm diameter gradient index (GRIN) lens (2 P doublet probes; NEM-100-25-10-860-DS, GRINTECH) to the DS. Before implantation, dexamethasone (dissolved in 0.9% saline, 2 mg/kg, s.c.) was administered to reduce brain edema. To facilitate GRIN lens insertion, a 25-gauge needle was attached to the micromanipulator of the stereotaxic frame and inserted from the brain surface to a depth of about 200 μm above the GRIN lens target position. Then, the needle was slowly withdrawn, and the lens was lowered (centered at + 0.9 AP and ±1.8 ML) to approximately 200 μm directly above the injection site using a lens holder (Inscopix), at a slow and constant rate of ~200 μm/min. A thin layer of Kwik Sil (WPI) was applied to fix the lens to the craniotomy site. The lens was then affixed to the skull using a thick layer of dental acrylic resin, extending up the sides of the GRIN lens to build a headcap (coated with black nail polish), after which the lens holder grip was loosened. Silicone sealant was applied over the lens to prevent any damage. Analgesia (carprofen, 5 mg/kg, s.c.) was given for post-operative treatment, and daily carprofen tablets was administered for three subsequent days after surgery. All animals were monitored over a recovery period of 1–2 weeks, after which they were habituated to head restraint before behavioral testing and imaging commenced.

### Fluorescence imaging of iAChSnFR in brain slices

For the slice recording experiments, we first expressed iAChSnFR in the DS of mice by injecting 200 nL of AAV1-*ihSyn*-tTa-sv40/TRE-iAChSnFR-minWPRE at the following coordinates: AP, +1.0 mm; ML, ±1.7 mm; DV, –2.9 mm from bregma. After 3-4 weeks of viral expression, the animals were deeply anesthetized with 5% isoflurane and then perfused. The brains were rapidly extracted and placed directly in cold slicing buffer containing (in mM): 2.5 KCl, 1.2 NaH$_2$PO$_4$, 30 NaHCO$_3$, 20 HEPES, 25 glucose, 2 thiourea, 5 Na-ascorbate, 3 Na-pyruvate, 5 MgCl$_2$·6H$_2$O, and 2 CaCl2·2H$_2$O. They were then sectioned into 300 μm thick coronal slices with a vibratome (VT1200, Leica Biosystems) and quickly transferred into a container water-bathed at 34 °C with same buffer solution and then Na$^+$ spike-in procedure was performed according to the protocol provided in Ting et al.'s study[88]. Thereafter, all slices were transferred to a holding chamber (BSK4, Scientific Systems Design, Inc.) with holding solution containing (in mM): 92 NaCl, 2.5 KCl, 1.2 NaH$_2$PO$_4$, 30 NaHCO$_3$, 20 HEPES, 25 glucose, 2 thiourea, 5 Na-ascorbate, 3 Na-pyruvate, 2 CaCl$_2$·4H$_2$O, 2 MgSO$_4$·6H$_2$O), which was continuously bubbled with 95/5% O$_2$/CO$_2$ for at least 15 minutes at room temperature until recordings were

performed. The brain-slice recordings were performed at 32–34 °C. Imaging was performed using a sCMOS camera (PCO.Panda 4.2, Excelitas Technologies Co.), equipped with an Olympus 60x/0.9 NA water-immersion objective, at a frame rate of 20 Hz and synchronized with the puffing via TTL triggers. ACh sensor fluorescence was excited with an LED (Solis LED, Thorlabs) at a wavelength of 470 nm and collected with a 525/50 emission filter on an upright epifluorescence microscope (BX-51 WI, Olympus). To confirm that tissue expressing iAChSnFR was responsive to ACh, a glass pipette filled with recording ACSF containing 50 µM ACh was placed above the DS, and slight positive pressure (12 psi ≈ 80 kPa) was briefly applied. The duration of each puff was set at 30 ms, and time-locked fluorescence responses were recorded. Fluorescence responses to puffs of ACh was later blocked with recording ACSF containing 50 µM of the acetylcholinesterase inhibitor, neostigmine, which was bath applied by perfusion into the recording chamber. The image processing including, downsampling (to reduce file size and processing time), quadrat selection (to generate 1-D temporal signals), background correction and peak finding, were performed using custom scripts in Python. Changes in fluorescence DF/$F_0$ was computed as $(F − F_0)/F_0$, where $F$ is the pixel-wise fluorescence value at each time point and $F_0$ is the mean of the 4.5 s baseline fluorescence prior to puffing onset. The effect of photobleaching was removed by subtracting a B-Splines fit to the periods 0.5–4.5 s (initial parts) and 14.5–19.5 s (tail part) using Python (note that puffing onset and responses in the period of 4.5–19.5 s are masked). Peak responses were calculated as the maximum z-scored $\Delta F/F_0$ response recorded for each slice.

## Virtual-reality behavioral system

We designed a virtual reality (VR) task for head-fixed mice using the Jet Ball system (PhenoSys GmbH, Germany). The Jet Ball system is a VR experimental system for rodents, which is based on an air-supported styrofoam ball allowing a restrained animal to navigate a two-dimensional linear corridor in a virtual environment. The VR-Jet Ball setup comprised of two sub-units: the Jet Ball Unit and the Control Unit. The Jet Ball unit consisted of six 19" TFT surround monitors (270°, in octagonal arrangement) running at 75 Hz, and a styrofoam ball (20 cm diameter) that was floated using a pressurized air jet, enabling movements with minimal friction. The styrofoam ball sat in an aluminum bowl (ball holder), which contained multiple outlets from which a stream of compressed air was directed. Two X/Y motion sensors attached to the bowl measured ball rotations, i.e., mouse position and running speed, such that any movement of the ball was fed into a computer that generates and updates the virtual environment. The ball holder is also equipped with a liquid reward system mounted on a retractable operant device with a peristaltic pump from which reward (10% sucrose solution) was given to the mouse. The reward system consisted of the lick sensor (Piezo sensor) integrated into a cannula, a potentiometer for adjusting the sensitivity, and an articulated arm to adjust the position of the operant device in relation to the animal. Mice were positioned on top of the ball, allowing a slightly uphill run and head-fixed for behavioral training under a two-photon microscope. The Jet Ball unit was controlled via the control unit consisting of a control monitor that was situated on a movable rack with computer, power supply unit, flow meter (a pressure regulator to adjust airflow to the ball, 10–15 L/min), etc. The Y-maze consisted of a closed loop two-dimensional linear corridor that was 350 cm long and 30 cm wide (start region, −15–0 cm; stem, 0–280 cm; pre-outcome, 280–320 cm; outcome location, 320- cm). The walls of the maze (20 cm high) were textured with gray/black stripe patterns (start region up to the junction) and black dots (from junction to the reward location). The behavioral task was designed from the control unit using the program 'PhenoSoft Schedule', which offers a graphical interface for creating experimental schedules using state diagrams or flowcharts. Signals from all modules mounted on the Jet Ball unit were transferred

simultaneously from the ball holder controller to the control unit computer and analyzed.

## Mouse behavioral training: Virtual-Reality Response Learning (VR-RL) task

The behavioral training consisted of four phases, including the habituation, shaping (familiarization), acquisition, and reversal phases. To reduce stress, all mice were handled and habituated to the experimenter two days before head-fixation. During this period, each mouse was gently picked up from the cage, handled for a few minutes and allowed to return to their cages. Afterwards, all mice were habituated to head restraint for an additional 2–3 days. On the first day, each mouse was gently placed on the styrofoam ball and allowed to move freely for about 5–10 mins, without head fixation, and then returned to their cages. This was repeated 2–3 times. On the following days, mice were head-fixed on a metal head-holder (Thorlabs) for 10–15 min and given unpredicted drops of 10% sucrose solution (15 µL), at random intervals, through a reward spout placed directly in front of the mouth. This enabled mice to become habituated to the head-fixed condition and to interact with the VR system. Once mice began actively licking to consume rewards, they were subjected to successive stages of shaping. In this phase, mice were trained to run along a unidirectional, infinitely long virtual linear track (infinity corridor) with regularly spaced rewards. With this approach, mice adapted quickly to the VR, running for long distances within 1-2 weeks of initial exposure. In our experience, male mice adapted quicker and started free runs on the ball earlier than female mice. After these training sessions, mice were briefly introduced to the Y-maze task for a day, where each mouse was placed inside the maze and allowed to move freely and get rewards (warm-up trials), as used in previous studies[17,24]. These trials were not included in the analyses presented in the paper. Overall, it required 2-3 weeks of handling and pretraining for the mice to move freely on the ball and efficiently perform the VR task. Mice were then subjected to the initial task contingencies, i.e., acquisition phase. In this phase, the mice were trained on a self-initiated adaptive decision-making task requiring choice of the left or right arm of the Y-maze for reward. Mice were trained one session per day, with each session consisting of ~ 40 trials for a maximum duration of 1 h. Each trial began with a 3-kHz auditory tone (1 s; 75 dB) with 60 frames (2 s) before tone presentation serving as a baseline period, during which the virtual environment remained unresponsive. The mice learned through trial and error to identify which arm was rewarded. Entry into the outcome zone (320 cm from start) was detected by the VR function WaitForVrPositionInCompensation, which immediately activated the delivery of the chosen outcome. In rewarded outcomes, ~ 25 µL of sucrose solution was delivered for 1 s through a reward port, and mice had to extend their tongue out of the mouth to register as a lick. The reward delivery schedule was accomplished by applying the HandleDigOut function, which controlled the peristaltic pump via a TTL pulse from the control unit. In no-reward outcomes, the VR was deactivated and the visual scene immediately turned black, signifying trial failure. The StopMode function was used to freeze the VR environment at the time of the trial outcome. At the end of each trial, mice were teleported to the VR start position to initiate the next trial. Mice typically reached task proficiency (≥ 80% correct choices in at least one session) within 4–5 days. We defined choice as the percentage of the number of times the mouse chose the reward (correct) arm in a session. Mice were free to control the pace of each trial without time limits. The average duration from trial start to outcome decreased from 71.1 ± 10.4 s for early sessions to 28.1 ± 2.6 s for late sessions (Fig. 1h), over the 350 cm length of virtual corridors in the maze. This trial duration includes natural variations in approach speed, exploration and grooming.

After successfully learning the initial task contingencies, mice were then subjected to the reversal learning procedure, i.e., reversal phase. In the reversal session, we first subjected mice to 10 trials of the

initial task contingencies, after which mice were immediately subjected to the reversal procedure. During this phase, the reinforcement contingencies were reversed (switched), without any explicit signal or cue, such that the arm initially associated with reward became associated with no-reward (the reward-to-no reward reversal), and the arm associated with no-reward became associated with reward (the no reward-to-reward reversal). Before each training session, the styrofoam ball was wiped clean with 70% ethanol. All behavioral training and imaging were done in a dark environment with a constant background noise of ~ 50 dB.

## Two-photon imaging and data acquisition

About 4-5 weeks after the GRIN lens implantation, imaging experiments began. This was performed using a custom-built combined widefield and 2-photon microscope (INSS) integrated with the VR system. The 2-photon microscope was equipped with a tunable Ti:Sapphire excitation laser with a pulse duration of 100 fs and a repetition rate of $80 \pm 0.5$ MHz (Chameleon Discovery, Coherent) tuned to a wavelength of 930 nm. Scanning was performed using a resonant scanner (Thorlabs). Fluorescence photons were split with a dichroic mirror (DIC-495LP, Thorlabs). Green iAChSnFR fluorescence was isolated using a bandpass filter (525/50, Thorlabs) and detected using a GaAsP photomultiplier tube (H11706P-40, Hamamatsu Photonics). The signal was amplified using transimpedance amplifiers (TIA60, Thorlabs). Data were acquired and collected on a separate computer using ScanImage software[89] (Vidrio Technologies) that also controlled the microscope. Images were acquired at 30 Hz at a resolution of $512 \times 512$ pixels with a pixel dwell time of 87.9 ns. Imaging time series were synchronized with behavioral data logs via a TTL pulse from the control unit of the VR system. Laser power measured at the front of the objective (Plan Fluorite 20×, 0.5 NA, Nikon) ranged from 40–60 mW depending on the level of iAChSnFR expression. Optical power measurements were obtained prior to the start of the behavioral study using a thermal power detector (S370C, Thorlabs) connected to a meter console (PM100D, Thorlabs). At the start of each imaging session, we briefly employed widefield imaging to identify the imaging plane using images of the blood vessel pattern, and the focal plane was adjusted slowly until vascular structures and landmarks were clearly observed. Widefield imaging was accomplished with a CCD camera (Retiga Electro, Teledyne QImaging) in the green spectral range. Afterwards, we immediately switched to 2 P imaging without moving the animal. On subsequent days, all imaging planes were determined relative to a reference location, which was obtained on the first day of imaging in all mice. Light contamination in optical recordings was reduced by covering the VR monitors with a color filter (#342 Rose Pink, Rosco Laboratories Inc.). The 2 P/VR systems were seated on an air-spring vibration isolation table (Physio-Tech) and housed inside a frame box covered with a black curtain.

## Chemogenetic inhibition of cholinergic interneurons

For CIN inhibition experiments, we bilaterally injected a Cre-dependent hM4Di-mCherry, AAVDJ/8-hSyn-DIO-hM4D(Gi)-mCherry (AddGene #44362) into the DS of adult *ChAT*-Cre mice. During surgery, 300 nL of the viral construct was injected per hemisphere. Although injections were performed bilaterally in the dorsal striatum, they were centered primarily in the dorsomedial striatum (DMS), guided by both our imaging findings of enhanced medial ACh responses to non-reward and prior reports implicating the medial striatum in reversal learning and behavioral flexibility. Virus expression was achieved for a minimum of 2 weeks before behavioral training commenced. Following recovery, the mice were tested on the VR-RL task. Mice were first trained in task acquisition until they reached an acquisition criterion of 80% correct choice performance, after which the reversal phase began. A total of 10 mice successfully completed the initial task acquisition, and each mouse was assigned to one of two groups: experimental and

control groups. On the test day, the animals were weighed to determine the volume of solution that will be injected. Intraperitoneal injection of 0.3 mg/kg of DREADD-activating ligand, JHU37160 dihydrochloride (J60; HelloBio)[49] preceded reversal tests in experimental mice by ~30 minutes. JHU37160 dihydrochloride was first dissolved in sterile saline (0.9% NaCl), and stock solutions were prepared in advance. Control mice received the same amount of 0.9% sodium chloride (vehicle) before each test session. For behavioral testing, the required dosages were typically prepared and used on the same day. Any remaining solutions were stored at 4 °C and allowed to equilibrate to room temperature for at least one hour before using again.

## Quantification and statistical analysis
### Analysis of behavioral performance

Following multiple training sessions, mice behaviorally discriminated between the maze arms by displaying improved choice performance and a directional preference in anticipatory licking, particularly on approach to the reward arm. Using a selectivity index, choice performance was computed as:

$$\text{Choice selectivity index} = \frac{(\text{correct choices} - \text{incorrect choices})}{(\text{correct choices} + \text{incorrect choices})}$$

We classified training sessions as 'early' or 'late' in learning based on both selectivity score and day of training. A session was defined as 'early' in learning if the selectivity score was below 0.3, and the session was on day 2 or earlier. Sessions were defined as 'late' in learning if the selectivity score was $\geq 0.6$, and the session was on day 4 or later. Similarly, anticipatory licking was used as a measure of appetitive learning. We quantified anticipatory licking behavior in early and late sessions using a lick index, computed as:

$$\text{Lick frequency index} = \frac{(\text{licks to reward arm} - \text{licks to no} - \text{reward am})}{(\text{licks to reward arm} + \text{licks to no} - \text{reward am})}$$

To further confirm that mice had learned the task, we collected and analyzed the animals' velocities at which they completed trials, as a measure of adaptive motivation. The velocity data was obtained from the motion sensors that recorded all movements of the animal and translated it into VR coordinates. The mouse velocity was estimated as the moment-by-moment change in displacement, and computed as dY/dt, where dY was the change in Y-position displacement in the virtual environment, and dt was the elapsed time. To do this, we analyzed data in 40 cm bins for positions from 0 to 320 cm and averaged across trials for each mouse.

## Two-photon image processing

Offline data analysis and visualization of time-lapse recordings were performed with Fiji software (NIH; http://imagej.net/Fiji), MATLAB (MathWorks), and Prism 10 (GraphPad). First, raw movies acquired on the 2 P system were preprocessed in Fiji using a custom macro to automate the process. Time-lapse images were processed for x-y motion correction using a standard normalized cross-correlation method to eliminate any spatially uniform motion and movement artifact. This was accomplished with the 'Template Matching' plugin ('matchTemplate' function in the openCV library) in Fiji. The movies were then visually inspected to confirm drift correction. The movies were further processed by cropping out the edges of the imaging field to avoid artifacts due to lateral motion. Motion correction was carried out for each day of recording. Movies were denoised by applying a spatiotemporal Gaussian filter (sigma = 2 pixels in $x$, $y$, and $t$) to each frame of the field-of-view (FOV). Representative fluorescence (mean fluorescence projection) images were created by averaging the intensity Z projection of the motion-corrected image sequences. Our

imaging and data analysis pipeline minimized technical sources of noise by employing motion correction and baseline normalization.

**Selection of individual quadrats and calculation of ΔF/F.** To achieve this, we used a custom Fiji script to select individual quadrats and extract their corresponding fluorescence traces. We note that the iAChSnFR sensor was expressed in a non-cell-type-specific manner, labeling all neuronal elements, including somata, dendrites, and neuropil. As a result, the structures observed likely reflect a mixture of cell types and compartments, which can vary in size. Due to the diffuse expression of iAChSnFR sensor, the FOV was overlaid with equally sized quadrats (~50 × 50 μm) for selection and trace extraction. The use of this grid size was informed by several considerations beyond convenience. First, the choice of 50 × 50 μm quadrats was based on a practical compromise between spatial resolution and signal reliability. This size was chosen to ensure sufficient signal-to-noise ratio within each quadrat, while also capturing meaningful local variations in ACh dynamics across the FOV (Supplementary Fig. 4e). Second, CINs, which are the primary source of striatal ACh, are sparsely distributed but possess extensive axonal arborizations, spanning several hundred micrometers[90–92]. These broad arbors form spatially diffuse neuromodulatory fields that overlap and influence local microcircuits. However, 2-photon microscopy optically sections the image in the dorsoventral plane, and thus samples only a fraction of the arborization of any particular neuron. In addition, some spatial averaging is required to achieve a meaningful signal-to-noise ratio on a trial-by-trial basis. To ensure that our observations were not biased by our choice of quadrat size, we also conducted control analyses using different quadrat sizes (Supplementary Fig. 4e). Based on these, we divided the FOV into 15 × 15 grids and obtained a total of 225 individual quadrats in each mouse. The fluorescence trace of individual quadrats over time was calculated by averaging the corresponding pixel values within each specified region. Time-series traces of all processed movies were extracted and imported into MATLAB using custom-written scripts. Fluorescence time series were converted to ΔF/F by normalizing signals to the baseline fluorescence such that for each quadrat, the relative fluorescence change (referred to as ΔF/F) was calculated as:

$$\frac{\Delta F}{F} t = \frac{F(t) - F_0(t)}{F_0(t)}$$

where F(t) is the instantaneous fluorescence from the raw time-series data and $F_0(t)$ is the mean of the baseline fluorescence. Mean full-field fluorescence was calculated by averaging the mean fluorescence change of all quadrats in the FOV. iAChSnFR signals were then extracted around relevant behavioral events (e.g., trial outcome) and averaged across trials. For analysis of ACh responses in the outcome period during the acquisition phase, traces of ΔF/F from 1-s before to 1-s after trial outcome were used for the analyses. To quantify the changes in iAChSnFR signals across multiple animals or conditions, the ΔF/F time-series signal was further normalized using z-score (subtraction of mean and division by standard deviation calculated from a 1-s baseline period, such that z score = (ΔF/F − μ) / SD) to account for potential differences in signal variance across animals and sessions. Mean iAChSnFR traces were calculated using session-averaged traces from individual mice. The dataset of 425 trials (reward trials, 291 and no-reward, 134; acquisition phase) and 416 trials (reward trials, 213 and no-reward, 203; reversal phase) from 11 mice was used for the core of this study.

**Hierarchical clustering of quadrat responses and analysis of temporal dynamics.** Using a custom MATLAB program, we analyzed the spatiotemporal characteristics of ACh responses during the outcome period. For this purpose, we classified the response patterns of all recorded quadrats (2475 quadrats from 11 mice) using principal component analysis (PCA) to reduce dimensionality, followed by a hierarchical

clustering algorithm. Principal components explaining 90% of the total variance (Supplementary Fig. 2b) were retained to preserve key features. Hierarchical clustering was then applied to the PCA-transformed data using the Euclidean distance metric and the complete linkage method, which resulted in a clear separation of response clusters.

The optimal number of clusters was determined using the Elbow Method, based on the within-cluster sum of squares (WCSS). Finally, the clustered data was visualized in three-dimensional space using the first three principal components. Using this, the hierarchical clustering approach identified four clusters based on response kinetics, with distinct dynamics. Across the population, clusters were unequally represented, ranging from 2.3–60.1% for rewarded outcomes and 0.04–32.9% for unrewarded outcomes. The observed clusters displayed differences in response amplitudes and temporal kinetics. Thus, the four clusters were accordingly labeled as: transient decrease in ACh (Cluster 1), transient increase followed by a decrease (Cluster 2), transient decrease followed by a rebound burst (Cluster 3), and others consisting of an increase in ACh (Cluster 4). Following hierarchical clustering, we applied strict post hoc classification criteria to validate the functional significance of each response type using statistical thresholds (±2 SD). The 2 SD threshold applied here is a standard criterion for defining reliable responses in neural imaging[93]. A quadrat was classified as "decrease type" if its activity did not exceed 1 SD above baseline for any time steps from 0 to 1000 ms of reward delivery, and the minimum activity was less than 2 SD below baseline. A quadrat was identified as "increase-decrease" if its activity exceeded 2 SD above baseline for at least one time step from 0 to 400 ms of reward delivery, followed by a pause. A rebound burst after the decrease was defined as maximum activity exceeding 2 SD for at least one time step from 500 to 1000 ms of reward delivery. A quadrat was classified as "increase type" if the activity exceeded 2 SD above baseline during the time of reward delivery. To determine the temporal profiles of a quadrat, we calculated the time-to-dip as well as the dip amplitude (minimal change of iAChSnFR signal, 0-1 s after R/NR onset) of the z-scored ΔF/F response. Similarly, the time-to-peak and peak amplitude (maximal change) were estimated. While four distinct ACh response patterns were identified during late learning (*i.e.*, pause, burst-pause, pause-burst, and increase) (Fig. 3b–d), only two primary response clusters emerged during early learning, which we classify as long-latency pause (~21%) and short-latency pause (79%) (Supplementary Fig. 4g, h).

**Identification of switch and stay quadrats.** For the analysis of switch/stay encoding quadrats, we first grouped outcome responses on the basis of whether the animal would, in the upcoming trial(s), return to the previously incorrect arm (defined as "stay" trials) or select the now correct arm (defined as "switch" trials). To estimate the trial-by-trial fluctuations in ACh activity, we then calculated the mean evoked activity for individual quadrats by averaging the fluorescence signals across the corresponding trial type (Fig. 5e). For each quadrat, whether an outcome-evoked response was significantly different was determined by comparing the mean evoked fluorescence change (time window: 0.3–1.3 s after NR) against the baseline, which was the 1-s period preceding trial outcome. A quadrat was considered to encode "switch" or "stay" if it exhibited a significant change in mean activity (two-tailed, paired *t* test, $P < 0.05$) relative to the baseline activity, preceding future switching (switch-encoding quadrats) or staying (stay-encoding quadrats), respectively. Non-selective quadrats showed no significant changes from the baseline ($P > 0.05$). Afterwards, individual quadrats were grouped as switch+ (increased activity preceding future switching and non-responsive to future staying), switch- (decreased activity preceding future switching and non-responsive to future staying), stay+ (increased activity preceding future staying and non-responsive to future switching), stay- (decreased activity preceding future staying and non-responsive to future switching), etc. Thus,

we identified nine distinct groups post hoc based on the magnitude and direction of modulation. Color-coded maps of all quadrats were constructed by projecting the trial-averaged values of individual quadrats onto a two-dimensional space for data visualization.

## 3D spatial cluster mapping of imaged quadrats

To enable cross-subject comparison of quadrat spatial distribution, all fields of view were spatially registered onto a common reference framework. For each animal, anatomical coordinates of GRIN lens implant locations were estimated using the nearest corresponding coronal section of the mouse brain atlas, based on histological verification. The mediolateral (ML) and anteroposterior (AP) center of each imaging field was defined by the implant location, serving as the spatial origin for quadrat transformation.

Within each FOV, the spatial position of each quadrat was mapped onto a standardized $15 \times 15$ grid representing a fixed imaging field (typically 225 quadrats), with each grid square corresponding to a constant physical spacing between quadrats. Quadrat positions were computed relative to the ML and AP implant center, scaled by a fixed spacing constant ( ~ 50 μm) and adjusted by a uniform scaling factor of 1. Coordinates were also mirrored along the AP or ML axis if the orientation of the GRIN lens required anatomical inversion, as specified in the metadata for each mouse (described below). This normalization approach ensured that quadrat locations across animals were anatomically aligned in a common reference plane, allowing for spatially consistent visualization and comparison of activity patterns.

Spatial localization of quadrats within the imaging plane was achieved using a custom script developed in MATLAB. The input data consisted of multiple excel sheets, each containing quadrat indices alongside functional classifications (e.g., cluster 1, cluster 2, cluster 3, cluster 4 in Fig. 3g, and increase, decrease, non-selective in Fig. 5d). Physical coordinates were centered using recorded ML and AP metadata values extracted from the input sheets. Notably, the script accounted for potential axis inversions resulting from optical properties of the GRIN lens (e.g., mirroring or flipping across ML and AP axes) and image orientation effects introduced by the two-photon imaging system. Each quadrat was visualized as an open circle plotted using scatter and scatter3 functions, with edge colors assigned to each functional category. Two-dimensional and three-dimensional scatter plots were generated per sheet, and an aggregate 2D and 3D map was created by merging all sheets, using the built-in MATLAB plotting engine.

## Heatmap z-score visualization: Central vs Peripheral quadrat mapping

To spatially visualize mean z-score activity across quadrats (Supplementary Fig. 7c, d), a custom MATLAB script was employed. Input data for each imaging session consisted of excel sheets containing paired columns for quadrat indices and their corresponding mean z-scores. Each quadrat was assigned to a standard $15 \times 15$ spatial grid based on quadrat order, enabling consistent cross-session mapping. Z-score values were visualized using MATLAB's heatmap function. The imaging field was subdivided into "central" and "peripheral" regions based on quadrat proximity to the center: Quadrats located within grids 3–13 in both ML and AP axes were defined as "central," while those within the two outermost rows and columns (grids 1–2 and 14–15) were classified as "peripheral".

## Analysis of behavioral measures in reversal learning and chemogenetics experiment

To examine the treatment effects, behavioral measures, including perseverative and regressive errors, were analyzed for the reversal learning session, as reported in previous studies[18,19]. Perseverative errors are repeated incorrect choices made after the reversal but before the mouse has exhibited learning of the correct response.

Specifically, a perseverative error was defined as errors made in a block of four consecutive trials after reversal, of which 3 or more were incorrect. The measure of perseveration was the number of such errors that occurred in sequence. The end of the sequence was defined by the occurrence of a block of four trials in which 2 or more trials were correct. Regressive errors are incorrect choices that occur after learning has occurred, as evidenced by a sequence of correct choices. Here, a regressive error was defined as any block of four trials in which an error occurred after a block in which 2 or more trials were correct. Perseverative and regressive measures are thus distinguished from individual win-stay and lose-shift responses that are trial-by-trial choices made in the context of the outcome of the previous trial. A two-way repeated measures (RM) ANOVA followed by Tukey's post hoc test was used to further analyze differences in choice performance between treatment groups. Following all behavioral tests, mice were transcardially perfused, brains sectioned, and CIN-specific expression confirmed using mCherry reporter expression, which co-localized exclusively with *ChAT* immunoreactivity.

## Effect size analysis

To complement null-hypothesis significance testing and better quantify the magnitude of observed group differences, we calculated Cohen's *d* for all pairwise comparisons. For unpaired *t* tests, Cohen's d was computed as the standardized difference in means between treatment groups, using the pooled standard deviation as the denominator. 95% confidence intervals were also estimated to provide an index of precision. Effect sizes were interpreted according to conventional benchmarks (d ≈ 0.2: small; d ≈ 0.5: medium; $d \geq 0.8$: large; negative values in figures indicate direction of effect)[94]. Reporting effect sizes is particularly important in studies with modest sample sizes, as it provides information about the strength and potential biological relevance of effects beyond p values alone.

## Histology

After completion of behavioral experiments, all mice were examined for histology to confirm viral expression and GRIN lens placements. Briefly, mice were anesthetized with 0.3–0.4 ml (i.p.) of thymylal sodium (Nichi-Iko) and transcardially perfused with 20–30 ml of phosphate-buffered saline (PBS, Wako) followed by 10–15 ml of 4% paraformaldehyde (PFA, Nacalai Tesque) in PBS. The brains were dissected and post-fixed in 4% PFA overnight at 4 °C, and then cryoprotected in 30% PBS-buffered sucrose solution for about 3–5 days. Afterwards, the brain was embedded into tissue-freezing medium and sectioned coronally on a retoratome (REM-710, Yamato) at 40 μm and stored at 4 °C in PBS until use. Histological verification of sensor expression was performed following standard procedures. To label iAChSnFR, slices were first rinsed (3 × 10 min) in PBS. Next, the sections were blocked with 5% normal goat serum (Jackson ImmunoResearch) in PBST (0.5% Triton X-100 in PBS) (Sigma-Aldrich, Cat#T8787) for 1 hr at room temperature (RT), and then incubated overnight at 4 °C with chicken anti-GFP primary antibody (Abcam, Cat#ab13970, 1:500) and 2% normal goat serum in PBST. On the next day, sections were washed in PBS (3 × 10 min, RT) and incubated with Alexa Fluor 488 goat anti-chicken secondary antibody (Invitrogen, Cat#A32931, 1:500) for 3–4 hr at RT. Then, the slices were washed (3 × 5 min in PBS) and mounted on glass slides. For *ChAT* staining, sections were incubated overnight with goat anti-ChAT primary antibody (Millipore, Cat#AB144P, 1:100) after blocking. The sections were then washed with PBS (5 × 10 min) and then incubated (3–4 hr, RT) with Alexa Fluor 488 donkey anti-goat secondary antibody (Invitrogen, Cat#A11055, 1:500) in PBST and washed again. Nuclei were stained with DAPI (4′,6-diamidino-2-phenylindole; Invitrogen, Cat#R37606; 70 μL/mL) in PBS. Sections were coverslipped using mounting medium (ProLong glass antifade mountant, Invitrogen). Whole sections showing viral expression and GRIN lens placements were imaged using an epifluorescence

microscope (BZ9000, Keyence) whiles cellular resolution images of immunostained slices were acquired using a laser scanning confocal microscope (LSM 780, Zeiss).

## Graphical and schematic illustration

Schematics of experimental setups and brain outlines depicted in the paper were created using Adobe Illustrator (Adobe Creative Cloud, version 29.8.3; Adobe Inc.). Each element was hand-drawn using the built-in Shape tools (Ellipse, Rectangle, Line Segment) and Symbols (vector icons available within Adobe Illustrator's default Libraries).

## Statistical analyses

All statistical analyses were performed using GraphPad Prism 10. Significance for comparisons were set to $*P < 0.05$, $**P < 0.01$, $***P < 0.001$, and $****P < 0.0001$. We used two-tailed, paired or unpaired Student's $t$ test for two-group comparisons, and Two-way RM ANOVA followed by Tukey's post hoc test for multiple comparisons. In box plots, center lines depict the median, box limits represent the 25th and 75th percentiles, and whiskers, data range. Correlation analyses were performed using Pearson's correlation coefficients ($P < 0.05$).

## Data availability

The data generated in this study have been deposited in the Zenodo database under accession code 10.5281/zenodo.17421544 (https://doi.org/10.5281/zenodo.17421544). Source data are provided in this paper.

## Code availability

Custom MATLAB codes, demo data/videos and Fiji scripts used for data analysis are available on github. https://github.com/gasarpong/Sarpong-etal_ACh_spatiotemporal_dynamics and https://github.com/kang62489/PG_006?tab=readme-ov-file.

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

## Acknowledgements

This work was funded by a research grant from the Human Frontier Science Program (HFSP) No. RGP0062/2019 to J.R.W., subsidy fund from OIST, and supported by the Japan Society for the Promotion of Science (JSPS) KAKENHI (Grant numbers, 22K15633 and 24K10485) to G.A.S. We appreciate Dr. Nobuyoshi Kitamura for technical assistance in optimizing the 2-photon system for in vivo imaging, and Ms. Yukako Suzuki for administrative assistance. We are grateful for the help and support provided by the Animal Resources Section and Scientific Imaging Section of Core Facilities at Okinawa Institute of Science and Technology Graduate University.

## Author contributions

Conceptualization: G.A.S. & J.R.W.; Methodology: G.A.S., R.P., K.C., Y.A., J.A.C., and J.R.W.; Software: G.A.S., K.L., and K.C.; Analysis and Interpretation: G.A.S., K.L., K.C., and J.R.W; Visualization: G.A.S. and J.R.W; Resources: K.K., L.L.L., and J.R.W.; Data Curation: G.A.S., K.C., and J.R.W.; Writing and Revision: G.A.S., K.C., and J.R.W.; Supervision: J.R.W.; Project Administration and Funding Acquisition: J.R.W. All authors have read and edited the manuscript.

## Competing interests

The authors declare no competing interests.
