## [Transparent Peer Review file · Nature Communications]

Spatially heterogeneous acetylcholine dynamics in the striatum promote behavioral flexibility

Corresponding Author: Professor Jeffery Wickens

Version 0:

Reviewer comments:

Reviewer #1

(Remarks to the Author)
Sarpong et al

This is a very interesting paper that examines changes in acetylcholine (ACh) release in the dorsal striatum of mice during performance on a Y maze discrimination and after the reversal of the discrimination. The protocol uses head fixed mice on a virtual maze allowing clear measures of ACh dynamics and localised changes in identified clusters of regionally defined subregions with the field of view in the striatum divided into a 15 x 15 grid. This provided further detail by providing an assessment of the heterogeneity of changes in dynamic responses. The authors focus their assessment of ACh activity to the period during which reward was delivered after a correct response in the maze or during a comparable period of non-reward after an incorrect response. This provides a relevant but limited period of assessment. Thus, for example, ACh activity during the anticipation of reward was not reported, either pre- or post-choice as the animal approach the choice point and after it had selected an arm in the maze. This could have been informative. Nevertheless, the authors report considerable behavioural data from this period, including measure of anticipation, such as anticipatory licking. The findings are reasonably clear, showing large differences in responding to reward and non-reward during initial training and after reversal. Interestingly, reward-related activity was found to be heterogeneous across regions: in most clusters showing a pause, sometimes after a preceding increase in activity, but also, in other clusters, a rebound excitation after the pause within the field of view. Even more interesting was the finding that an increase in ACh activity after non-reward predicted a shift in arm selection (i.e., lose-shift performance) but only after reversal suggesting increased ACh release is relevant to acquiring the reversed contingencies. Finally, to establish the causal relevance of ACh release to reversal performance, a Cre-dependent inhibitory DREADD was infused into the dorsal striatum in Chat-Cre mice. Inhibition of ACh reduced reversal performance as well as lose-shift probability.

Overall, this is a nice set of findings. Nevertheless, I had a few issues with the presentation and some questions regarding the interpretation of the findings.

1. The paper is framed in terms of the role of ACh in cognitive flexibility and, indeed, a large number of variables were analysed from the behavior of the mice during the acquisition of maze learning. However, monitoring of ACh activity was limited to reward delivery or non-delivery period only. Despite the framing in terms of flexibility, therefore, the relationship between ACh release and performance is difficult to draw. The first 3 figures focus on ACh release during initial acquisition of performance on the Y maze. However, although performance is differentially rewarded, and there are both rewarded and unrewarded movements, at this stage, according to the release dynamics during reward/non-reward (without consideration of the degree to which it is anticipated) ACh appears to be responding solely to reward. When no reward is delivered there is no, or at least a much diminished, response. So even when the animal makes an incorrect response it is hard to see that ACh is signalling anything useful to the animal to alter its behaviour. How does this accord with the behavioral flexibility hypothesis? Surely, acquiring performance on the maze requires learning and so some degree of flexibility?
2. Along these lines, what do the animals actually do when unrewarded? I was interested in two things here: (i) Is the degree of ACh response to non-reward at all modulated by ACh release in anticipation of reward? Does it differ according to the degree the animals are anticipating reward prior to non-reward? (ii) Is the ACh response influenced by the animals' behavioral responses after non-reward at all - e.g., degree of orienting/search or activity perhaps related to frustration? We are not told. There is considerable detail regarding the animals' behaviour up to reward delivery but not afterwards.

3. Relatedly, I found myself wondering if the change in ACh response after reversal in the presence of non-reward is in anyway related to motor activity/orienting/attentional changes and whether these can be detected in animals' behavior as a change in activity after reversal. The increase in ACh after an incorrect response during reversal is interesting but, within-animals, how different is this response to that observed to non-reward during training and during the period immediately prior to reversal? The authors state: "Thus, the direction and magnitude of ACh signals during the no-reward outcome period predicted future behavioral shifts." Do they predict this because they are indicative of learning or because of some reaction the animals have to non-reward? There is a quite clear analysis of behavioral changes associated with reward. What would a similarly complete analysis of behavioral responses to non-reward look like and could this anticipate the responses after reversal? I suppose it is worth also considering here the role of shifts in 'mood' – by which I mean responses to frustrative non-reward, and in orienting rather than simply changes in behavior generally.

4. Where changes in ACh signals observed in anticipation of reward? One assumes that the signals would have been similar prior to correct or incorrect choices if the animals were anticipating reward similarly prior to choice. However, only 1s of baseline was used prior to reward signals. Does df/f change with a longer baseline?

Reviewer #2

(Remarks to the Author)

The manuscript describes findings from a series of experiments that demonstrates the dynamic changes in dorsal striatal acetylcholine efflux during learning and reversal learning. There are several innovations carried out in these studies that provide significant insights of how actions of striatal cholinergic interneurons underlie learning and behavioral flexibility.

At the core there are three main experiments. The first combining a genetically encoded fluorescent acetylcholine sensor with 2-photon microscopy to determine how acetylcholine within subregions of the dorsal striatum changes during learning and reversal learning. A second study was conducted in striatal slices demonstrating that the fluorescent sensor was actually measuring acetylcholine. A third experiment employing chemogenetics selectively reduced striatal acetylcholine activity and showed that this manipulation impaired reversal learning consistent with what would be predicted from the findings in their first experiment.

What stands out about this manuscript is the creativity in carrying out their first and main experiment investigating striatal acetylcholine changes during learning and reversal learning. Integrating modern techniques the main experiment is able to reveal how dynamic changes in dorsal striatal acetylcholine output supports learning and behavioral flexibility at a spatiotemporal scale that has not been achieved before. Further, the breadth and depth of their analyses really provides a comprehensive picture of the how dynamic changes in acetylcholine occur in striatal subregions. Overall, the findings from these experiments make significant advances in revealing how striatal cholinergic interneuron activity facilitates learning and behavioral flexibility. These findings will have a broad interest for anyone interested in the brain mechanisms that support learning and behavioral flexibility.

I have two main comments related to the manuscript:

1. The authors use the term "disappointment" referring to when a mouse makes an incorrect choice that does not lead to a reward. This term really seems like a projection of a human experience onto the mouse that is not necessary. The results are extremely strong and convincing so using an alternative term that more simply captures the behavior would suffice, i.e. unexpected non-reward.

2. A real strength of the manuscript is the hierarchical clustering of acetylcholine responses within subregions. I was not clear whether the data included in this analysis was for all trials in all sessions mice were tested or only for a subset of trials, i.e. late learning? Related, did the same clustering pattern occur throughout testing or was it more prominent in a particular stage of learning, i.e. late vs. early.

Reviewer #3

(Remarks to the Author)

Sarpong et al. investigate the spatiotemporal dynamics of acetylcholine (ACh) in the dorsal striatum, using a genetically encoded sensor in head-fixed mice choosing between left and right paths in a virtual environment. This is an important topic and they observe some interesting results. The primary finding is that under reversal conditions (reward location switches from right to left) arrival at the no-longer-rewarded location causes an increase in ACh, rather than the dip seen with rewards. The size of this no-reward increase is predictive of behavioral switching, and suppressing ACh interneurons makes switching slower. They also find limited evidence for spatial heterogeneity in the ACh response. Overall there are some noteworthy observations here, but the key claims lack adequate support and/or are over-interpreted.

One major concern is about the claimed spatial heterogeneity of ACh (which is the primary novel aspect of this work). To investigate this they divide their field of view into 50x50 micron regions of interest, and analyze them separately. They perform cluster analysis on the ACh response and claim distinct functional clusters. This was not very convincing for several reasons:

- First, there's not a clear separation between clusters. It appears more like a continuum with subtle differences between the relative heights of the increases and decreases. There's no clear indication of statistical significance in the difference

between response types, which is not surprising given that a) the response is highly variable from one trial to the next, and b) there are very few trials per day (~40), and that includes both rewarded and unrewarded, so the number for one category or the other in a given recording session may be very small indeed). How can the authors rule out the alternative possibility that the underlying ACh signal is exactly the same throughout the FOV, but with spatially variable noise?

- Second, why 50x50 microns? This seems like an arbitrary choice that has nothing to do with the circuit organization of the striatum. Are the results different if this choice is different? It's not reasonable to refer to these arbitrary zones as striatal "subregions". And without some reason to choose these particular 50x50 zones, they remain a "bulk" measure of neuromodulator signaling like fiber photometry, and the approach is less useful than it should be (less useful than e.g. recent micro-fiber array work such as the cited Bouabid et al reference).

- Third, the images shown show clear spatial structure in the fluorescence signal. As expected for a GRIN lens the signal is much fainter at the edges - yet the edges are equally included in the analysis and seem to show similar results. Furthermore, the center of the image shows clear structure of some form... are these dendrites or what? What happens if the ROI is chosen to match the locations of expression - does the behavior-linked signal get stronger, as might be expected? I.e. how does $\Delta F/F$ depend on F ? It was also a concern that the apparent neurites visible in the images don't seem to scale in size in a way consistent with the scale bars (e.g. comparing Fig. 2b to 2d).

Many prior studies have shown the distinct properties of different dorsal striatal subregions (defined especially along the medial-lateral axis, also the anterior-posterior axis). This aspect is not adequately considered here in the experimental design and analyses. E.g. the GRIN lens covers a fairly wide medial-lateral extent of the dorsal striatum, and based on prior work we might expect more medial parts to be more involved in reversal learning / behavioral flexibility - this should be examined.

The behavioral task is problematic in several aspects. Besides the low number of trials, just a single reversal further reduces the amount of data available for analysis. Some analyses have notable confounds - e.g. in Fig 4j. It's not reasonable to directly compare the lose-shift and lose-stay trials in this way, because (on average) they will be happening at different times after reversal (later and earlier respectively). Each trial is surprisingly long - around ~35s even in the "late" phase of learning. Why are the mice so slow, and what are they doing in all this time? Overall there seems to be a poor match between the behavioral task design, recording methodology and the analysis approach that precludes strong novel conclusions about striatal ACh function.

Other points:

To assess the imaging conditions it would also be helpful if the sensor used here was better characterized - is the preprint from 2020 still the most up-to-date reference?

e.g. line 131: "iAChSnFR is very fast, so the timing of these dips is reflective of ACh, not a sensor artifact."

- we need specific support for these assertions.

Discussion is often repetitive and/or confusingly written. Some examples:

- line 270 "subregions responded to various combinations of expected and unexpected rewards and non-rewards associated with both shift and stay responses."

- line 285 "However, these behavioral tasks did not provide a reward alternative."

- the two sentences on lines 293-297 seem to say the same thing.

- too many mentions of "the causal relationship" between ACh and lose-shift behavior.

- mention of travelling v standing waves not clear enough for most readers, especially if they haven't read the cited paper.

In terms of interpreting the results...

- how much of the ACh increase is about the omission of an expected reward, and how much about the active presentation of unexpected non-reward cue?

- line 314 "Thus, we propose that neural pathways exposed to increased ACh on unexpected non-reward may be preserved, protecting old learning" - wouldn't that lead to "lack" of behavioral change, rather than enhanced change (switching)?

line 63: "mice were trained in daily sessions of ~40 trials until they reached acquisition criterion (80% correct choices)"

- over what ... a session?

"A1-A5" not explained - presumably the acquisition sessions 1-5?

line 71: "Increases in anticipatory licking were positively correlated with advancement in the maze"

- actually what is shown is "lick index" - the proportion of licks that are to the rewarded side (if I understood correctly). So this is not quite the same as an increase in anticipatory licking.

line 81: "reduced trial latency"

- latency here seems to mean duration - it would be better to just say duration, as latency is used in other work to mean e.g. time to begin a trial, rather than complete it.

line 93: "The response to ACh was later blocked with neostigmine."

- define "later".

- why is the neostigmine trace noisier, and why does it decrease with ACh puff?

Fig. 2d: ROIs very hard to see.

Figs 2d, e, etc.. state the number of individual trials included in each average.

line 99: "The features of these dips in ACh levels (Fig. 2f,g; dip amplitude, -1.64 ± 0.07 z-scored $\Delta F/F$; onset, 0.23 ± 0.01 s; dip latency i.e., time to dip, 0.48 ± 0.03 s, mean \pm s.e.m., $n = 11$) are consistent with pauses in tonic activity"

- what exactly makes these numbers consistent with pauses?

line 105: "We also noted some variability in reward responses across trials (example mouse in Fig. 2e, left), as previously reported"

- how do the authors distinguish real biological variability from trial to trial, from measurement noise?

Fig.3bii error bars are very small for these clusters - looks like simultaneously recorded ROIs from the same FOV are being treated as independent? would make more sense to average within clusters from the same experiment first, then calculate error bars at the animal level.

Fig3D is hard to understand - e.g. does the cluster 2 NR 100% bar actually indicate that none of the subregions had a dip within 1 second? It would be useful to show the mean NR responses for each cluster.

Fig. 4b: could be helpful to include trials before reversal, as is standard.

line 141: "The proportion of perseverative errors was $17.1\% \pm 2.7\%$ "

- here and elsewhere, need to specify exactly which trials/sessions are included.

line 152: " Incorrect responses, however, were associated with a delayed, progressive increase in the spatially averaged levels of ACh"

- what exactly makes it "progressive"? Not clear enough.

- how does the high variability shown in 4i relate to the outcome of the previous (and earlier) trials?

- where in the FOV are the 4i examples taken from?

line 239: "Additionally, cumulative errors over the first 80 trials post-reversal were significantly higher in J60-treated mice compared to saline-treated controls (Fig. 6f)."

- not clear what this additional analysis adds to the story.

line 245: "While the difference in perseverative errors was not significant ($P = 0.47$, unpaired t-test), a significant difference in the proportion of regressive errors was observed between the two groups"

- need to explain "perseverative" vs "regression" and contrast to lose-stay v win-shift (seems different given Fig. 6g)

line 253: "Although our inhibition protocol lacked temporal specificity, our findings align with our initial observation of decreased switch probability in mice with reduced ACh levels."

- the initial observation doesn't measure ACh "levels".

Reviewer #4

(Remarks to the Author)

Version 1:

Reviewer comments:

Reviewer #1

(Remarks to the Author)

The authors have very thoroughly revised the original submission. they have added considerably more data regarding the behavior of their mice during the training trials, after choice and reward exposure and, in the process, have addressed the points I raised with regard to their original submission. I was particularly impressed by how thoroughly they addressed the points regarding the definition of cognitive flexibility here. I do agree that CINs are likely to be centrally involved in updating prior learning after contingencies have changed and I think that they have now clarified where and why those changes occur in the current task. I was also impressed by the way they have handled alternative explanations of their data. I find my prior comments have been addressed and that I have no further comments to make. From my perspective it is acceptable in its current form.

Reviewer #2

(Remarks to the Author)

The manuscript describes findings from a series of experiments that demonstrates the dynamic changes in dorsal striatal acetylcholine efflux during learning and reversal learning. There are several innovations carried out in these studies that provide significant insights of how actions of striatal cholinergic interneurons underlie learning and behavioral flexibility.

The original submission represented a very important set of experiments and results addressing an important issue about brain-behavioral relationships. In particular, how does cholinergic signaling in the striatum contribute to behavioral flexibility. In the revised version of the manuscript, the authors have clarified a lot of questions raised by reviewers by conducting several different analyses. These additional analyses clarify what conditions lead to changes in striatal ACh signaling and what conditions do not change striatal ACh signaling. Overall, the comprehensive analyses conducted provide some real insights and make a strong argument for striatal ACh contributing to behavioral flexibility.

Reviewer #3

(Remarks to the Author)

Re-review of Sarpong et al.

As noted in my initial review, this manuscript includes some interesting observations, and I applaud the authors' efforts to use 2p imaging to probe the spatiotemporal organization of ACh signals. At the same time, these efforts, as currently presented, do not seem to have been particularly successful. To see this, look at the title and abstract: essentially none of the results mentioned there rely on the imaging - they could more easily have been obtained with simple fiber photometry. This feels like a missed opportunity. It's also a mismatch to how the manuscript is presented, which repeatedly emphasizes the novelty of and "new insight" gained from the imaging method. When the image is divided into 50um x 50um zones the authors observe some heterogeneity between zones. But beyond the apparent fact of heterogeneity, there's not much learned about the spatiotemporal scales or patterns of ACh release, and how these relate to anatomy or physiology. Overall the results are somewhat interesting to specialists but represent a modest advance over our prior knowledge of striatal ACh and behavioral flexibility.

Other points:

line 37: "more formally..." - this doesn't seem especially more "formal" than the prior sentences.

line 65: why was the reversal performed between, rather than within, sessions? Wouldn't within-session provide more power to detect neural changes (although the low number of trials may cancel this out).

line 69: "average percent correct was 82.1%..." when? during the sessions defined as happening after an 80% correct session?

line 116: "...time to dip (0.48s) is comparable to the timecourse of physiological action of ACh..." why is the response time of striatal neurons a reasonable comparison, given time taken for GPCR-mediated action of ACh?

line 126: "(confirmed by comparison with ACh insensitive signals...". The tdTomato signal is barely used in the manuscript - all the main results should include comparison to tdTomato to help rule out contributions of movement artifacts etc.

line 151: "...indicating weak separation between clusters lying along a continuum (Extended Data Fig. 4d). This is consistent with the fact that the concentration measures in the analysis are from physically adjacent quadrats along continuous concentration gradients"

- if true, this seems like an acknowledgement that clustering doesn't really make sense here.

- although, the quadrats within the clusters don't seem to generally appear physically contiguous, at least in the examples shown. Is this because the real spatial scale of ACh signals is comparable (or smaller) to 50µm? Or because the statistical criterion - an arbitrarily chosen 2SD - is not rigorous enough? Not enough is done to address such concerns.

line 202 "...increase in spatially averaged ACh levels that occurred gradually over seconds...". This is certainly an interesting observation, although given that the size of the increase is very modest (<1 Z) and ACh can change very fast, it's harder to have confidence this is a discrete, specific signal being used by the brain.

line 232: "...confirming that this signal is not only present but more spatially widespread during behavioral switching (Extended Data Fig. 6d)."

- this figure panel is important enough to the authors' argument to be in the main figures.

line 233: "...the ACh increases observed in response to non-reward during acquisition, present in a subset of individual quadrats (cluster 4 in Fig. 4b-d), may reflect a preexisting substrate for outcome-related ACh modulation". If this is correct, would it not imply that the specific quadrats with increases during acquisition would also show (bigger) increases during

reversal? Is this true?

More generally, how consistent are the quadrat-level signals from one imaging day to the next? Showing consistency could boost confidence in the quadrat-level results. Determining the spatial scale of such consistency might allow the authors to pick a quadrat size in a principled manner.

line 239: "The change in ACh response after unexpected non-reward could, in theory, be related to motor activity due to orienting responses. To address this, we analyzed the trial-by-trial velocities ..."

- velocities are not the same as orienting.

line 288-294: The new results about subregions are interesting and relevant and as the authors note they are consistent with prior results, though it's surprising that these findings barely, or do not, reach significance (and are not visible in the "3D map") What if "quadrats" are dropped and the authors analyze much broader fields-of-view?

line 348: "...hM4D(Gi)-mCherry bilaterally into the DS of ChAT-Cre mice..."

the "DS" here is actually "DMS", which makes sense given prior results but the imaging field is not consistently DMS - why?

line 360: "Cumulative errors over the first 80 trials post-reversal were higher in CIN-inhibited mice, reflecting impaired overall task adaptation beyond specific error types (Fig. 6f)." How does this observation support this conclusion?

line 364: "While CIN inhibition reduced lose-shift behavior (Fig. 6g, left; unpaired t-test, $P = 0.015$), it did not alter win-stay probability following the reversal (Fig. 6g, right; $P = 0.14$). " Judging by Fig. 6g, this is not a convincing result. The analysis appears to lack statistical power due to few, scattered data points.

line 497: "we propose that neural pathways exposed to decreased ACh on reward delivery – causing a pause after successfully shifting – would experience synaptic potentiation". It's not fully clear from the figures, but it looks like decrease ACh on reward delivery is strong even at the end of behavioral acquisition - would it really be helpful to keep potentiating synapses at that point?

Fig. 1c: please show the data divided into the individual sessions.

Fig. 1f/g (and later figures): since the point is to compare "early" and "late", the statistical test should incorporate both these conditions (e.g. as a factor in ANOVA) rather than running separate tests for each condition.

Fig. 4b : why show the orange and blue ticks twice each?

Ex Fig 6b: why are these particular three "representative" mice shown, rather than all of them? The right part of this panel is hard to follow and the argument presented from these individual mice is unconvincing.

Reviewer #4

(Remarks to the Author)

Version 2:

Reviewer comments:

Reviewer #3

(Remarks to the Author)

I don't have much to add to the previous rounds of review. I appreciate that the authors have fixed some minor things here and there, and I appreciate that a lot of work went into collecting and analyzing this data. But my major concern is that the imaging just hasn't revealed much about the functional properties of striatal ACh signaling, beyond the fact of heterogeneity. I'm very interested in this subfield, but despite that I don't find the concrete results here especially novel or interesting and I do not think they will have broad impact.

Reviewer #4

(Remarks to the Author)

RESPONSE TO REVIEWER COMMENTS

We thank the reviewers for their detailed and thoughtful assessment of our work and for recognizing the importance and broader implications. Their comments have been invaluable in improving the quality of our manuscript. We have addressed all comments as follows. Changes to the manuscript text are highlighted in red.

Reviewer #1 (Remarks to the Author):

Sarpong et al

Reviewer comment

This is a very interesting paper that examines changes in acetylcholine (ACh) release in the dorsal striatum of mice during performance on a Y maze discrimination and after the reversal of the discrimination. The protocol uses head fixed mice on a virtual maze allowing clear measures of ACh dynamics and localised changes in identified clusters of regionally defined subregions with the field of view in the striatum divided into a 15 x 15 grid. This provided further detail by providing an assessment of the heterogeneity of changes in dynamic responses. The authors focus their assessment of ACh activity to the period during which reward was delivered after a correct response in the maze or during a comparable period of non-reward after an incorrect response. This provides a relevant but limited period of assessment. Thus, for example, ACh activity during the anticipation of reward was not reported, either pre- or post-choice as the animal approach the choice point and after it had selected an arm in the maze. This could have been informative. Nevertheless, the authors report considerable behavioural data from this period, including measure of anticipation, such as anticipatory licking. The findings are reasonably clear, showing large differences in responding to reward and non-reward during initial training and after reversal. Interestingly, reward-related activity was found to be heterogeneous across regions: in most clusters showing a pause, sometimes after a preceding increase in activity, but also, in other clusters, a rebound excitation after the pause within the field of view. Even more interesting was the finding that an increase in ACh activity after non-reward predicted a shift in arm selection (i.e., lose-shift performance) but only after reversal suggesting increased ACh release is relevant to acquiring the reversed contingencies. Finally, to establish the causal relevance of ACh release to reversal performance, a Cre-dependent inhibitory DREADD was infused into the dorsal striatum in Chat-Cre mice. Inhibition of ACh reduced reversal performance as well as lose-shift probability.

Overall, this is a nice set of findings. Nevertheless, I had a few issues with the presentation and some questions regarding the interpretation of the findings.

1. The paper is framed in terms of the role of ACh in cognitive flexibility and, indeed, a large number of variables were analysed from the behavior of the mice during the acquisition of maze learning. However, monitoring of ACH activity was limited to reward delivery or non-delivery

period only. Despite the framing in terms of flexibility, therefore, the relationship between ACh release and performance is difficult to draw

Author response

We agree that it would be useful to add information about the ACh activity during the approach to the choice point. We have added data and further analyses of ACh signaling during approach (Extended data Fig 3) and report this in the manuscript as follows (line 129):

“As mice approached the outcome point of the maze, spatially averaged ACh signals increased progressively until the mouse reached the outcome zone (Extended Data Figure 3a). The increase was similar in rewarded and non-rewarded outcomes. ACh signals across trials were classified according to whether linear regression analysis showed significant positive or negative slope during the last 5 s preceding trial outcome. The greatest proportion of rewarded and unrewarded trials showed positive ramping (52% and 48%, respectively), and less commonly negative (32% and 31%, respectively) or non-significant changes (Extended Data Figure 3b, c). These distributions were not statistically different across conditions ($X^2(2) = 1.52, P = 0.46$).”

Reviewer comment

The first 3 figures focus on ACh release during initial acquisition of performance on the Y maze. However, although performance is differentially rewarded, and there are both rewarded and unrewarded movements, at this stage, according to the release dynamics during reward/non-reward (without consideration of the degree to which its anticipated) ACh appears to be responding solely to reward. When no reward is delivered there is no, or at least a much diminished, response. So even when the animal makes an incorrect response its hard to see that Ach is signalling anything useful to the animal to alter its behaviour.

Author response

We did not explain what we meant by “behavioral flexibility” very well in the original manuscript. By “behavioral flexibility” we mean “the ability to adjust behavior in response to changes in the environment”. Thus, we consider behavioral flexibility to be a step beyond adapting to fixed contingencies in a constant environment, toward adopting a new strategy when contingencies change. By this definition, learning the initial maze is not behavioral flexibility because the contingencies (i.e. the rewarded side) are kept constant during the initial learning. We now define what we mean more clearly in the manuscript, as follows (line 36):

“More formally, we define behavioral flexibility as the ability to adjust behavior in response to changes in the environmental contingencies. Thus, behavioral flexibility requires adopting a new strategy, which is a step beyond learning under constant rules.”

Having clarified our intended meaning, we agree with the reviewer that during the initial learning under constant reward conditions, ACh appears to be responding solely to reward. When no reward

is delivered there is a diminished response, in the same direction as when reward is delivered, but much smaller. We think it is consistent with our hypothesis that non reward due to incorrect responding in a constant environment does not engage ACh signalling. We have added the following comment to the manuscript to clarify this point (line 442):

“During initial learning ACh appears to be responding solely to reward. When no reward is delivered due to an incorrect choice there is a much diminished response in the same direction. This suggests that when the animal makes an incorrect response in a constant environment ACh signalling is not engaged and behavioral alterations are driven by other neuromodulators.”

Reviewer comment

How does this accord with the behavioral flexibility hypothesis?

Author response

Our hypothesis is that when the animal makes a previously rewarded response, but does not receive a reward, ACh signals the condition of “unexpected non-reward”. Thus, the ACh signal in response to non-reward is proposed to be specific to when a “previously correct” response is, without warning, suddenly not rewarded. In this sense “expected” is defined statistically by the history of previous reward or non-reward. We do not impute a subjective state of the animal. We have added a sentence to the manuscript to explain this point more clearly in the discussion (line 203):

“We propose that when the mouse makes a previously rewarded response, but does not receive a reward, ACh signals the condition of “unexpected non-reward”. Thus, the ACh signal in response to non-reward is proposed to be specific to when a “previously correct” response, without warning, ceases to be rewarded.”

Reviewer comment

Surely, acquiring performance on the maze requires learning and so some degree of flexibility?

Author response

We thank the reviewer for this comment. To be clear, we are not hypothesizing that CINs are essential for all learning but rather that they play a specific role in behavioral flexibility when the environmental contingencies change. Previously we¹ and others² have shown that a lesion of CINs does not reduce initial learning but slows switching to new responses after a rule change. We acknowledge that we did not explain this clearly in the original manuscript, and we have added the following sentence to the discussion to explain this point more clearly (line 446):

“Thus, we suggest that CINs are not essential for all learning but play a specific role in behavioral flexibility when the environmental contingencies change. This is consistent with previous work^{1,2}

showing that a lesion of CINs does not reduce initial learning but slows switching to new responses after a rule change.

Reviewer comment

2. Along these lines, what do the animals actually do when unrewarded? I was interested in two things here: (i) Is the degree of ACh response to non-reward at all modulated by ACh release in anticipation of reward? Does it differ according to the degree the animals are anticipating reward prior to non-reward?

Author response

We agree that this is an interesting question, and we have undertaken additional analyses of ACh activity preceding outcome, as previously described. These analyses show there is a modest ramping in ACh levels prior to both rewarded and non-rewarded outcomes. However, a comparison of ramping between rewarded and non-reward outcomes showed no significant difference between these outcomes, suggesting that the differences in ACh responses are not modulated by anticipation of reward. We have added the additional analysis to the manuscript and reported it as follows (line 129):

“As mice approached the outcome point of the maze, spatially averaged ACh signals increased progressively until the mouse reached the outcome zone (Extended Data Figure 3a). The increase was similar in rewarded and non-rewarded outcomes. ACh signals across trials were classified according to whether linear regression analysis showed significant positive or negative slope during the last 5 s preceding trial outcome. The greatest proportion of rewarded and unrewarded trials showed positive ramping (52% and 48%, respectively), and less commonly negative (32% and 31%, respectively) or non-significant changes (Extended Data Figure 3b, c). These distributions were not statistically different across conditions ($X^2(2) = 1.52, P = 0.46$).”

We address this again below (page 8) in response to an overlapping comment.

Reviewer comment

(ii) Is the ACh response influenced by the animals' behavioral responses after non-reward at all - e.g., degree of orienting/search or activity perhaps related to frustration? We are not told. There is considerable detail regarding the animals' behaviour up to reward delivery but not afterwards.

3. Relatedly, I found myself wondering if the change in ACH response after reversal in the presence of non-reward is in anyway[sic] related to motor activity/orienting/attentional changes and whether these can be detected in animals' behavior as a change in activity after reversal.

Author response

We have added more detail concerning the animals' behavioral responses after non-reward. In particular, we have expanded our analysis of trials after unexpected non-reward by including the following additional measures: trial duration, locomotor speed, and anticipatory licking following unrewarded trials, particularly focusing on early (lose-stay) vs. late (lose-shift) transitions. We also assessed post-outcome differences in locomotor velocity between rewarded and unrewarded trials during the reversal, as follows.

Velocity and trial duration

We assessed post-outcome differences in locomotor velocity between rewarded and unrewarded trials during the entire reversal session and found no significant differences between correct and incorrect trials relative to pre-reversal phase (Extended Data Fig. 5g). However, during the first 15-20 trials after reversal, we did observe slightly longer durations on lose-shift trials (i.e. entries to the new reward arm) compared to pre-reversal (correct) trials (Extended Data Fig. 5h), possibly indicating increased “deliberation” or “search-and-evaluate process” in adaptive decision-making (Redish, *Nat Rev Neurosci* 2016).

Locomotor speed

To address whether ACh responses were due to outcome-related motor activity, we analyzed trial-by-trial velocity during the outcome period (Fig. 4k and Extended data Fig. 6c). Our observations show that mice typically decelerated or stopped during the outcome period, with velocity near zero during the same time window in which ACh levels decreased during task acquisition (Fig. 2e) but increased during reversals (Fig. 4g). These findings argue against movement as a primary driver of the ACh signal. Importantly, the enhanced ACh signals we observe after reversal appear to be predictive of subsequent behavioral adaptation, suggesting that they may reflect more than just a reactive motor signal.

Anticipatory licking

We found a trend toward reduced anticipatory licking over time in trials following unexpected non-reward trials (Extended data Fig. 5f), suggesting a gradual decrease in reward expectation (on entries to the previously rewarded arm) as mice adapted to the new contingency.

These behavioral adjustments follow the heightened ACh increase observed prior to lose-shift decisions, supporting a potential link between ACh release and adaptive behavioral updating after feedback.

In relation to reactions immediately after unexpected non-reward we agree that linking ACh responses to behavioral state variables such as orienting, attentional engagement, or motor activity is interesting and important. However, our ability to do this is limited because of the head-fixed configuration used in our study. While we can measure locomotor activity on the ball, the mouse cannot make the head movements and postural orientations that are typical of the orienting

response. Although in principle it would be possible to measure eye movements, we are not equipped for eye-tracking and need to avoid contamination of the 2-photon signal by extraneous light sources. We have nevertheless undertaken additional analyses of locomotor activity during the outcome period and added these to the manuscript.

We have clarified these findings in the revised manuscript as follows (line 236).

“The change in ACh response after unexpected non-reward could, in theory, be related to motor activity due to orienting responses. To address this, we analyzed the trial-by-trial velocities during the outcome period and found no significant correlation between mean ACh signals and velocity at the time of no-reward (Fig. 4k). Our analysis showed that mice typically decelerated or stopped during the outcome period, with velocity near zero during the same time window in which ACh levels decreased during task acquisition (Fig. 2e) but increased during reversal (Fig. 4g and Extended Data Fig. 6c). As previously described, we also found a trend toward reduced anticipatory licking over time in trials following unexpected non-reward (Extended Data Fig. 5f), suggesting a gradual decrease in reward expectation (on entries to the previously rewarded arm) as mice adapted to the new contingency. These behavioral adjustments follow the heightened ACh increase observed prior to lose-shift decisions, supporting a potential link between ACh release and adaptive behavioral updating after feedback. These observations argue against movement as a primary driver of the ACh signal. Importantly, the enhanced ACh signals we observed after reversal appear to be predictive of subsequent behavioral adaptation, rather than just a reactive motor signal.

It is also theoretically possible that the long-latency ACh increases in response to unexpected non-reward were related to attentional shifts. If these signals were primarily driven by attentional engagement with task-relevant cues or trial events, one would expect them to be present more uniformly across different task epochs (Extended Data Fig. 6d), particularly during initial learning phases when task demands and attentional load are also high. However, our behavioral data demonstrated that the magnitude and timing of these long-latency ACh responses correlate with behavioral adaptation following unexpected non-reward, particularly during lose-shift decisions (Fig. 4i). This link between ACh dynamics and adaptive switching supports a role in learning from outcome violations as indicated by the pre-outcome anticipatory responses (Extended Data Fig. 6e), rather than purely attentional tracking of task features. In addition, the long-latency ACh increases occurred specifically in the context of contingency reversals and were absent during earlier phases of learning.”

Reviewer Comment

The increase in Ach after an incorrect response during reversal is interesting but, within-animals, how different is this response to that observed to non-reward during training and during the period immediately prior to reversal?

Author Response:

Following the Reviewer's suggestion, we compared ACh responses to non-reward before and after reversal within the same animals, using per-mouse averages. We report these in the revised result section as follows (line 226):

“Comparison of ACh responses to non-reward before and after reversal within the same animals, using per-mouse averages, revealed a significant increase in the spatially averaged ACh response to non-reward during reversal compared to non-reward during task acquisition (training i.e., early and pre-reversal periods i.e., late), confirming that this signal is not only present but more spatially widespread during behavioral switching (Extended Data Fig. 6d). Notably, the ACh increases observed in response to non-reward during acquisition, present in a subset of individual quadrats (cluster 4 in Fig. 4b-d), may reflect a preexisting substrate for outcome-related ACh modulation. This modulation appears to be amplified and more broadly recruited during reversal, when behavioral flexibility demands are heightened”

Reviewer comment

The authors state: “Thus, the direction and magnitude of ACh signals during the no-reward outcome period predicted future behavioral shifts.” Do they predict this because they are indicative of learning or because of some reaction the animals have to non-reward? There is a quite clear analysis of behavioral changes associated with reward. What would a similarly complete analysis of behavioral responses to non-reward look like and could this anticipate the responses after reversal? I suppose it is worth also considering here the role of shifts in ‘mood’ – by which I mean responses to frustrative non-reward, and in orienting rather than simply changes in behavior generally.

Author Response

We thank the reviewer for this insightful question. Our interpretation is that the ACh increases observed following unexpected non-reward reflect an early component of behavioral adaptation, likely serving as a learning signal (i.e. the cholinergic system has learned this outcome is no longer valid) rather than a direct consequence of a change in mood. However, we agree that distinguishing whether ACh changes reflect learning per se or more immediate affective/motivational reactions (e.g. frustration) that may coincide or be influenced by changes in reward prediction, is critical. To address this comment we discuss this in the revised discussion as follows (line 451):

“We speculate that the ACh increases observed following unexpected non-reward reflect an early component of behavioral adaptation, serving as a learning signal in which the CINs are signalling that the expected outcome did not occur. However, distinguishing whether ACh changes reflect learning per se or more immediate affective/motivational reactions (e.g. frustration) that may coincide or be influenced by changes in reward prediction, is critical. These mechanisms cannot

be separated with the current dataset, and further experiments will be required to investigate these aspects.”

Since our response overlaps with an earlier response made above, we note here that we also expanded our analysis of behavioral responses to non-rewarded outcomes. Specifically, we now include measures of trial duration, locomotor speed, and anticipatory licking following unrewarded trials (Extended Data Fig. 5f-h).

Reviewer Comment

4. Where[sic] changes in ACh signals observed in anticipation of reward? One assumes that the signals would have been similar prior to correct or incorrect choices if the animals were anticipating reward similarly prior to choice. However, only 1s of baseline was used prior to reward signals. Does dff change with a longer baseline?

Author Response:

To address this point, we extended our analysis window to include a 5-second pre-outcome period. Across the population, this revealed similar dF/F ramping signals during the approach phase for both correct and incorrect trials (Extended Data Fig. 3). These data indicate that outcome-evoked ACh transients were not preceded by outcome-specific anticipatory activity. We have added this observation to the revised manuscript as follows (line 129):

“As mice approached the outcome point of the maze, spatially averaged ACh signals increased progressively until the mouse reached the outcome zone (Extended Data Figure 3a). The increase was similar in rewarded and non-rewarded outcomes. ACh signals across trials were classified according to whether linear regression analysis showed significant positive or negative slope during the last 5 s preceding trial outcome. The greatest proportion of rewarded and unrewarded trials showed positive ramping (52% and 48%, respectively), and less commonly negative (32% and 31%, respectively) or non-significant changes (Extended Data Figure 3b, c). These distributions were not statistically different across conditions ($X^2(2) = 1.52, P = 0.46$).”

Reviewer #2 (Remarks to the Author):

Reviewer Comment

The manuscript describes findings from a series of experiments that demonstrates the dynamic changes in dorsal striatal acetylcholine efflux during learning and reversal learning. There are several innovations carried out in these studies that provide significant insights of how actions of striatal cholinergic interneurons underlie learning and behavioral flexibility.

At the core there are three main experiments. The first combining a genetically encoded

fluorescent acetylcholine sensor with 2-photon microscopy to determine how acetylcholine within subregions of the dorsal striatum changes during learning and reversal learning. A second study was conducted in striatal slices demonstrating that the fluorescent sensor was actually measuring acetylcholine. A third experiment employing chemogenetics selectively reduced striatal acetylcholine activity and showed that this manipulation impaired reversal learning consistent with what would be predicted from the findings in their first experiment.

What stands out about this manuscript is the creativity in carrying out their first and main experiment investigating striatal acetylcholine changes during learning and reversal learning. Integrating modern techniques the main experiment is able to reveal how dynamic changes in dorsal striatal acetylcholine output supports learning and behavioral flexibility at a spatiotemporal scale that has not been achieved before. Further, the breadth and depth of their analyses really provides a comprehensive picture of the how dynamic changes in acetylcholine occur in striatal subregions. Overall, the findings from these experiments make significant advances in revealing how striatal cholinergic interneuron activity facilitates learning and behavioral flexibility. These findings will have a broad interest for anyone interested in the brain mechanisms that support learning and behavioral flexibility.

I have two main comments related to the manuscript:

1. The authors use the term "disappointment" referring to when a mouse makes an incorrect choice that does not lead to a reward. This term really seems like a projection of a human experience onto the mouse that is not necessary. The results are extremely strong and convincing so using an alternative term that more simply captures the behavior would suffice, i.e. unexpected non-reward.

Author Response

Thank you for the suggestion. We have replaced “disappointment” with “unexpected non-reward”.

Reviewer Comment

2. A real strength of the manuscript is the hierarchical clustering of acetylcholine responses within subregions. I was not clear whether the data included in this analysis was for all trials in all sessions mice were tested or only for a subset of trials, i.e. late learning? Related, did the same clustering pattern occur throughout testing or was it more prominent in a particular stage of learning, i.e. late vs. early.

Author Response

The clustering analysis presented in the manuscript was based on session averages of all mice in the late phase of learning, specifically when animals had reached criterion performance (80% correct). To further explore how these patterns may evolve over learning, we have performed additional analyses (and clustering) on sessions from the early acquisition phase, while also acknowledging the unstable nature of learning at such an early phase. Nonetheless, while four

distinct ACh response patterns were identified during late learning (i.e., pause, burst-pause, pause-burst, and increase) (Fig. 3b-d), only two primary response clusters emerged during early learning, which we classify as long-latency pause (~21%) and short-latency pause (79%). We now include this additional analysis in Extended data Fig. 4g,h and address the reviewer's comments as follows (line 1349):

“... we classified the response patterns of all recorded quadrats (2475 quadrats from 11 mice) using principal component analysis (PCA) to reduce dimensionality followed by hierarchical clustering algorithm.”

and as follows (line 1376):

“While four distinct ACh response patterns were identified during late learning (i.e., pause, burst-pause, pause-burst, and increase) (Fig. 3b-d), only two primary response clusters emerged during early learning, which we classify as long-latency pause (~21%) and short-latency pause (79%) (Extended Data Fig. 4g,h).”

Reviewer #3 (Remarks to the Author):

Sarpong et al. investigate the spatiotemporal dynamics of acetylcholine (ACh) in the dorsal striatum, using a genetically encoded sensor in head-fixed mice choosing between left and right paths in a virtual environment. This is an important topic and they observe some interesting results. The primary finding is that under reversal conditions (reward location switches from right to left) arrival at the no-longer-rewarded location causes an increase in ACh, rather than the dip seen with rewards. The size of this no-reward increase is predictive of behavioral switching, and suppressing ACh interneurons makes switching slower. They also find limited evidence for spatial heterogeneity in the ACh response. Overall there are some noteworthy observations here, but the key claims lack adequate support and/or are over-interpreted.

One major concern is about the claimed spatial heterogeneity of ACh (which is the primary novel aspect of this work). To investigate this they divide their field of view into 50x50 micron regions of interest, and analyze them separately. They perform cluster analysis on the ACh response and claim distinct functional clusters. This was not very convincing for several reasons:

- First, there's not a clear separation between clusters. It appears more like a continuum with subtle differences between the relative heights of the increases and decreases.

Author Response

To emphasize, hierarchical clustering often reveals gradations in signal morphology rather than sharply discrete groups in complex biological datasets. To quantify separation between clusters,

we have further computed the silhouette coefficient for the hierarchical clustering using Euclidean distance, giving a mean of 0.21 and 0.15 with a range of $-0.55 - +0.64$ and $-0.44 - +1.0$ for reward and no-reward clusters, respectively. This value for the silhouette coefficient suggests that the separation between clusters is weak and there may be an overlap. We have added this analysis to the revised manuscript as follows (line 149):

“The clustering approach revealed gradations in signal morphology rather than sharply discrete groups (silhouette coefficient mean 0.21, range, $-0.55 - +0.64$ for reward clusters, and mean 0.15, range, $-0.44 - +1.0$ for no-reward clusters) indicating weak separation between clusters lying along a continuum (Extended Data Fig. 4d). This is consistent with the fact that the concentration measures in the analysis are from physically adjacent quadrats along continuous concentration gradients. These clusters provide a heuristic grouping that captures local similarity in response patterns.”

Reviewer Comment

There's no clear indication of statistical significance in the difference between response types, which is not surprising given that a) the response is highly variable from one trial to the next, and b) there are very few trials per day (~40), and that includes both rewarded and unrewarded, so the number for one category or the other in a given recording session may be very small indeed). How can the authors rule out the alternative possibility that the underlying ACh signal is exactly the same throughout the FOV, but with spatially variable noise?

Author Response

We have taken several steps to ensure that our clustering reflects meaningful functional differences. Following unsupervised hierarchical clustering, we applied strict post hoc classification criteria to validate the functional relevance of each response type (Dorst et al., *Nat Commun* 2020). For instance, a pause was deemed significant if the decrease in activity exceeded 2 standard deviations (SD) below baseline; a burst-pause required a preceding increase >2 SD above baseline followed by a pause; a pause-burst required a >2 SD burst following a pause; and an increase if the activity >2 SD above baseline. This approach ensured that cluster identities were not arbitrary but reflected statistically defined signal changes. We have explained this in the methods section as follows (line 1365):

“... we applied strict post hoc classification criteria to validate the functional significance of each response type using statistical thresholds (± 2 SD). A quadrat was classified as ‘decrease type’ if its activity did not exceed 1 SD above baseline for any time steps from 0 to 1000 ms of reward delivery, and the minimum activity was less than 2 SD below baseline. A quadrat was identified as increase-decrease if its activity exceeded 2 SD above baseline for at least one time step from 0 to 400 ms of reward delivery followed by a pause. A rebound burst after the decrease was defined as maximum activity exceeding 2 SD for at least one time step from 500 to 1000 ms of reward

delivery. A quadrat was classified as ‘increase type’ if the activity exceeded 2 SD above baseline during the time of reward delivery.”

Following this criteria, ~79% and 45% of quadrats were classified as reward and no-reward responsive, respectively (n = 2475 quadrats from 11 mice). Our revised manuscript specifies more explicitly that the differences in distribution of response types were tested with the Chi-square test, which revealed a significant difference between reward and no-reward conditions (Chi-square: $X^2(3) = 232.7$, $P < 0.0001$). We have reported these differences in the revised text and updated Figure 3 as follows (line 172):

“Overall, the proportions of the observed response types were significantly different between reward and no-reward outcomes (Chi-square test: $X^2(3) = 232.7$, $P < 0.0001$), indicating differential ACh dynamics depending on trial outcome.”

And (line 858):

“A total of 2475 quadrats were imaged in 11 mice. Of these, ~79% and 45% were classified as reward and no-reward responsive, respectively. The response types between R and NR outcomes were significantly different in proportion: $X^2(3) = 232.7$, $P < 0.0001$.”

We also observed that certain response types (e.g., pause, pause-burst) were spatially organized within the dorsal striatum — more so than expected by chance, showing that clusters were not randomly distributed across the field of view. These findings align with emerging evidence of topographic variation in cholinergic microcircuits^{3,4,5}. In brief, quadrats exhibiting decreased (pause) and burst-pause response patterns were more medially distributed ($P = 0.05$ and $P = 0.04$, respectively) and showed a trend toward anterior localization (median position, ~0.9 mm from bregma). In contrast, quadrats classified as pause-burst were preferentially located in relatively more posterior regions ($P = 0.005$) of the dorsal striatum. We have added this data to Figure 3f and to the results section as follows (line 162):

“Quadrats exhibiting decreased (pause) and burst-pause response patterns were more medially distributed ($P = 0.05$ and $P = 0.04$, respectively) and showed a trend toward anterior localization (median position, ~0.9 mm from bregma). In contrast, quadrats exhibiting pause-burst response patterns were preferentially distributed in relatively more posterior regions ($P = 0.005$) of the DS (Fig. 3f).”

For clustering, ACh responses were averaged across multiple trials within each phase (i.e., late learning), to reduce noise and focus on consistent patterns. We believe this approach enhanced the interpretability of the observed ACh responses by focusing on sessions in which animals had reached behavioral criterion, thereby ensuring that the signals were more reliably associated with

well-learned task performance. The dataset of 425 trials (reward trials, 291 and no-reward, 134) was used for the core of this analysis.

We discuss the additional analyses as follows (line 459):

“The distinct response clusters we observed during task acquisition—particularly decrease, increase, and biphasic patterns—were not only defined by statistical thresholds but also displayed non-random spatial organization within the DS. This supports the existence of spatial heterogeneity in cholinergic signaling, consistent with recent anatomical and functional studies demonstrating topographical variation in cholinergic interneuron activity and receptor distribution^{4,6,7}. These findings suggest that different microdomains within the striatum may differentially process action outcomes, potentially contributing to the flexible updating of behavior.”

Reviewer Comment

- Second, why 50x50 microns? This seems like an arbitrary choice that has nothing to do with the circuit organization of the striatum. Are the results different if this choice is different? It's not reasonable to refer to these arbitrary zones as striatal "subregions". And without some reason to choose these particular 50x50 zones, they remain a "bulk" measure of neuromodulator signaling like fiber photometry, ...

Author Response:

We agree that it is important to justify the choice of quadrat size. We have added text to the methods section to address concerns about the arbitrariness of ROI size, and conducted control analyses using different ROI sizes (50×50 μm vs 150×150 μm), as follows (line 1313):

“The use of this grid size was informed by several considerations beyond convenience. First, the choice of 50×50 μm quadrats was based on a practical compromise between spatial resolution and signal reliability. This size was chosen to ensure sufficient signal-to-noise ratio within each quadrat, while also capturing meaningful “local” variations in ACh dynamics across the FOV (Extended Data Fig. 4e). Second, CINs, which are the primary source of striatal ACh, are sparsely distributed but possess extensive axonal arborizations, spanning several hundred micrometers^{8,9,10}. These broad arbors form spatially diffuse neuromodulatory fields that overlap and influence local microcircuits. However, 2-photon microscopy optically sections the image in the dorsoventral plane, and thus samples only a fraction of the arborization of any particular neuron. In addition, some spatial averaging is required to achieve meaningful signal-to-noise ratio on a trial-by-trial basis. To ensure that our observations were not biased by our choice of quadrat size we also conducted control analyses using different quadrat sizes (Extended Data Fig. 4e).

Reviewer Comment

“...and the approach is less useful than it should be (less useful than e.g. recent micro-fiber array work such as the cited Bouabid et al reference)”.

Author Response

We respectfully but firmly disagree that 2-photon imaging is “less useful” than micro-fiber array recordings in general, and in particular for our question. Different approaches measure different things and address different questions. Individual fibers in micro-fiber array recordings sample unknown volumes because the spread of excitation is unknown, and falls off with distance from the fiber tip, while the volume of collection of fluorescence also falls off with distance from the tip. In the paper the reviewer cites, Bouabid et al *estimated* a tapered collection volume extending 100 μm axially and 25 μm radially from each fiber (thus sampling a circle of varying diameter – depending on distance from the tip – up to 50 μm in the horizontal plane). According to their estimate of location, the *minimum* separation of fiber tips in their arrays was 220 μm radially and 250 μm axially (i.e. a Euclidean distance of at least 330 μm between tips). Thus, the fiber tips do not sample contiguous volumes (by design) but rather cones of estimated maximum radius 25 μm separated by at least 330 μm . Because the tips are scattered in three dimensions throughout the volume of the striatum, these microfiber arrays are good for detecting *regional* differences but not good for measuring the spatial dimensions of ACh distribution. On the other hand, 2-photon microscopy provides gap-free measurements across a narrower field of view, which is useful for spatially continuous mapping of temporospatial dynamics with high spatial resolution in defined optical sections of known dimensions. This complements the discontinuous measurements obtained with microfiber techniques by revealing the areas that go undetected between fiber tips, and providing absolute spatially defined and localized measurements.

Reviewer Comment

Third, the images shown show clear spatial structure in the fluorescence signal. As expected for a GRIN lens the signal is much fainter at the edges - yet the edges are equally included in the analysis and seem to show similar results. Furthermore, the center of the image shows clear structure of some form... are these dendrites or what? What happens if the ROI is chosen to match the locations of expression - does the behavior-linked signal get stronger, as might be expected? I.e. how does $\Delta F/F$ depend on F ?

Author Response

We agree that the baseline fluorescence signal (F_0) is fainter at the edges of the field of view, as expected due to spherical aberrations in GRIN lens imaging. However, in our analysis, ROIs were selected based on functional responsiveness, not by F_0 . Furthermore, although we used a 1 mm GRIN lens for implantation, our imaging field (set by the objective lens) covered between 750-800 μm , and we limited our ROI selection to this central area. Thus, a large portion of the GRIN lens periphery — where optical quality is typically degraded — was not included in our analysis. Although F_0 drops off at the image periphery this does not indicate a lack of behavior-linked

activity. Since our analyses are based on $\Delta F/F$, which normalizes the signal relative to F_0 , regions with lower overall brightness can still exhibit robust $\Delta F/F$ responses. We have clarified this in the text as follows (line 331):

“To correct for spherical aberrations inherent in GRIN lens imaging, which results in lower baseline fluorescence signal (F_0) at the edges of the field of view compared to the center, we used $\Delta F/F_0$ normalization, which scales the signal relative to F_0 . To ensure that this normalization effectively accounts for baseline intensity differences and that our main findings are not simply driven by expression gradients or brightness alone, we assessed whether $\Delta F/F$ values were dependent on F_0 . We found no significant correlation between the mean of the behavior-linked signals and baseline fluorescence across quadrats (Extended Data Fig. 8d,e). Thus, regions with lower overall brightness still exhibited robust $\Delta F/F$ responses. We also verified that quadrats in peripheral and central areas show similar patterns of behavioral modulation (Extended Data Fig. 8f,g). These analyses indicate that our results are not biased by location within the field of view.”

Regarding the reviewers comment:

Furthermore, the center of the image shows clear structure of some form... are these dendrites or what?

This is addressed together with the following comment.

Reviewer comment

It was also a concern that the apparent neurites visible in the images don't seem to scale in size in a way consistent with the scale bars (e.g. comparing Fig. 2b to 2d).

Author response

We appreciate the reviewer's careful observation. The visible structure in the center of the image likely reflects neuropil and fine, unresolved processes rather than individually distinguishable dendrites or somata. This is consistent with the widespread expression pattern of iAChSnFR sensor which labels all neuronal compartments within the striatum, including somata, dendrites, axons, and neuropil. The observed features may arise from densely arborized dendritic and axonal processes of both cholinergic interneurons and other neuron types. The apparent discrepancy in neurite size between Figures 2b and 2d likely arises from differences in zoom factor and imaging purpose. Figure 2b shows a representative image with a higher magnification mainly for visualization purposes, while Figure 2d reflects the actual zoom levels used during functional imaging. The scale bars in each panel are accurate for their respective magnifications. Additionally, we note that the iAChSnFR sensor was expressed in a non-cell-type-specific manner, labeling all neuronal elements, including somata, dendrites, and neuropil. As a result, the structures observed likely reflect a mixture of cell types and compartments, which can vary in size. Therefore, direct comparison of apparent neurite dimensions across fields should be interpreted with caution,

particularly when the underlying cellular composition may differ. We have revised the figure legend to clarify these points, adding text as follows (line 803):

“Differences in apparent feature size across panels reflect varying zoom levels; all scale bars are accurate to the respective imaging conditions. Additionally, we note that the iAChSnFR sensor was expressed in a non-cell-type-specific manner, labeling all neuronal elements, including somata, dendrites, and neuropil. As a result, the structures observed likely reflect a mixture of cell types and compartments, which can vary in size. Therefore, direct comparison of apparent neurite dimensions across fields should be interpreted with caution”

Reviewer comment

Many prior studies have shown the distinct properties of different dorsal striatal subregions (defined especially along the medial-lateral axis, also the anterior-posterior axis). This aspect is not adequately considered here in the experimental design and analyses. E.g. the GRIN lens covers a fairly wide medial-lateral extent of the dorsal striatum, and based on prior work we might expect more medial parts to be more involved in reversal learning / behavioral flexibility - this should be examined.

Author Response:

We agree regional differences in function are of interest. Although our experiments were not designed to look for regional differences, in light of the reviewer’s comment, we have undertaken additional analyses to test for differences along the medial-lateral and anterior-posterior axes, to the extent possible with our limited field of view. We have added these results to the manuscript as follows (line 285):

“Examination of quadrats along the medial-lateral and anterior-posterior axes in response to unexpected non-reward revealed functional differences. In particular, quadrats with increased ACh responses to non-reward were significantly more medially distributed ($P = 0.03$; median position, 1.4 mm from midline) and showed a trend toward anterior portions of the imaging field (median position, 0.9 mm from bregma). Conversely, quadrats with decreased responses tended to be relatively more laterally distributed ($P = 0.046$; median position, 1.6 mm from midline) in the dorsal striatum (Figure 5d).”

While this anatomical differentiation was not a central focus of the current study, it offers an important layer of interpretation and has been fully incorporated into the revised discussion as follows (line 469):

“Our analyses showed that quadrats showing increased ACh responses to no-reward outcomes were predominantly located in more medial and anterior regions of the imaging field. These results align well with prior findings suggesting a stronger role for the medial striatum in feedback

processing and behavioral adaptation^{11,12,13}. This pattern is consistent with previous studies highlighting the role of the dorsomedial striatum in behavioral flexibility and outcome monitoring. Although regional differences were not the primary focus of our study, these findings suggest that the enhanced ACh responses observed during reversal may arise, at least in part, from functional subpopulations within the DS. Future studies with more targeted imaging or manipulations could further delineate these regional contributions.”

Reviewer Comment

The behavioral task is problematic in several aspects. Besides the low number of trials, just a single reversal further reduces the amount of data available for analysis.

Author Response

We acknowledge a need to better justify our choice of behavioral task, and have added the following text to the manuscript to the discussion to address the reviewer’s concern (line 502):

“Two-photon imaging imposes practical limitations on session duration due to photobleaching risk, laser heating, and imaging stability — especially in awake, behaving animals^{14,15}. Unlike brief Pavlovian trials, our VR task produces longer, continuous ACh recordings during decision-making. We therefore used fewer, longer trials, consistent with maze reversal studies that often use just two sessions: acquisition and reversal^{16,17,18}. Our task extended this by using multiple acquisition sessions, with reversal triggered only after mice reached 80% accuracy. This ensured learning was complete before reversal, thereby providing a clear contrast at the moment of contingency change. Although this design led to fewer trials per session, it enhanced the interpretability of ACh responses specifically tied to well-learned behavior. We used a single reversal to elicit responses to an unexpected change in reinforcement contingencies under well-learned conditions. We elected not to use repeated reversals, which diminish error-driven responses due to growing expectation of change^{19,20}, potentially reducing the salience of non-reward. While this limits trial count, it strengthens the relevance of the observed ACh dynamics to unanticipated outcome shifts requiring behavioral flexibility. In general, a lower number of trials (40 per session across five sessions) does not impair learning if there is sufficient interval between rewards²¹ (average of ~50 s in our task).”

Reviewer Comment

Some analyses have notable confounds - e.g. in Fig 4j. it’s not reasonable to directly compare the lose-shift and lose-stay trials in this way, because (on average) they will be happening at different times after reversal (later and earlier respectively).

Author Response

We agree that the comparison of lose-stay vs. lose-shift trials may be confounded by the timing of these trials post-reversal. To address this, we reanalyzed the data by matching trials across a

common post-reversal trial window i.e., first and last 5 trials for each condition. We found that ACh responses remained relatively greater for lose-shift trials suggesting that this difference is not solely driven by when the trials occurred. Hopefully, this approach emphasizes the distinction between a behavioral effect and a time-based confound. We have revised the relevant figure and discussion to acknowledge the temporal structure and interpret the findings with appropriate caution, as follows (Figure 4i), line 217):

“To avoid potentially confounding the effect of time with the choice of strategy after reversals, we also analyzed the data by matching trials across a common post-reversal trial window i.e., first and last 5 trials for each condition. We found that ACh responses remained relatively greater for lose-shift trials suggesting that this difference is not solely driven by when the trials occurred (Fig. 4i, right).”

Reviewer Comment

Each trial is surprisingly long - around ~35s even in the “late” phase of learning. Why are the mice so slow, and what are they doing in all this time?.

Author Response

The virtual reality Y-maze used in our task spans ~350 cm, and mice control the pace of each trial. We intentionally avoided imposing hard time constraints so as not to artificially shape behavior or exclude natural variability in approach speed, exploration or grooming. We have added further explanation and discussion of this point as follows (line 1210):

“Mice were free to control the pace of each trial without time limits. The average duration from trial start to outcome decreased from 71.1 ± 10.4 s for early sessions to 28.1 ± 2.6 s for late sessions (Fig 1h), over the 350 cm length of virtual corridors in the maze. This trial duration includes natural variations in approach speed, exploration and grooming.”

Reviewer Comment

Overall there seems to be a poor match between the behavioral task design, recording methodology and the analysis approach that precludes strong novel conclusions about striatal ACh function

Author Response

We thank the reviewer for their generally thoughtful critique and helpful comments. We believe we have addressed the all the specific concerns underlying this overall comment in full, and added clarification of the reasons for the behavioral task design and choice of the recording methodology. We have undertaken additional analyses as suggested. We feel these responses have strengthened the conclusions of the paper.

Other points:**Reviewer Comment**

To assess the imaging conditions it would also be helpful if the sensor used here was better characterized - is the preprint from 2020 still the most up-to-date reference?

e.g. line 131: “iAChSnFR is very fast, so the timing of these dips is reflective of ACh, not a sensor artifact.”

- we need specific support for these assertions.

Author Response

We have added more details, supporting the fast response time of the sensor, as follows (line 93)

“iAChSnFR is derived from a microbial periplasmic binding protein (PBP), and like other sensors derived from PBPs²² has rise and decay kinetics on a millisecond time scale thus permitting useful imaging at >1 kHz. Previous studies using purified iAChSnFR and stopped-flow experiments confirmed that activation has a time constant of ~140 ms at 10 μM ACh and just a few ms at 1 mM ACh, with k_{on} of $0.62 \mu\text{M}^{-1}\text{s}^{-1}$ and k_{off} of 0.73s^{-1} ^{23,24}.

Reviewer Comment

Discussion is often repetitive and/or confusingly written.

Author Response:

We agree and we have revised the discussion to reduce repetition and clarify conceptual points.

Reviewer Comment

Some examples:

- line 270 “subregions responded to various combinations of expected and unexpected rewards and non-rewards associated with both shift and stay responses. “

Author Response:

We have reworded the sentence as follows (line 385):

“Dividing the field of view into a matrix of quadrats revealed non-uniform spatiotemporal dynamics of ACh release. Different quadrats showed differential responses to rewards, non-rewards after errors, and unexpected non-rewards.”

Reviewer Comment

- line 285 “However, these behavioral tasks did not provide a reward alternative

Author Response:

The sentence has been rewritten to more directly contrast our task with those lacking alternative choices, in order to better highlight the importance of behavioral flexibility, as follows (line 401):

“However, unlike Pavlovian conditioning paradigms, our task incorporates two active choice outcomes, allowing direct assessment of behavioral flexibility through action selection.”

Reviewer Comment

”- the two sentences on lines 293-297 seem to say the same thing.

- too many mentions of “the causal relationship” between ACh and lose-shift behavior

Author Response

These sentences have been rewritten to avoid redundancy and to streamline the interpretation of ACh modulation, as follows (line 436)

“Our observation that ACh increase predicted lose-shift switching suggests the increase not only signals the mismatch implicit in unexpected non-reward but also contributes to the behavioral change. This hypothesis is supported by our finding that lose-shift behavior was reduced after CIN inhibition, indicating a direct causal relationship between CIN activity and lose-shift behavior.”

Reviewer Comment

- mention of travelling v standing waves not clear enough for most readers, especially if they haven't read the cited paper.

Author Response

The section discussing traveling versus standing waves has been expanded slightly and reworded for clarity, as follows (495):

“Further, the spatially heterogeneous ACh responses we observed—occurring without clear propagation across the imaging field—may resemble a ‘standing wave’ profile²⁵ in which acetylcholine locally activates dopamine fibers without inducing a traveling wave of activation. This suggests that local ACh dynamics can generate spatially confined patterns of neuromodulator release, potentially contributing to functional compartmentalization within the striatum.

Reviewer Comment

In terms of interpreting the results...

- how much of the ACh increase is about the omission of an expected reward, and how much about the active presentation of unexpected non-reward cue?

Author Response:

We appreciate the Reviewer’s question and the opportunity to clarify this point. In our task, non-rewarded outcomes were signaled by an abrupt black screen, serving as an explicit cue for non-reward. While this visual signal could contribute to the observed ACh response, we believe it is unlikely to fully account for the increase. We have detailed these reasons in the revised discussion as follows (line 417):

“An alternative interpretation of the ACh response we observed in response to unexpected non-reward is the possibility that the signal indicating non-reward could act as a salient sensory cue and contribute to the ACh signal under certain contexts. However, the ACh increase we observed following non-reward was specific to the reversal phase and not present during initial acquisition — even though the same cue was used throughout. If the ACh response were primarily driven by the non-reward cue, we would expect a similar signal during task acquisition. The fact that this increase emerges specifically after reversal suggests that it reflects a mismatch between expected and actual outcomes, rather than mere cue salience. Additionally, the long-latency nature of the ACh signal (following anticipatory licking) suggests it encodes a violation of reward expectation, consistent with a prediction error-like signal. The response latency is inconsistent with a purely sensory response to the cue, which would occur more immediately after the screen goes black. Thus, it may reflect an internal (cognitive) state of expectant waiting, disrupted by its violation, which is more characteristic of a prediction error signal than a cue-driven sensory response. Furthermore, several studies of electrophysiological recordings from presumed CINs have reported a brief decrease in activity after the presentation of a visual stimulus, usually referred to as the pause response. In some cases, a rebound activation often occurred immediately after the pause^{26,27,28,29}. The timing of these responses suggests that the non-reward cue does not account for the increased ACh activity we observed in our study.”

Reviewer Comment

- line 314 “Thus, we propose that neural pathways exposed to increased ACh on unexpected non-reward may be preserved, protecting old learning” - wouldn’t that lead to **lack** of behavioral change, rather than enhanced change (switching)?

Author Response:

We agree this is unclear and have rewritten this part of the discussion as follows (line 487):

“Recent work shows that long-term potentiation of cortical inputs to striatal projection neurons requires coincidence of pauses in CINs with phasic dopamine activation⁵⁷. Thus, we propose that neural pathways exposed to decreased ACh on reward delivery – causing a pause after successfully shifting – would experience synaptic potentiation. Conversely, pathways exposed to increased ACh on unexpected non-reward would not be potentiated, preventing reinforcement of incorrect responses. This would ultimately lead to a reduction in interference from pre-existing memory¹⁶.

Our observations of spatially heterogeneous responses suggest that both processes may occur in parallel in different quadrats. Thus ACh activity in the striatum supports diverse adaptive responses, because of its spatiotemporal heterogeneity.”

Reviewer Comment

*line 63: “mice were trained in daily sessions of ~40 trials until they reached acquisition criterion (80% correct choices)”
- over what ... a session?*

Author Response

Yes, that is correct. We have added text to clarify that point (line 64):

“...until they reached acquisition criterion of 80% correct choices in at least one session”

Reviewer Comment

“A1-A5” not explained - presumably the acquisition sessions 1-5?

Author Response

Thanks for spotting this oversight. Yes, A1-A5 refers to acquisition sessions 1-5. We have clarified this in the revised text (line 765):

“A1-A5, acquisition sessions 1-5”

Reviewer Comment

*line 71: “Increases in anticipatory licking were positively correlated with advancement in the maze”
- actually what is shown is “lick index” - the proportion of licks that are to the rewarded side (if I understood correctly). So this is not quite the same as an increase in anticipatory licking.*

Author Response

Thank you for this helpful clarification. We agree that our metric reflects a bias in anticipatory licking toward the ultimately rewarded side, rather than an absolute increase in licking frequency. We have revised the text to clarify this distinction, as follows (line 1274):

“a directional preference in anticipatory licking”

Reviewer Comment

line 81: “reduced trial latency”

- latency here seems to mean duration - it would be better to just say duration, as latency is used in other work to mean e.g. time to begin a trial, rather than complete it.

Author Response

Thank you. We have replaced “latency” with “duration”.

Reviewer Comment

line 93: “The response to ACh was later blocked with neostigmine.”

- define “later”.

- why is the neostigmine trace noisier, and why does it decrease with ACh puff?

Author Response

We have defined “later” in the text as follows (line 102):

“To validate sensor specificity, we first applied exogenous ACh to brain slices in ACSF, followed by application of neostigmine (50 μ M), an acetylcholinesterase inhibitor, 15-20 minutes later, which blocked the response to ACh (Fig. 2c, $n = 4$ mice).”

We agree that in the originally submitted manuscript the trace showing the response to ACh in the presence of neostigmine appeared noisier and showed a small decrease in fluorescence following ACh puffing. On re-examination of the figure we found that the calculation of the Z-scores for these traces was based on the entire trace. Including the large response to ACh in the ACSF condition resulted in a larger standard deviation in that condition, which after conversion to a Z-score, scaled the ACSF signal down relative to the neostigmine condition, where the response was blocked, giving the appearance of more noise in the neostigmine condition. We recalculated the Z scores using the standard deviation from the mean of the baseline before ACh puff, which is the correct way to compare the two conditions. When this is done it can be seen that the noise levels are the same in the neostigmine and ACSF traces, and the decrease with the ACh puff in the neostigmine is a small deflection presumably due to a movement artifact that is hidden by the large amplitude positive response in the ACSF condition. We have corrected the figure as shown (Fig 1c).

Reviewer Comment

Fig. 2d: ROIs very hard to see.

Author Response

Thank you. ROIs have been color-coded and magnified.

Reviewer Comment

Figs 2d, e, etc.. state the number of individual trials included in each average.

Author Response

We have included the number of trials accordingly.

Reviewer Comment

line 99: “The features of these dips in ACh levels (Fig. 2f,g; dip amplitude, -1.64 ± 0.07 z-scored $\Delta F/F$; onset, 0.23 ± 0.01 s; dip latency i.e., time to dip, 0.48 ± 0.03 s, mean \pm s.e.m., $n = 11$) are consistent with pauses in tonic activity”

- what exactly makes these numbers consistent with pauses?

Author Response

We have added some text to the manuscript to clarify why we consider these numbers to be consistent with pauses, as follows (line 113):

“...the time to dip (0.48 s) is comparable to the timecourse of physiological action of ACh after a pause in firing, which has a latency of 0.40 s³⁰. This timing is also consistent with the timecourse of pauses in firing activity of CINs (onset around 0.16 – 0.20 s and duration 0.20 – 1.0 s)^{28,31,32,33} when combined with the kinetics of ACh diffusion and hydrolysis³⁴.”

Reviewer Comment

line 105: “We also noted some variability in reward responses across trials (example mouse in Fig. 2e, left), as previously reported”

- how do the authors distinguish real biological variability from trial to trial, from measurement noise?

Author Response:

We agree that it is important to distinguish real biological variability from trial to trial from measurement noise. To do so we have shown the variability observed in the ACh signal compared that with the noise measured from an ACh insensitive fluorophore (tdTomato) expressed in the same region and observed under similar conditions (Extended Data Figure 2a). We have added the following text to the manuscript (line 122):

“We also noted some trial-to-trial variability in responses to rewards (example mouse in Fig 2e, left; Extended Data Figure 2a) as previously reported³⁶, which was greater than measurement noise (confirmed by comparison with ACh insensitive signals recorded under the same conditions). These fluctuations in ACh may encode within session variations in behavioral contexts as perceived by the mouse, potentially reflecting dynamic recruitment of different cholinergic response modes depending on contextual or internal state factors.”

Reviewer Comment

Fig.3bii error bars are very small for these clusters - looks like simultaneously recorded ROIs from the same FOV are being treated as independent? would make more sense to average within clusters from the same experiment first, then calculate error bars at the animal level.

Author Response

Following the Reviewer's suggestion, we averaged the responses within clusters and calculated the mean and error bars at the animal level (Figure 3c). In addition, we also show the mean responses of each cluster and an activity heat map of all clusters (Figure 3d).

Reviewer Comment

Fig3D is hard to understand - e.g. does the cluster 2 NR 100% bar actually indicate that none of the subregions had a dip within 1 second? It would be useful to show the mean NR responses for each cluster.

Author Response

Following the statistical threshold we applied for outcome responsiveness, only 1 ROI met the criteria, showing a relatively long latency dip. We have included the mean NR responses, as suggested by the Reviewer (Extended Data Fig. 4a,b).

Reviewer Comment

Fig. 4b: could be helpful to include trials before reversal, as is standard.

Author Response

We thank the Reviewer for pointing this out. We have included trials before reversal (Figure 4b).

Reviewer Comment

*line 141: "The proportion of perseverative errors was $17.1\% \pm 2.7\%$ "
- here and elsewhere, need to specify exactly which trials/sessions are included.*

Author Response

We have added information on which the trials/sessions are included (line 188).

Reviewer Comment

*line 152: "Incorrect responses, however, were associated with a delayed, progressive increase in the spatially averaged levels of ACh"
- what exactly makes it "progressive"? Not clear enough.*

Author Response:

We have clarified this point as follows (line 199):

“Incorrect responses, however, were associated with an increase in spatially averaged ACh levels that occurred gradually over seconds, in contrast to the rapid phasic decreases seen on rewarded trials.”

Reviewer Comment

- *how does the high variability shown in 4i relate to the outcome of the previous (and earlier) trials?*

Author Response

For reference, the previous Figure 4i is now Extended Data Fig. 6a, which shows representative examples of responses to unexpected non-reward across trials. As shown in the updated Figure, in the early trials after reversal, responses often showed a slow onset with a late peak. In contrast, responses in later reversal trials tended to have earlier peaks. We have added a brief clarification of this observation in the Results section to help contextualize the variability shown in 4i (line 221):

“...This also revealed that certain response profiles—particularly those with delayed peaks—tended to occur more frequently during early trials.”

Reviewer Comment

- *where in the FOV are the 4i examples taken from?*

Author Response

These traces are spatially averaged population responses across the entire imaged region showing the temporal profiles of ACh signals across different trials. They do not represent individual quadrats in the FOV. To clarify this we have modified the legend as follows (line 1701):

“**a.** ...Representative traces of single trial, spatially averaged population responses to no-reward outcomes (right). Dashed black line indicates no-reward event.”

Reviewer Comment

line 239: “Additionally, cumulative errors over the first 80 trials post-reversal were significantly higher in J60-treated mice compared to saline-treated controls (Fig. 6f).”

- *not clear what this additional analysis adds to the story.*

Author Response:

The cumulative error analysis provides a broader behavioral context by summarizing the overall impairment in reversal performance beyond specific trial types. While individual analyses such as lose-shift probability and error subtypes (e.g., regressive errors) offer insight into discrete behavioral strategies, the cumulative error metric captures the overall learning curve post-reversal.

The higher cumulative errors in CIN-inhibited mice reinforce the conclusion that cholinergic signalling supports flexible updating and overall task adaptation. We have clarified this rationale in the revised text, as follows (line 355):

“Cumulative errors over the first 80 trials post-reversal were higher in CIN-inhibited mice, reflecting impaired overall task adaptation beyond specific error types (Fig. 6f).”

Reviewer Comment

line 245: “While the difference in perseverative errors was not significant ($P = 0.47$, unpaired t -test), a significant difference in the proportion of regressive errors was observed between the two groups”

- need to explain “perseverative” vs “regression” and contrast to lose-stay v win-shift (seems different given Fig. 6g)

Author Response:

We agree that this was unclear in the original manuscript. We edited the relevant section and provided clearer definitions of “perseverative” vs “regression” and also clarified how these measures are related to lose-shift and win-stay trials, as follows (line 1446):

“Perseverative errors are repeated incorrect choices made after the reversal but before the mouse has exhibited learning of the correct response. Specifically, a perseverative error was defined as errors made in a block of four consecutive trials after reversal, of which 3 or more were incorrect. The measure of perseveration was the number of such errors that occurred in sequence. The end of the sequence was defined by occurrence of a block of four trials in which 2 or more trials were correct. Regressive errors are incorrect choices that occur after learning has occurred, as evidenced by a sequence of correct choices. Here a regressive error was defined as any block of four trials in which an error occurred after a block in which 2 or more trials were correct. Perseverative and regressive measures are thus distinguished from individual win-stay and lose-shift responses that are trial-by-trial choices made in the context of the outcome of the previous trial.”

Reviewer Comment

line 253: “Although our inhibition protocol lacked temporal specificity, our findings align with our initial observation of decreased switch probability in mice with reduced ACh levels.”

- the initial observation doesn't measure ACh "levels".

Author Response

We agree with the reviewer. Our initial observation measured changes in levels, not absolute levels. We have now revised the text as follows (line 370),

“Although our inhibition protocol lacked temporal specificity, the behavioral findings are consistent with the observed changes in ACh associated with lose-shift behavior”

Reviewer #4 (Remarks to the Author):

Author Response

Thank you for co-reviewing the manuscript.

References cited in response to reviewers

1. Aoki S, Liu AW, Zucca A, Zucca S, Wickens JR. Role of striatal cholinergic interneurons in set-shifting in the rat. *J Neurosci* **35**, 9424-9431 (2015).
2. Bradfield LA, Bertran-Gonzalez J, Chieng B, Balleine BW. The thalamostriatal pathway and cholinergic control of goal-directed action: interlacing new with existing learning in the striatum. *Neuron* **79**, 153-166 (2013).
3. Duhne M, Mohebi A, Kim K, Pelattini L, Berke JD. A mismatch between striatal cholinergic pauses and dopaminergic reward prediction errors. *Proc Natl Acad Sci U S A* **121**, e2410828121 (2024).
4. Gonzales KK, Smith Y. Cholinergic interneurons in the dorsal and ventral striatum: anatomical and functional considerations in normal and diseased conditions. *Ann N Y Acad Sci* **1349**, 1-45 (2015).
5. Matamales M, Gotz J, Bertran-Gonzalez J. Quantitative Imaging of Cholinergic Interneurons Reveals a Distinctive Spatial Organization and a Functional Gradient across the Mouse Striatum. *PLoS One* **11**, e0157682 (2016).
6. Duhne M, Mohebi A, Kim K, Pelattini L, Berke JD. A mismatch between striatal cholinergic pauses and dopaminergic reward prediction errors. *P Natl Acad Sci USA* **121**, (2024).
7. Matamales M, Skrbis Z, Hatch RJ, Balleine BW, Gotz J, Bertran-Gonzalez J. Aging-related dysfunction of striatal cholinergic interneurons produces conflict in action selection. *Neuron* **90**, 362-373 (2016).
8. Takagi H, Somogyi P, Smith AD. Aspiny neurons and their local axons in the neostriatum of the rat: A correlated light and electron microscopic study of Golgi-impregnated material. *Journal of Neurocytology* **13**, 239-265 (1984).
9. Chang HT, Kitai ST. Large neostriatal neurons in the rat: an electron microscopic study of gold-toned Golgi-stained cells. *Brain Res Bull* **8**, 631-643 (1982).
10. Bolam JP, Ingham CA, Smith AD. The section Golgi-impregnation procedure 3. Combination of Golgi-impregnation with enzyme histochemistry to characterize acetylcholinesterase-containing neurons in the rat neostriatum. *Neuroscience* **12**, 687-709 (1984).
11. Ragozzino ME, Mohler EG, Prior M, Palencia CA, Rozman S. Acetylcholine activity in selective striatal regions supports behavioral flexibility. *Neurobiol Learn Mem* **91**, 13-22 (2009).

12. Inokawa H, Matsumoto N, Kimura M, Yamada H. Tonicly Active Neurons in the Monkey Dorsal Striatum Signal Outcome Feedback during Trial-and-error Search Behavior. *Neuroscience* **446**, 271-284 (2020).
13. Yin HH, Knowlton BJ. The role of the basal ganglia in habit formation. *Nat Rev Neurosci* **7**, 464-476 (2006).
14. Kondo M, Kobayashi K, Ohkura M, Nakai J, Matsuzaki M. Two-photon calcium imaging of the medial prefrontal cortex and hippocampus without cortical invasion. *eLife* **6**, (2017).
15. Podgorski K, Ranganathan G. Brain heating induced by near-infrared lasers during multiphoton microscopy. *J Neurophysiol* **116**, 1012-1023 (2016).
16. Ragozzino ME, Jih J, Tzavos A. Involvement of the dorsomedial striatum in behavioral flexibility: role of muscarinic cholinergic receptors. *Brain Res* **953**, 205-214 (2002).
17. Floresco SB, Ghods-Sharifi S, Vexelman C, Magyar O. Dissociable roles for the nucleus accumbens core and shell in regulating set shifting. *J Neurosci* **26**, 2449-2457 (2006).
18. Brown HD, Baker PM, Ragozzino ME. The parafascicular thalamic nucleus concomitantly influences behavioral flexibility and dorsomedial striatal acetylcholine output in rats. *J Neurosci* **30**, 14390-14398 (2010).
19. Costa VD, Tran VL, Turchi J, Averbeck BB. Reversal learning and dopamine: a bayesian perspective. *J Neurosci* **35**, 2407-2416 (2015).
20. Dalton GL, Wang NY, Phillips AG, Floresco SB. Multifaceted Contributions by Different Regions of the Orbitofrontal and Medial Prefrontal Cortex to Probabilistic Reversal Learning. *J Neurosci* **36**, 1996-2006 (2016).
21. Burke DA, Taylor A, Jeong H, Lee S, Wu B, Floeder JR, VM KN. Reward timescale controls the rate of behavioural and dopaminergic learning. *bioRxiv*, (2024).
22. Marvin JS, *et al.* An optimized fluorescent probe for visualizing glutamate neurotransmission. *Nature Methods* **10**, 162-170 (2013).
23. Borden PM, *et al.* A fast genetically encoded fluorescent sensor for faithful *in vivo* acetylcholine detection in mice, fish, worms and flies. *bioRxiv*, 2020.2002.2007.939504 (2020).
24. Zhu PK, *et al.* Nanoscopic visualization of restricted nonvolume cholinergic and monoaminergic transmission with genetically encoded sensors. *Nano letters* **20**, 4073-4083 (2020).
25. Matityahu L, Gilin N, Sarpong GA, Atamna Y, Tiroshi L, Tritsch NX, Wickens JR, Goldberg JA. Acetylcholine waves and dopamine release in the striatum. *Nat Commun* **14**, 6852 (2023).
26. Aosaki T, Tsubokawa H, Ishida A, Watanabe K, Graybiel AM, Kimura M. Responses of tonically active neurons in the primate's striatum undergo systematic changes during behavioral sensorimotor conditioning. *J Neurosci* **14**, 3969-3984 (1994).
27. Apicella P, Legallet E, T'rouche E. Responses of tonically discharging neurons in the monkey striatum to primary rewards delivered during different behavioral states. *Exp Brain Res* **116**, 456-466 (1997).
28. Morris G, Arkadir D, Nevet A, Vaadia E, Bergman H. Coincident but distinct messages of midbrain dopamine and striatal tonically active neurons. *Neuron* **43**, 133-143 (2004).
29. Yamada H, Matsumoto N, Kimura M. Tonicly active neurons in the primate caudate nucleus and putamen differentially encode instructed motivational outcomes of action. *J Neurosci* **24**, 3500-3510 (2004).

30. Zucca S, Zucca A, Nakano T, Aoki S, Wickens J. Pauses in cholinergic interneuron firing exert an inhibitory control on striatal output in vivo. *eLife* **7**, doi: 10.7554/eLife.32510. (2018).
31. Aosaki T, Kimura M, Graybiel AM. Temporal and spatial characteristics of tonically active neurons of the primate's striatum. *J Neurophysiol* **73**, 1234-1252 (1995).
32. Schulz JM, Oswald MJ, Reynolds JN. Visual-induced excitation leads to firing pauses in striatal cholinergic interneurons. *J Neurosci* **31**, 11133-11143 (2011).
33. Sharott A, Doig NM, Mallet N, Magill PJ. Relationships between the firing of identified striatal interneurons and spontaneous and driven cortical activities in vivo. *J Neurosci* **32**, 13221-13236 (2012).
34. Nosaka D, Wickens JR. Striatal cholinergic signaling in time and space. *Molecules* **27**, 1202 (2022).

RESPONSE TO REVIEWER COMMENTS

Reviewer #1 (Remarks to the Author):

The authors have very thoroughly revised the original submission. they have added considerably more data regarding the behavior of their mice during the training trials, after choice and reward exposure and, in the process, have addressed the points I raised with regard to their original submission. I was particularly impressed by how thoroughly they addressed the points regarding the definition of cognitive flexibility here. I do agree that CINs are likely to be centrally involved in updating prior learning after contingencies have changed and I think that they have now clarified where and why those changes occur in the current task. I was also impressed by the way they have handled alternative explanations of their data. I find my prior comments have been addressed and that I have no further comments to make. From my perspective it is acceptable in its current form.

Author response

We thank the Reviewer for their thoughtful assessment and positive response to our work. We are glad that our revisions and additional analyses have addressed all concerns raised.

Reviewer #2 (Remarks to the Author):

The manuscript describes findings from a series of experiments that demonstrates the dynamic changes in dorsal striatal acetylcholine efflux during learning and reversal learning. There are several innovations carried out in these studies that provide significant insights of how actions of striatal cholinergic interneurons underlie learning and behavioral flexibility.

The original submission represented a very important set of experiments and results addressing an important issue about brain-behavioral relationships. In particular, how does cholinergic signaling in the striatum contribute to behavioral flexibility. In the revised version of the manuscript, the authors have clarified a lot of questions raised by reviewers by conducting several different analyses. These additional analyses clarify what conditions lead to changes in striatal ACh signaling and what conditions do not change striatal ACh signaling. Overall, the comprehensive analyses conducted provide some real insights and make a strong argument for striatal ACh contributing to behavioral flexibility.

Author response

Thank you for reviewing our manuscript and for the valuable feedback. We are pleased that the reviewer found the revised manuscript to be improved and insightful.

Reviewer #3 (Remarks to the Author):

Re-review of Sarpong et al.

Reviewer comment

As noted in my initial review, this manuscript includes some interesting observations, and I applaud the authors' efforts to use 2p imaging to probe the spatiotemporal organization of ACh signals. At the same time, these efforts, as currently presented, do not seem to have been particularly successful. To see this, look at the title and abstract: essentially none of the results mentioned there rely on the imaging - they could more easily have been obtained with simple fiber photometry. This feels like a missed opportunity. It's also a mismatch to how the manuscript is presented, which repeatedly emphasizes the novelty of and "new insight" gained from the imaging method. When the image is divided into 50um x 50um zones the authors observe some heterogeneity between zones. But beyond the apparent fact of heterogeneity, there's not much learned about the spatiotemporal scales or patterns of ACh release, and how these relate to anatomy or physiology. Overall the results are somewhat interesting to specialists but represent a modest advance over our prior knowledge of striatal ACh and behavioral flexibility.

Author response

We thank the reviewer for acknowledging the contribution of our 2-photon approach beyond what could be obtained with fiber photometry. We have modified the title and abstract to better match the findings, and hopefully make their significance clearer to the non-specialist, as follows:

Title: "Spatially heterogeneous acetylcholine dynamics in the striatum promote behavioral flexibility"

Abstract: (line 28): "Analysis of 2-photon images by tiling the field of view into small quadrats revealed spatially inhomogeneous and temporally distinct acetylcholine signals within the dorsal striatum. Distinct patterns of outcome-related responses were differentially distributed across the striatum, revealing functionally diverse acetylcholine microdomains."

Reviewer comment

Other points:

line 37: "more formally..." - this doesn't seem especially more "formal" than the prior sentences.

Author response

We have deleted "more formally" from the text (line 39).

Reviewer comment

line 65: why was the reversal performed between, rather than within, sessions? Wouldn't within-session provide more power to detect neural changes (although the low number of trials may

cancel this out).

Author response

In principle, a within-session reversal design might provide more statistical power for detecting neural changes. However, in the present study we opted for between-session reversals due to practical constraints of two-photon imaging. Specifically, the duration and number of trials per session are limited by exposure (laser heating) and photobleaching of the ACh sensor and by the need to maintain stable optical access and imaging quality across the session. These factors reduce the feasibility of including a sufficient number of pre- and post-reversal trials within a single imaging session. As the reviewer notes, the limited trial count would likely offset the potential power gains of a within-session design.

We agree that exploring within-session reversals would be valuable, and we view our current approach as a first step toward characterizing striatal ACh dynamics during behavioral flexibility. We hope that future studies, potentially using complementary methods with higher throughput (e.g., fiber photometry or miniature endoscopes), will build on our findings and test within-session reversal designs.

Reviewer comment

line 69: “average percent correct was 82.1%...” when? during the sessions defined as happening after an 80% correct session?

Author response

We appreciate the opportunity to clarify this point. The reported value of 82.1% correct corresponds to the **average percent correct choices in the late learning phase across mice**, specifically during the **last session preceding reversal**. We have revised the text to explicitly state this to avoid confusion. We now state that as follows (line 74):

“In the late learning phase, defined as the final session preceding reversal, mice performed with an average correct choices across subjects of $82.1\% \pm 0.9\%$ (mean \pm s.e.m, $P < 0.0001$, $n = 11$ mice)”

Reviewer comment

line 116: “...time to dip (0.48s) is comparable to the timecourse of physiological action of ACh...” why is the response time of striatal neurons a reasonable comparison, given time taken for GPCR-mediated action of ACh?

Author response

We agree this text may cause confusion and we have deleted it.

Reviewer comment

line 126: “(confirmed by comparison with ACh insensitive signals...”. The tdTomato signal is barely used in the manuscript - all the main results should include comparison to tdTomato to help rule out contributions of movement artifacts etc.

Author response

The tdTomato experiment was added in response to this reviewer’s previous comment “*how do the authors distinguish real biological variability from trial to trial, from measurement noise?*” To demonstrate the actual measurement noise for the reviewer we showed the signal from an ACh insensitive fluorophore (tdTomato) expressed in the same region and observed under similar conditions. We did not record tdTomato and iAChSnFR simultaneously in all trials because excitation of each fluorophore requires different laser wavelengths and applying both lasers at the same time increases the power and potential for overexposure, bleaching, and heating of the tissue. Therefore, tdTomato was imaged separately in a subset of trials. It should be noted that unlike isosbestic emission commonly used in fiber photometry the tdTomato signal is not emitted from the same sensor molecules: tdTomato is expressed in different subcellular locations and therefore is not suitable for subtraction from signal. As noted in the Methods, all images were processed for x-y motion correction using a standard normalized cross-correlation method to eliminate any spatially uniform motion and movement artifact. Being able to see the detailed field of view is one of the advantages of two-photon imaging over fiber photometry, and our approach is consistent with prior two-photon imaging studies (e.g., Patriarchi et al., *Science*, 2018; Sun et al., *Nat. Methods*, 2020). We have clarified this in the revised text as follows (line 131):

“...(confirmed by comparison with ACh-insensitive tdTomato signals; Extended Data Fig. 2). tdTomato was imaged in a subset of trials and did not display the large positive-going transients observed with iAChSnFR, supporting the conclusion that our reported ACh signals reflect genuine cholinergic activity rather than measurement noise^{1,2}.”

Reviewer comment

line 151: “...indicating weak separation between clusters lying along a continuum (Extended Data Fig. 4d). This is consistent with the fact that the concentration measures in the analysis are from physically adjacent quadrats along continuous concentration gradients“

- if true, this seems like an acknowledgement that clustering doesn’t really make sense here.
- although, the quadrats within the clusters don’t seem to generally appear physically contiguous, at least in the examples shown. Is this because the real spatial scale of ACh signals is comparable (or smaller) to 50µm? Or because the statistical criterion - an arbitrarily chosen 2SD - is not rigorous enough? Not enough is done to address such concerns.

Author response

As we have previously emphasized in the manuscript, the clustering results indicate gradations rather than sharply discrete groups. This result does not undermine the validity of the approach; rather, it reveals that the ACh signal is inhomogeneous and graded in the dorsal striatum, like the topography of many natural landscapes. Hierarchical clustering is an **unsupervised, unbiased method** well-suited for exploring datasets where the true underlying structure is unknown. In many biological systems, activity patterns exist on a **continuum rather than as discrete categories**, and hierarchical clustering allows the visualization of such graded relationships while still grouping local similarities (Xu & Wunsch, *IEEE Trans Neural Netw.*, 2005). Thus, the finding of weak separation is an informative result that indicates that ACh signals across quadrats are better conceptualized as varying along a gradient, rather than forming categorical “types.”

It is also important to note that structurally the cell bodies of CINs in the dorsal striatum are not clustered but have an inhomogeneous random spatial distribution, as we^{3,4} and others⁵ have shown. Each CIN produces an estimated 55 mm of axon⁶ that arborises profusely to fill a spheroidal volume with long axis of 500-700 micrometers^{7,8,9}. Therefore, the axonal arborizations individual CINs extend over multiple 50 μm x 50 μm quadrats. Given that with 2-photon imaging we can reliably detect ACh efflux from single action potentials in individual CINs using similar-sized regions of interest¹⁰, the sampling spatial frequency obtained by using 50 x 50 μm quadrats is appropriate^{11,12,13}.

As we have previously suggested, macroscopic features such as travelling and stationary waves can arise from the dynamical interactions of multiple CIN axon arborizations overlapping in the same volumes¹⁰. While we agree that it would be of interesting to characterize the spatial spread, contours and dimensions of these features, our study was not designed to do this but rather focused on the dynamics associated with behavioral flexibility.

Regarding the reviewer’s concern about spatial contiguity: it is not surprising that quadrats from the same cluster are not always physically adjacent given that interactions among CINs can occur over distances greater than 50 μm ¹⁴ and corticostriatal afferents typically project to multiple, widely separated striatal locations^{15,16}.

Regarding 2SD threshold we applied – this was not “arbitrarily chosen”. It is a standard criterion widely used to assess signal selectivity and confidence in calcium and neurotransmitter imaging studies (e.g. Chen et al., *Nature*, 2013)¹⁷. This threshold ensures that identified responses are reliably distinct from baseline fluctuations while avoiding overfitting or false positives.

In summary, the clustering objectively reveals **local similarities in response morphology** and shows that ACh release patterns in the striatum exist along a continuum rather than in discrete categories. This aligns with the broader literature, where clustering is often used not only to

define strong categorical differences, but also to explore and describe **heterogeneous and graded neural response patterns** (Cohen & Kohn, *Nat. Neurosci.*, 2011)¹⁸.

We have added this clarification as follows:

Line 156:

“Hierarchical clustering provides an unsupervised and unbiased method, heuristic framework to capture local similarities and to visualize graded relationships, even in the absence of sharply discrete groups.”

Line 1459:

“The 2SD threshold applied here is a standard criterion for defining reliable responses in neural imaging¹⁷.”

Line 481:

Structurally, the cell bodies of CINs in the dorsal striatum are not clustered but have a random spatial distribution^{3,4,5}. The axon of individual CINs arborises profusely to fill a spheroidal volume with long axis of 500-700 micrometers^{6,7,8,9}. Therefore, the axonal arborization of each CIN would extend over multiple 50 μm x 50 μm quadrats. This might explain why the clustering results indicate gradations rather than sharply discrete groups. On the other hand, the clusters are not necessarily due to anatomical structures but may be dynamically generated. Macroscopic features such as travelling and stationary waves can arise from the dynamical interactions of multiple CIN axon arborizations overlapping in the same volumes¹⁰. Finally, given that interactions among CIN axons can occur over distances greater than 50 μm ¹⁴ and corticostriatal afferents typically project to multiple striatal locations^{15,16} it is not surprising that quadrats from the same cluster are not always physically adjacent.

Reviewer comment

line 202 “...increase in spatially averaged ACh levels that occurred gradually over seconds...”. This is certainly an interesting observation, although given that the size of the increase is very modest (<1 Z) and ACh can change very fast, it’s harder to have confidence this is a discrete, specific signal being used by the brain.

Author response

In addition to the observed signal, the evidence suggesting this signal is used by the brain is that it is predictive of the lose-shift behavior, and blocking CIN activity reduces lose-shift behavior. While we also agree that ACh concentration can change rapidly, that does not preclude slow change and operation on multiple time scales, as we have shown here (Extended Data Fig. 6a).

Reviewer comment

line 232: “...confirming that this signal is not only present but more spatially widespread during

behavioral switching (Extended Data Fig. 6d).”

- this figure panel is important enough to the authors’ argument to be in the main figures.

Author response

We thank the reviewer for this helpful suggestion. We agree that this panel provides important support for our central argument. A related version of this analysis has been included in the main figures (Fig. 4g,h), which compares ACh dynamics before and after reversal and thus directly addresses changes across training phases. The Extended Data panel (Extended Data Fig. 6d) was added in response to Reviewer 1’s comment to illustrate phase-specific differences in greater detail. To avoid redundancy while still highlighting the dynamic changes during, before and after task switching, we have clarified in the text such that the Extended Data Fig. 6d panel specifically emphasizes the differences (line 238). This way, both figures highlight complementary aspects of the data without duplication.

“Comparison of ACh responses to non-reward before and after reversal within the same animals, using per-mouse averages, revealed a significant increase in the spatially averaged ACh response to non-reward during reversal compared to non-reward during task acquisition (training phase *i.e.*, early, and pre-reversal periods *i.e.*, late), confirming that this signal is not only present but more spatially widespread during behavioral switching (Fig. 4g,h and Extended Data Fig. 6d).”

Reviewer comment

line 233: “...the ACh increases observed in response to non-reward during acquisition, present in a subset of individual quadrats (cluster 4 in Fig. 4b-d), may reflect a preexisting substrate for outcome-related ACh modulation”. If this is correct, would it not imply that the specific quadrats with increases during acquisition would also show (bigger) increases during reversal? Is this true?

Author response

The quadrats with increases during acquisition may be the forerunners. We do not imply they increase their activity further during reversal, rather that they show pre-existing increases. As shown in Fig. 5b–d, there is a clear population-level shift in quadrat responses from few increases and predominantly decreases during acquisition to predominantly increases during reversal, indicating that other quadrats become active later.

Reviewer comment

More generally, how consistent are the quadrat-level signals from one imaging day to the next? Showing consistency could boost confidence in the quadrat-level results. Determining the spatial scale of such consistency might allow the authors to pick a quadrat size in a principled manner.

Author response

As shown in Extended Data Fig. 4g,h, we examined quadrat-level response patterns across different stages of learning. During early acquisition, only two main response motifs emerged (short-latency pause, ~79%; long-latency pause, ~21%), whereas during late learning four distinct patterns were observed (pause, burst-pause, pause-burst, and increase). Comparison on the 2D score plot revealed that the emergence of these additional motifs appears to arise from within the same space and boundaries of the early-phase responses, where reward-decreased cells reduce in proportion to give rise to more differentiated motifs. We speculate that early-phase motifs reflect heightened sensitivity to outcome novelty or reward prediction error, consistent with prior reports of strong pause responses to unexpected rewards or salient cues in CINs^{19,20}. With learning, the additional motifs likely represent more specialized roles in expectation tracking and behavioral updating. Although we did not explicitly quantify spatial consistency of individual quadrats across imaging days, the reproducibility of these motifs across sessions and mice supports the reliability of our quadrat-based approach.

Reviewer comment

line 239: “The change in ACh response after unexpected non-reward could, in theory, be related to motor activity due to orienting responses. To address this, we analyzed the trial-by-trial velocities ...”

- velocities are not the same as orienting.

Author response

While we agree that locomotor velocity is not equivalent to orienting per se, we included this data in response to a comment from Reviewer 1 who wondered “...if the change in ACh response after reversal in the presence of non-reward is related to motor activity/ orienting/ attentional changes?” Because mice were head-fixed in our setup, they could not make postural or head- movements typical of orienting responses. Locomotor velocity on the ball was therefore the only measurable motor activity. Reviewer 1 was happy that we report that. However, given the concern raised, we have modified the text as follows:

Line 248:

“The change in ACh activity after unexpected non-reward could, in theory, be related to motor activity due to orienting responses. In our setup, mice were head-fixed and therefore unable to perform the postural or head movements typical of orienting responses. However, they could express some degree of orienting through changes in locomotor activity. We analyzed locomotor velocity on the ball on a trial-by-trial basis during the outcome period and found no significant correlation between mean ACh signals and velocity following non-reward (Fig. 4k).”

Reviewer comment

line 288-294: The new results about subregions are interesting and relevant and as the authors note they are consistent with prior results, though it’s surprising that these findings barely, or do

not, reach significance (and are not visible in the “3D map”) What if “quadrats” are dropped and the authors analyze much broader fields-of-view?

Author response

Our analysis along the medial–lateral axis did reveal a significant bias for quadrats with increased ACh responses to unexpected non-reward toward the medial striatum (Pearson’s $r = -0.55$, $P = 0.03$, simple linear regression). These findings are consistent with prior reports linking medial striatum ACh signaling to reversal learning. We therefore do not interpret the results as “barely” reaching significance but rather as reflecting a reproducible trend aligned with existing literature. Broader averaging across the entire field of view would obscure the regional differences we observed. Finally, the GRIN lens positioning provided coverage of the dorsal striatum sufficient to capture the medial–lateral and anteroposterior distributions, giving us confidence in the spatial patterns reported (Fig. 5d).

Reviewer comment

line 348: “...hM4D(Gi)-mCherry bilaterally into the DS of ChAT-Cre mice...”

the “DS” here is actually “DMS”, which makes sense given prior results but the imaging field is not consistently DMS - why?

Author response

We thank the reviewer for noting this. Our imaging experiments via the GRIN lens covered dorsal striatal regions that included both medial and lateral portions of the DS, whereas the chemogenetic manipulations targeted primarily the dorsomedial striatum (DMS). This focus was guided by two considerations: first, our imaging analyses revealed that increases in ACh responses to unexpected non-reward were most prominently distributed medially; second, prior pharmacological and lesion studies have consistently implicated the medial striatum in reversal learning and flexible behavior^{21,22,23}. However, we do not view the term “DS” or the injection sites as a discrepancy, but rather as a targeted experimental design choice that complements our imaging findings and leverages prior literature to test the functional role of medial striatal ACh signaling.

We have added this to the Methods section as follows (line 1347):

“Although injections were performed bilaterally in the dorsal striatum, they were centered primarily in the dorsomedial striatum (DMS), guided by both our imaging findings of enhanced medial ACh responses to non-reward and prior reports implicating the medial striatum in reversal learning and behavioral flexibility.”

Reviewer comment

line 360: “Cumulative errors over the first 80 trials post-reversal were higher in CIN-inhibited mice, reflecting impaired overall task adaptation beyond specific error types (Fig. 6f).” How

does this observation support this conclusion?

Author response

Our conclusion that cumulative errors over the first 80 trials post-reversal reflect impaired overall task adaptation is supported by two complementary observations. First, CIN-inhibited mice showed a significant reduction in correct choices specifically during the early reversal phase (Fig. 6d,e), when flexible updating is most critical, whereas performance in later sessions was unaffected. This indicates that CIN activity is particularly important for the initial adaptive response to changing contingencies rather than for late, steady-state performance. Second, our imaging results further support this interpretation: unexpected non-reward after reversal elicited widespread increases in acetylcholine signals, especially in lose-shift trials, where animals adapted by switching to the opposite arm. Thus, higher ACh levels were predictive of adaptive switching behavior. By contrast, chemogenetic inhibition of CINs reduced this adaptive shift, resulting in greater cumulative errors that could not be explained by specific error types alone. Collectively, these findings suggest that CIN-dependent ACh release supports early task adaptation following contingency change, and its inhibition impairs this process.

Reviewer comment

line 364: “While CIN inhibition reduced lose-shift behavior (Fig. 6g, left; unpaired t-test, $P = 0.015$), it did not alter win-stay probability following the reversal (Fig. 6g, right; $P = 0.14$). “ Judging by Fig. 6g, this is not a convincing result. The analysis appears to lack statistical power due to few, scattered data points.

Author response

Lose-shift behavior showed a robust and statistically significant reduction following CIN inhibition. We agree that we cannot exclude an effect on win-stay probability. The sample size ($n = 5$ per group) used in our chemogenetic experiments is within the range (5-9 animals/group) commonly used in CIN manipulation studies (Okada et al., *Eur J Neurosci.*, 2015). To address the reviewer’s concern, we (i) now report exact p-values and included effect sizes (Cohen’s d to quantify the magnitude of effect) for all primary comparisons, (ii) have shown individual animal data to display the underlying distribution. These additional analyses indicated that the primary effect (e.g., reduced lose-shift probability after CIN inhibition) is consistent across subjects and of large magnitude, increasing confidence in the biological effect. For the comparisons reported in Fig. 6d,g, the effect sizes were very large (e.g., $d = (-)1.54$, $(-)1.01$, $(-)1.77$, and $(-)0.94$), respectively, indicating robust group differences despite modest power. Even in the case of a smaller effect ($d = 0.43$ and 0.21 ; Ext. Data Fig. 9b,d), the observed direction was consistent with our hypothesis. Thus, although the small sample number may limit statistical power, the effect size analysis suggests that the observed differences are biologically meaningful and reproducible.

The Methods section has been revised as follows (line 1554):

“Effect size analysis

To complement null-hypothesis significance testing and better quantify the magnitude of observed group differences, we calculated Cohen's d for all pairwise comparisons. For unpaired t -tests, d was computed as the standardized difference in means between treatment groups, using the pooled standard deviation as the denominator. 95% confidence intervals were also estimated to provide an index of precision. Effect sizes were interpreted according to conventional benchmarks ($d \approx 0.2$: small; $d \approx 0.5$: medium; $d \geq 0.8$: large; negative values in figures indicate direction of effect)²⁴. Reporting effect sizes is particularly important in studies with modest sample sizes, as it provides information about the strength and potential biological relevance of effects beyond p values alone."

The figure legend has been updated as follows (line 1094):

"...Cohen's $d = (-)1.54$, 95% CI [-2.98, -0.09]) but comparable in Test 5 ($P = 0.11$, Cohen's $d = (-)1.01$, 95% CI [-2.35, 0.32]). Data shown as mean \pm SEM. Effect sizes (Cohen's d) was calculated to quantify the magnitude of differences (see Methods for details). These values highlight that, despite the modest sample size, the observed differences represent substantial biological effects in most comparisons."

Results section has been revised as follows (line 374):

Lose-shift behavior was significantly reduced by CIN inhibition (Fig. 6g, left; unpaired t -test, $P = 0.015$, Cohen's $d = (-)1.77$, 95% CI [-3.27, -0.26]). However, CIN inhibition effects on win-stay probability following the reversal were not significant (Fig. 6g, right; $P = 0.14$, Cohen's $d = (-)0.94$, 95% CI [-2.27, 0.38]).

Reviewer comment

line 497: "we propose that neural pathways exposed to decreased ACh on reward delivery – causing a pause after successfully shifting – would experience synaptic potentiation". It's not fully clear from the figures, but it looks like decrease ACh on reward delivery is strong even at the end of behavioral acquisition - would it really be helpful to keep potentiating synapses at that point?

Author response

As noted in the preceding sentence of the discussion, long-term potentiation of cortical inputs to striatal projection neurons requires coincidence of pauses in CINs with phasic dopamine activation. Phasic dopamine release generally occurs in response to unexpected reward but not in response to reward that is expected. At the end of behavioral acquisition, it is likely that reward will be expected, and dopamine will not be released, so there will be no potentiation. To make this clearer we have revised the text as follows (line 516):

"Recent work shows that long-term potentiation of cortical inputs to striatal projection neurons requires coincidence of pauses in CINs with phasic dopamine activation²⁵, which occurs in

response to unexpected reward. Thus, we propose that neural pathways exposed to decreased ACh on unexpected reward delivery – causing a CIN pause and dopamine activation after successfully shifting – would experience synaptic potentiation. At the end of behavioral acquisition it is likely that reward will be expected, and dopamine will not be released, so there will be no potentiation. Pathways exposed to increased ACh on unexpected non-reward would not be potentiated, preventing reinforcement of incorrect responses. This would ultimately lead to a reduction in interference from pre-existing memory²⁶.”

Reviewer comment

Fig. 1c: please show the data divided into the individual sessions.

Author response

We have added information about the demarcation of individual sessions, in the figure legend (line 842):

“c, Sample choice behavior in representative trials (200 trials across 5 sessions; 40 trials per session) from an example mouse...”

Reviewer comment

Fig. 1f/g (and later figures): since the point is to compare “early” and “late”, the statistical test should incorporate both these conditions (e.g. as a factor in ANOVA) rather than running separate tests for each condition.

Author response

Response:

We thank the reviewer for this suggestion. Our original paired t-tests within “early” and “late” were intended to provide simple, within-animal contrasts for each phase. However, we agree that a model that includes Session (Early vs. Late) and Outcome (Reward vs. No reward) in the same analysis is statistically stronger. Accordingly, we have re-analyzed velocity and licking rate using a within-subject repeated-measures framework with factors Session (2 levels) and Outcome (2 levels) in ANOVA.

For anticipatory licking, this analysis revealed a significant main effect of Outcome ($F(1,20) = 5.14, P = 0.03$), indicating that mice exhibited higher anticipatory licking when expecting reward relative to non-reward outcomes. The main effect of Session did not reach significance ($F(1,20) = 3.51, P = 0.076$), and the Session \times Outcome interaction also did not reach significance ($F(1,20) = 3.78, P = 0.066$). Combined, these findings suggest that anticipatory licking is robustly driven by reward outcome across learning, with a trend toward higher overall levels in late sessions. Importantly, this confirms our conclusion that anticipatory behavior reflects sensitivity to reward expectation.

In the case of velocity, this analysis revealed a significant main effect of session ($F(1,20) = 11.69, P = 0.0027$), indicating that overall velocities increased from early to late learning. The main effect of outcome did not reach significance ($F(1,20) = 0.17, P = 0.68$), though the trend was consistent with higher velocities in rewarded trials. The Session \times Outcome interaction was also not significant ($F(1,20) = 0.28, P = 0.60$). Combined, these results indicate that the clearest effect was a robust increase in approach velocity across learning, supporting our original conclusion that motivational drive strengthens with task experience. Thus, the direction of the effect is consistent with our within-phase t-tests, which suggested higher velocity in rewarded trials during late learning. The respective figures (Fig. 1f,g and 4d,e) have been updated.

Reviewer comment

Fig. 4b : why show the orange and blue ticks twice each?

Author response

The lower raster plot shows trial-by-trial choices, where blue ticks denote correct trials and orange ticks denote incorrect trials. Above this, a heatmap depicts the same trial outcomes in binary format (1 = correct, 0 = incorrect), aligned to the raster plot for visual comparison. Thus, the *raster + binary heatmap* give complementary information: the raster conveys variability across trials, while the heatmap makes the correct/incorrect sequence categorical and easy to read.

Reviewer comment

Ex Fig 6b: why are these particular three “representative” mice shown, rather than all of them? The right part of this panel is hard to follow and the argument presented from these individual mice is unconvincing.

Author response

We appreciate the reviewer’s concern. We note, however, that the individual activity patterns and mean responses of individual mice are already presented in Fig. 4g (heatmaps) and Fig. 4j (scatter plot of lose-shift correlation). The representative traces in Extended Data Fig. 6b were included primarily as a visual illustration of how cholinergic responses to unexpected non-reward can influence choice behavior across reversal, complementing the full dataset shown in Fig. 4g,j. This approach, using representative animals to illustrate trial dynamics alongside group-level statistics, has been employed in previous studies of striatal acetylcholine and TAN activity for example^{19,27,28}. We therefore believe these examples are appropriate and serve as a heuristic aid, while the statistical conclusions are drawn from the complete dataset ($n = 11$).

Reviewer #4 (Remarks to the Author):

I co-reviewed this manuscript with one of the reviewers who provided the listed reports. This is

part of the Nature Communications initiative to facilitate training in peer review and to provide appropriate recognition for Early Career Researchers who co-review manuscripts.

Author Response

Thank you for co-reviewing the manuscript.

References cited

1. Patriarchi T, *et al.* Ultrafast neuronal imaging of dopamine dynamics with designed genetically encoded sensors. *Science* **360**, 1-8 (2018).
2. Sun F, *et al.* Next-generation GRAB sensors for monitoring dopaminergic activity in vivo. *Nat Methods* **17**, 1156-1166 (2020).
3. Carrasco A, Oorschot DE, Barzaghi P, Wickens JR. Three-dimensional spatial analyses of cholinergic neuronal distributions across the mouse septum, nucleus basalis, globus pallidus, nucleus accumbens, and caudate-putamen. *Neuroinformatics* **20**, 1121-1136 (2022).
4. Zhang R, Wickens JR, Carrasco A, Oorschot DE. Absolute Number of Thalamic Parafascicular and Striatal Cholinergic Neurons, and the Three-Dimensional Spatial Array of Striatal Cholinergic Neurons, in the Sprague-Dawley Rat. *J Comp Neurol* **533**, e70050 (2025).
5. Matamales M, Gotz J, Bertran-Gonzalez J. Quantitative Imaging of Cholinergic Interneurons Reveals a Distinctive Spatial Organization and a Functional Gradient across the Mouse Striatum. *PLoS One* **11**, e0157682 (2016).
6. Williams SR, Zhou X, Fletcher LN. Compartment-specific dendritic information processing in striatal cholinergic interneurons is reconfigured by peptide neuromodulation. *Neuron* **111**, 1933-1951 e1933 (2023).
7. Wilson CJ, Chang HT, Kitai ST. Firing patterns and synaptic potentials of identified giant aspiny interneurons in the rat neostriatum. *J Neurosci* **10**, 508-519 (1990).
8. Kawaguchi Y. Physiological, morphological, and histochemical characterization of three classes of interneurons in rat neostriatum. *J Neurosci* **13**, 4908-4923 (1993).
9. Goldberg JA, Wilson CJ. Chapter 7 - The Cholinergic Interneuron of the Striatum. In: *Handbook of Behavioral Neuroscience* (eds Steiner H, Tseng KY). Elsevier (2016).
10. Matityahu L, Gilin N, Sarpong GA, Atamna Y, Tiroshi L, Tritsch NX, Wickens JR, Goldberg JA. Acetylcholine waves and dopamine release in the striatum. *Nat Commun* **14**, 6852 (2023).
11. Nyquist H. Certain topics in telegraph transmission theory (Reprinted from Transactions of the A. I. E. E., February, pg 617-644, 1928). *Proceedings of the Ieee* **90**, 280-305 (2002).
12. Jerri AJ. The Shannon sampling theorem—Its various extensions and applications: A tutorial review. *Proceedings of the IEEE* **65**, 1565-1596 (1977).
13. Shannon CE. Communication in the Presence of Noise. *Proceedings of the Institute of Radio Engineers* **37**, 10-21 (1949).
14. Dorst MC, Tokarska A, Zhou M, Lee K, Stagkourakis S, Broberger C, Masmanidis S, Silberberg G. Polysynaptic inhibition between striatal cholinergic interneurons shapes

- their network activity patterns in a dopamine-dependent manner. *Nat Commun* **11**, 5113 (2020).
15. Faull RLM, Nauta WJH, Domesick VB. The visual cortico-striato-nigral pathway in the rat. *Neuroscience* **19**, 1119-1132 (1986).
 16. Hunnicutt BJ, Jongbloets BC, Birdsong WT, Gertz KJ, Zhong H, Mao T. A comprehensive excitatory input map of the striatum reveals novel functional organization. *eLife* **5**, (2016).
 17. Chen TW, Wardill TJ, Sun Y, Pulver SR, Renninger SL, Baohan A, Schreiter ER, Kerr RA, Orger MB, Jayaraman V, Looger LL, Svoboda K, Kim DS. Ultrasensitive fluorescent proteins for imaging neuronal activity. *Nature* **499**, 295-300 (2013).
 18. Cohen MR, Kohn A. Measuring and interpreting neuronal correlations. *Nat Neurosci* **14**, 811-819 (2011).
 19. Aosaki T, Tsubokawa H, Ishida A, Watanabe K, Graybiel AM, Kimura M. Responses of tonically active neurons in the primate's striatum undergo systematic changes during behavioral sensorimotor conditioning. *J Neurosci* **14**, 3969-3984 (1994).
 20. Apicella P. Leading tonically active neurons of the striatum from reward detection to context recognition. *Trends Neurosci* **30**, 299-306 (2007).
 21. Brown HD, Baker PM, Ragozzino ME. The parafascicular thalamic nucleus concomitantly influences behavioral flexibility and dorsomedial striatal acetylcholine output in rats. *J Neurosci* **30**, 14390-14398 (2010).
 22. Ragozzino ME, Choi D. Dynamic changes in acetylcholine output in the medial striatum during place reversal learning. *Learn Mem* **11**, 70-77 (2004).
 23. Ragozzino ME, Mohler EG, Prior M, Palencia CA, Rozman S. Acetylcholine activity in selective striatal regions supports behavioral flexibility. *Neurobiol Learn Mem* **91**, 13-22 (2009).
 24. Lakens D. Calculating and reporting effect sizes to facilitate cumulative science: a practical primer for t-tests and ANOVAs. *Front Psychol* **4**, 863 (2013).
 25. Reynolds JNJ, Avvisati R, Dodson PD, Fisher SD, Oswald MJ, Wickens JR, Zhang YF. Coincidence of cholinergic pauses, dopaminergic activation and depolarisation of spiny projection neurons drives synaptic plasticity in the striatum. *Nat Commun* **13**, 1296 (2022).
 26. Bradfield LA, Bertran-Gonzalez J, Chieng B, Balleine BW. The thalamostriatal pathway and cholinergic control of goal-directed action: interlacing new with existing learning in the striatum. *Neuron* **79**, 153-166 (2013).
 27. Morris G, Arkadir D, Nevet A, Vaadia E, Bergman H. Coincident but distinct messages of midbrain dopamine and striatal tonically active neurons. *Neuron* **43**, 133-143 (2004).
 28. Krok AC, Maltese M, Mistry P, Miao X, Li Y, Tritsch NX. Intrinsic dopamine and acetylcholine dynamics in the striatum of mice. *Nature* **621**, 543-549 (2023).